# Synthesis and In Silico Profile Modeling of 6-*O*-Fluoroalkyl-6-*O*-desmethyl-diprenorphine Analogs

**DOI:** 10.3390/ijms26199427

**Published:** 2025-09-26

**Authors:** János Marton, Dávid Gombos, Paul Cumming, Tamás Fehér, Alexander Milentyev, Beate Bauer, Frode Willoch, Bent Wilhelm Schoultz, Sándor Benyhe, Ferenc Ötvös

**Affiliations:** 1ABX Advanced Biochemical Compounds Biomedizinische Forschungsreagenzien GmbH, Heinrich-Glaeser-Strasse 10-14, D-01454 Radeberg, Germany; milentyev@abx.de (A.M.);; 2Institute of Biochemistry, HUN-REN Biological Research Center, Temesvári krt. 62, H-6726 Szeged, Hungary; gombosda@brc.hu (D.G.); feher.tamas@brc.hu (T.F.), benyhe.sandor@brc.hu (S.B.); 3Doctoral School of Theoretical Medicine, Faculty of Medicine, University of Szeged, Dugonics tér 13, H-6720 Szeged, Hungary; 4School of Psychology and Counselling, Queensland University of Technology, Brisbane, QLD 4059, Australia; paul.k.cumming@gmail.com; 5Department of Nuclear Medicine, Bern University Hospital, Freiburgstraße 18, CH-3010 Bern, Switzerland; 6Synthetic and Systems Biology Unit, HUN-REN Biological Research Center, 62 Temesvari krt., H-6726 Szeged, Hungary; 7Institute of Basic Medical Sciences, University of Oslo, P.O. Box 1110, Blindern, N-0317 Oslo, Norway; frode@willoch.no; 8Department of Physics, University of Oslo, P.O. Box 1048, Blindern, N-0316 Oslo, Norway; b.w.schoultz@fys.uio.no; 9Norwegian Medical Cyclotron Centre Ltd., Sognsvannsveien 20, N-0372 Oslo, Norway

**Keywords:** opioid receptors, 6,14-ethenomorphinans, diprenorphine, 6-*O*-fluoroalkyl-6-*O*-desmethyl-diprenorphine, in silico docking, positron emission tomography

## Abstract

We present the first preparation of novel 6-*O*-(fluoroalkyl)-6-*O*-desmethyl-diprenorphine analogs and 6-*O*-(tosyloxyalkyl)-6-*O*-desmethyl-3-*O*-trityl-diprenorphine-type precursors for the radiosynthesis of 6-*O*-([^18^F]fluoroalkyl)-6-*O*-desmethyl-diprenorphine radiotracers for molecular imaging by positron emission tomography (PET). The synthesis sequence to the new opioid receptor ligands consists of eleven steps starting from the poppy alkaloid thebaine. The precursor molecules were prepared in a three-step synthesis process from the «*Luthra precursor*» (TDDPN). We report the complete ^1^H- and ^13^C-NMR assignment of the new 6-*O*-(substituted)-6-*O*-desmethyl-diprenorphine derivatives, as well as the results of docking studies in silico for diverse novel opioid receptor ligands, including a new series of 6-*O*-(fluoroalkyl)- and 6-*O*-(hydroxyalkyl)-6-*O*-desmethyl-diprenorphine derivatives.

## 1. Introduction

The word opium [1] derives from the ancient Greek term (οπιον, opion) originally describing the juice of any plant, but today, it refers specifically to the air-dried latex juice obtained after incising of the unripe seed capsule of the opium poppy (*Papaver somniferum*) [2]. The opium poppy, which presumably originated in Asia Minor [3], is one of the few plants that was already domesticated in the neolithic period [2,4] and remains under cultivation today for its tasty seeds and oil as food and for the medicinal properties of its extruded latex. Sumerian cuneiform tablets discovered in Mesopotamia (mainly modern Iraq) describe a method for the preparation of poppy juice and present its medical indications. Archaeological, botanical, and literary [5,6] sources show that cultivation of the opium poppy was widespread in the ancient Near East, Mesopotamia, Egypt, the Mediterranean basin, and Central Europe [7,8]. Morphine [9] was first isolated from opium [10] in 1804 by the Paderborn apothecary Friedrich Wilhelm Adam Sertürner and remains to this day an important pharmaceutical for the treatment of chronic and severe pain. Since the time of Sertürner, more than eighty alkaloids have been isolated from the opium poppy [11].

Opioid receptors (ORs; μ, κ, δ, and NOP) are members of the G-protein-coupled receptor family (GPCR) of plasma membrane proteins [12,13], which have widespread expression in the central nervous system and in peripheral tissues and neurons, including the enteric nervous system. The ORs control diverse physiological and pathophysiological processes, with notable involvement in pain modulation, and are implicated in diverse neuropsychiatric disorders, such as Alzheimer’s disease (AD), epilepsy, and substance abuse. The activation of ORs by endogenous opioid peptides (EOPs) and by opiate alkaloids and several other classes of agonist drugs results in intracellular signal transduction via inhibition of the adenyl cyclase enzyme [13]. The resultant decrease in the levels of the second messenger cAMP leads to inhibition of certain calcium ion channels and activation of inwardly rectifying K^⊕^ channels and various other signal transduction pathways [14]. The resulting neuronal hyperpolarization dampens the release of various neurotransmitters (e.g., γ-aminobutyric acid (GABA), acetylcholine, dopamine, and norepinephrine), which leads to physiological effects such as euphoria, analgesia, sedation, and respiratory inhibition.

The molecular imaging of ORs by positron emission tomography (PET) [15,16,17,18,19] began four decades ago with the advent of the μ-OR selective agonist ligand [^11^C]carfentanil (**1**, [^11^C]Caf; Figure 1) [20] and the μ/κ-OR selective [^18^F]cyclofoxy ([^18^F]FcyF, **2**) [21]. Other PET tracers include the δ-selective antagonist naltrindole (NTI) derivatives developed by Lever et al., namely, *N*1′-[^11^C]methyl-naltrindole (**3**, [^11^C]MeNTI) [22] and *N*1′-(2-[^18^F]fluoroethyl)naltrindole (**4**, [^18^F]FE-NTI, BU97001) [23]. Ravert et al. [24] reported the radiosynthesis of the κ-OR selective agonist arylacetamidopiperazine derivative (*R*)-(-)-[^11^C]GR103545 (**5**) in 1999. Subsequently Pike et al. [25] synthesized the labeled NOP (ORL1) ligand [^11^C]NOP-1A (**6**).

Several non-selective orvinol ligands [15] have served for the PET imaging of ORs in the living brain, beginning with the [^11^C]-labeling of *N*^17^-cyclopropylmethyl-dihydronororvinone in *position-20* with [^11^C]methyllithium, as reported by Burns et al. in 1984 [26]. Subsequent years saw the development of tracers based on the 6,14-ethenomorphinan skeleton (Figure 2). For brain imaging by single-photon emission computer tomography (SPECT), Tafani et al. [27] synthesized the gamma-emitting *position-2* [^125^I]-labeled diprenorphine. Several *position-N*^17^ labeled diprenorphine derivatives (Figure 2) for PET imaging were synthesized from *N*^17^-*nor*-diprenorphine: [*N*^17^-cyclopropylmethyl-^11^C]-nordiprenorphine [28], *N*^17^-(2-[^18^F]fluoroethyl)-nordiprenorphine [29], *N*^17^-(3-[^18^F]fluoropropyl)-nordiprenorphine [30], and *N*^17^-((*S*)-3-[^18^F]fluoro-2-methylpropyl)-nordiprenorphine [29].

Since the publication of the fundamental study by Lever et al. [31], there have been numerous efforts to develop metabolically more stable orvinol-type tracers labeled in position-6, such as 6-*O*-([^125^I]iodoallyl)-6-*O*-desmethyl-diprenorphine ([^125^I]-*O*-IA-DPN) [32]; [^11^C]-labeled orvinols, such as [^11^C]diprenorphine ([^11^C]DPN, **7**) [31,33], [^11^C]buprenorphine ([^11^C]BPN, **8**) [33], and [^11^C]phenethyl-orvinol ([^11^C]PEO, **9**) [38]); and also [^18^F]fluoroethyl orvinol derivatives, such as (6-*O*-(2-[^18^F]fluoroethyl)-6-*O*-desmethyl-diprenorphine ([^18^F]FE-DPN, **10**) [34,35,36,37,39,40], 6-*O*-(2-[^18^F]fluoroethyl)-6-*O*-desmethyl-buprenorphine ([^18^F]FE-BPN, **11**) [39], and 6-*O*-(2-[^18^F]fluoroethyl)-6-*O*-desmethyl-phenethylorvinol ([^18^F]FE-PEO, **12**)) [39].

6,14-Ethenomorphinans are semisynthetic, thebaine, and/or oripavine derivatives that possess a ring-C 6,14-etheno bridge originating from a stereoselective [4+2] cycloaddition reaction of different asymmetric substituted olefinic dienophiles (e.g., acrolein and methyl vinyl ketone) to morphinan-6,8-dienes (e.g., thebaine and oripavine) [19,41,42,43,44,45]. The first members of this family of Diels–Alder adducts were prepared by Sandermann [46] and Schöpf et al. [47] in the 1930s. Starting in the mid-1950s, Bentley and associates made such substantial increments in the synthesis of 6,14-ethenomorphinans that they are often referred to as Bentley compounds [41]. 6,14-Ethenomorphinans [48] and their fluorinated analogs [49] are important targets of current investigations for the discovery of novel potent and more selective OR ligands, with potential for application as PET ligands. Very recently, we provided a comprehensive survey of the organic chemistry of 6,14-ethenomorphinans dating back nine decades [19]. Our longstanding program on the synthesis of labeled 6,14-ethenomorphinans for molecular brain imaging has resulted in numerous tracers: [^11^C]PEO [38], [^18^F]FE-DPN [34,35,36,37,39,40], [^18^F]FE-BPN [39], and [^18^F]FE-PEO [39,50].

In this paper, we describe the organic synthesis of a novel type of OR ligand with fluoroalkyl and hydroxyalkyl substituents in position-6 of the 6,14-ethenomorphinan scaffold. We prepared 6-*O*-(fluoroalkyl)-6-*O*-desmethyl-diprenorphine (**28a**–**d**, (CH_2_)_n_F, *n* = 2–5) and 6-*O*-(hydroxyalkyl)-6-*O*-desmethyl-diprenorphine derivatives (**29a**–**d**, (CH_2_)_n_OH, *n* = 2–5) in multi-step synthesis processes starting from the well-known «Luthra precursor» (TDDPN, **23**) [33]. Furthermore, we synthesized the corresponding 3-*O*-trityl-6-*O*-(tosyloxyalkyl)-6-*O*-desmethyl-diprenorphine precursors (**26a**–**d**) for the direct nucleophilic radiosynthesis of 6-*O*-([^18^F]fluoroalkyl)-6-*O*-desmethyl-diprenorphine ([^18^F]**28a**–**d**, (CH_2_)_n_F, *n* = 2–5) derivatives. We also performed in silico docking analyses of 6-*O*-(fluoroalkyl)- and 6-*O*-(hydroxyalkyl)-diprenorphine derivatives (n = 2–5), together with further 6,14-ethenomorphinans and another relevant OR ligands.

## 2. Results and Discussion

### 2.1. Chemistry

#### 2.1.1. Diprenorphine

Diprenorphine (**17**; DPN Revivon^®^, RX-5050M), a prominent representative of the orvinol family of OR ligands, is a semisynthetic thebaine derivative belonging to the class of ring-C bridged morphinan derivatives (6,14-ethenomorphinans, or Bentley compounds) [19,41]. According to Chemical Abstracts, diprenorphine (**17**) has the systematic name of (5*R*,6*R*,7*R*,9*R*,13*S*,14*S*)-(5α,7α)-17-cyclopropylmethyl-4,5-epoxy-18,19-dihydro-3-hydroxy-6-methoxy-α,α-dimethyl-6,14-ethenomorphinan-7-methanol (CAS RN [14357-78-9]). DPN (**17**) is a non-selective OR antagonist that is approximately 100 times more potent than *N*^17^-allyl-normorphine (nalorphine), with the following nanomolar range affinities: K_i_ (μOR) = 0.07 nM, K_i_ (δOR) = 0.23 nM, and K_i_ (κOR) = 0.02 nM [51]. [^3^H]DPN ([^3^H]**17**) is used for binding studies in vitro due to its high affinity. DPN (**17**) serves in veterinary medicine for remobilizing large animals (elephants, lions, tigers, and rhinoceros) previously sedated with the ultra-potent OR agonist carfentanil (Caf) or etorphine (Immobilon^®^).

#### 2.1.2. Synthesis of Diprenorphine

An eight-step procedure for the synthesis of diprenorphine (**17**) was already described by Bentley et al. in the late 1960s (route: **13** → **14** → **15** → **16a** → **16b** → **16c** → **16d** → **16e** → **17**; Figure 3) [52,53,54]. As noted above, the characteristic sub-structural unit of **17**, the 6,14-*ethano*-bridge in ring-C, originates from a Diels–Alder cycloaddition of asymmetric substituted ethylene dienophiles into the conjugated morphinan-6,8-dienes. In the first step of the synthesis, thebaine (**13**) is reacted in a steroselective [2+4] cycloaddition reaction with methyl vinyl ketone to result in a mixture of the 6,14-*etheno*-bridged 7α-acetyl- (**14**, 7α-thevinone; 98%) and 7β-acetyl derivatives (7β-thevinone; 2%) [52]. Thereafter the Δ^6,14^-double bond is hydrogenated by heterogenous catalytic reduction (H_2_, 10% Pd-C, EtOH, 4 bar, 50 °C) to dihydrothevinone (**15**). The Grignard addition of methylmagnesium iodide to **15** gives 20-methyl-dihydrothevinol (**16a**). Next, *N*-demethylation by application of the von Braun method via cyanamide (**16b**) results in 20-methyl-dihydronorthevinol (**16c**). The *N*^17^-cyclopropylmethyl substituent is introduced by acylation of the secondary amine (**16c**) with cyclopropylcarbonyl chloride and subsequent reduction of the *N*^17^-acyl derivative with LiAlH_4_ in THF or by direct alkylation with cyclopropylmethyl bromide. In the last step, diprenorphine-3-*O*-methylether (**16e**) is 3-*O*-demethylated with potassium hydroxide under strong/harsh conditions (diethylene glycol at 210–220 °C) to yield diprenorphine (**17**, DPN).

In the early 1990s, the Makleit research group developed several alternative routes for the synthesis of diprenorphine, buprenorphine, and their *N*^17^-substituted analogs [55,56]. In the first approach (**13** → **18a**–**b**, → **19a**–**b** → **20a**–**b** → **21** → **16e** → **17**; Figure 3), *N*^17^-formyl-northebaine (**18a**) and *N*^17^-benzyl-northebaine (**18b**) [57,58] are reacted with methyl vinyl ketone to yield the corresponding cycloadducts, *N*^17^-formyl-northevinone (**19a**) and *N*^17^-benzyl-northevinone (**19b**), respectively. The hydrogenation of **19a** in the presence of 10% Pd-C in ethanol (60 °C, 5 bar) gives *N*^17^-formyl-dihydronorthevinone (**20a**), which is converted into dihydronorthevinone (**20b**) by hydrolysis with HCl/EtOH (reflux, 4 h). Subsequently, **20b** is alkylated with cyclopropylmethyl bromide to *N*^17^-cyclopropylmethyl-dihydronorthevinone (**21**). The Grignard reaction of **21** with methylmagnesium iodide in a toluene–diethylether mixture results in 20-methyl-*N*^17^-cyclopropylmethyl-dihydronorthevinol (**16e**) in 94% yield. Thereafter compound **16e** is 3-*O*-demethylated to diprenorphine (**17**). In the second approach (route: **13** → **14** → **15** → **20b** → **21** → **16e** → **17**; Figure 3), dihydrothevinone (**15**) is *N*-demethylated with diethylazodicarboxylate (DEAD) to yield dihydronorthevinone (**20b**). Analogously to the previous route, dihydronorthevinone (**20b**) is converted into diprenorphine (**17**) via *N*^17^-alkylation, Grignard addition, and 3-*O*-demethylation.

#### 2.1.3. 6-*O*-Desmethyl-6,14-ethenomorphinans

6-*O*-Desmethyl-6,14-ethenomorphinans was unknown until the mid-1980s. The fortuitous/serendipitous discovery by Kopcho and Schaeffer [59] that the reaction of 7α-aminomethyl-6,14-ethenomorphinans with 15 equiv. LiAlH_4_ in THF containing CCl_4_ (4 equiv.) co-solvent can selectively result in the 6-*O*-demethylated derivatives was the first mention of this type of reaction: a selective 6-*O*-demethylation. Subsequently Lever et al. [31] successfully extended this method for the synthesis of orvinol derivatives. 6-*O*-Desmethyl-buprenorphine (DBPN; see Appendix A) and 6-*O*-desmethyl-diprenorphine (DDPN, **22**; Figure 4) are prepared and applied for the synthesis of PET precursor molecules: 3-*O*-*tert*-butyldimethylsilyl-6-*O*-desmethyl-diprenorphine (TBDMS-DDPN, the «Lever precursor» [31]; Figure 2) and 3-O-trityl-6-*O*-desmethy-diprenorphine (**23**, TDDPN, the «Luthra precursor» [33]; Figure 4). Despite 6-*O*-desmethyl-orvinols being successfully applied for the synthesis of PET precursors, their first biochemical investigations were only reported by the opioid research group of Benyhe [60] at HUN-REN BRC Szeged more than three decades after the first preparation of DDPN (**22**; Figure 4) by Lever et al. [31]. The 6-*O*-desmethyl derivatives (DBPN, DDPN (**22**), DDHE, and DPEO; see structures in Appendix A) of buprenorphine (BPN), diprenorphine (DPN, **17**; Figure 4), dihydroetorphine (DHE), and phenethyl-orvinol (PEO) showed higher binding affinity to μORs compared with the respective parent 6-*O*-methyl compounds [60].

#### 2.1.4. Synthesis of 6-*O*-Substituted-6-*O*-desmethyl-diprenorphine Derivatives

There are only a few reports available regarding the synthesis of 6-*O*-substituted-orvinols with something other than methyl substituents. In 1992, Musachio and Lever [32] developed 6-*O*-(3-tri-*n*-butylstannyl-prop-2-enyl)-diprenorphine for the preparation of 6-*O*-([^125^I]iodoallyl)-6-*O*-desmethyl-diprenorphine. In 2000, Wester et al. [34] described the radiosynthesis of 6-*O*-(2-[^18^F]fluoroethyl-6-*O*-desmethyl-diprenorphine ([^18^F]FE-DPN, **10**) starting from the «Luthra precursor» (TDDPN). Investigations seeking improved radiosynthesis of [^18^F]FE-DPN (**10**) via indirect radiofluorination from TDDPN were reported later by Schoultz et al. [35,39].

Subsequently, Czakó et al. [61] synthesized 6-*O*-substituted-20-methyl-orvinols (R = Et, *n*Pr, cyclopropylmethyl) starting from 6-*O*-substituted-northebaine derivatives. [4+2] Cycloaddition of 6-*O*-substituted-6,8-morphinandienes with methyl vinyl ketone gave the corresponding 6-*O*-substituted-thevinone analogs. Next, the 7α-ketons were reacted with methylmagnesium iodide to 6-*O*-alkyloxy-20-methylthevinols, which were 3-*O*-demethylated with potassium hydroxide in diethylene glycol to the target 6-*O*-substituted-20-methyl-orvinols [61]. 6-Phenyl-20-methylorvinol was also prepared from 6-bromo-6-demethoxythebaine, which is available from the poppy alkaloid thebaine in five subsequent reaction steps. 6-Bromo-6-demethoxythebaine was converted into 6-phenyl-6-demethoxythebaine in a Suzuki–Miyaura cross-coupling reaction (PhB(OH)_2_, Pd(PPh_3_)_4_, and Ba(OH)_2_·8H_2_O) in good yield. Next, the target orvinol was prepared in three consecutive steps analogously to the route described above: Diels–Alder reaction of the morphinandiene with methyl vinyl ketone, Grignard addition of MeMgI to the 7α-acetyl compound, and finally 3-*O*-demethylation [61]. In 2016, Lever et al. [62] synthesized 6-*O*-propylamido- and 6-*O*-hexylamido-[^111^In]-DOTA-diprenorphine conjugates for the SPECT imaging of ORs in a six-step procedure starting from diprenorphine (**17**).

Very recently, we developed a new precursor 6-*O*-(2-tosyloxyethyl)-6-*O*-desmethyl-diprenorphine (TE-TDDPN (**26a**), the «Henriksen precursor») for the preparation of [^18^F]FE-DPN (**10**) via direct nucleophile radiofluorination [40]. In 2023, the research group of Mikecz [37] in Debrecen performed optimalization studies of the [^18^F]FE-DPN radiosynthesis from TE-TDDPN. In the same year, the research group of Michaelides [36] at NIH performed PET experiments in mice by application of [^18^F]FE-DPN (**10**) prepared from the new precursor TE-TDDPN thorough direct one-pot, two-step nucleophilic radiosynthesis. Their PET investigation of μ-opioid receptor-knock-out (μOR KO) compared with wildtype mice confirmed that [^18^F]FE-DPN (**10**) preferentially binds to μORs in vivo, despite its incomplete selectivity in vitro. [36]

Continuing our research [36,37,40] in the field of radiolabeled orvinol derivatives, especially in the light of results by the Michaelides group concerning the unexpected selectivity of [^18^F]FE-DPN (**10**) for μORs in vivo [36], we set about to synthesize 6-*O*-fluoroalkyl-6-*O*-desmethyl-diprenorphine derivatives with longer side chains (i.e., 3-fluoropropyl, 4-fluorobutyl, and 5-fluoropentyl). This investigation also served to make 6-*O*-substituted-6-*O*-desmethyl-orvinol OR ligands for studying the influence of the 6-*O*-side chain on OR affinity and subtype selectivity.

As the starting point to our investigation, we used the «Luthra precursor» (TDDPN, **23**) [33], which was prepared from diprenorphine (**17**) in two steps. 6-*O*-(Fluoroalkyl)-6-*O*-desmethyl-diprenorphine derivatives (**28a**–**d**; Figure 5) were then prepared from TDDPN (**23**) in two steps. First, TDDPN (**23**) was alkylated with the corresponding fluoroalkyl toluene sulfonate (**33a**–**c**; Figure 6) to yield 6-*O*-fluoroalkyl-6-*O*-desmethyl-3-*O*-trityl-diprenorphine derivatives (**27a**–**d**), with yield in the range of 43–66%. Fluoroalkyl toluene sulfonate (**33a**–**c**; Figure 6) reagents were synthesized according to previously developed protocols: **33a** [63], **33b** [64], and **33c** [65]. Subsequently the trityl protecting group was removed by acidic hydrolysis with acetic acid–water (27:7 (*v*/*v*)) at 100 °C for 5 min to give the target fluoroalkylated derivatives (**28a**–**d**) in 71–78% yield.

Tosyloxyalkyl precursor molecules (**26a**–**d**) for direct nucleophilic one-pot, two-step nucleophilic radiofluorination for 6-*O*-([^18^F]fluoroalkyl)-6-*O*-desmethyl-diprenorphines ([^18^F]**28a**–**d**) were synthesized in a three-step procedure. The direct introduction of the tosyloxyalkyl side chain into position-6 by reacting TDDPN (**23**) with bistosyloxy alkanes (TosO(CH_2_)nOTos, n = 2–5) proved unfeasible in our hands [50], yielding undesired heterocyclic products. Consequently, we reverted to our previous strategy [40,50] for introduction of the desired tosyloxyalkyl side chain at the 6-*O*-position in three consecutive reaction steps (route: **24a**–**d** → **25a**–**d** → **26a**–**d**). First, the «Luthra» precursor (TDDPN, **23**) was alkylated with 2.3 equiv. of the corresponding *tert*-butyldiphenysilyl (TBDPS)-protected bromoalkanol (**34a**–**c**, Br(CH_2_)_n_OTBDPS, n = 3–5; Figure 6) in DMF in the presence of 10 equiv. sodium hydride to yield 6-*O*-(*tert*-butyldiphenylsilyloxy-alkyl)-6-*O*-desmethyl-3-*O*-trityl-diprenorphine derivatives (**24a**–**d**), with yield in a range of 40–71%. TBDPS-protected bromoalkanols (**34a**–**c**; Figure 6) were synthesized from the corresponding bromoalkanols (**30d**–**f**) with *tert*-butyldiphenylsilyl chloride (TBDPSCl) in dichloromethane in the presence of DMAP and DIPEA. Subsequently the TBDPS protecting group of compounds **24a**–**d** was removed by treatment with 1.28 equiv. of 1M tetrabutylammonium fluoride in THF (RT, argon, 4 h) to give 6-*O*-(hydroxyalkyl)-6-*O*-desmethyl-3-*O*-trityl-diprenorphines (**25a**–**d**), with 84–93% yield.

For introduction of the tosyloxy leaving group, the hydroxyalkyl-TDDPN compounds (**25a**–**d**) were reacted with 3.4 equiv. toluenesulfonic anhydride in dichloromethane in the presence of 4 equiv. pyridine (RT, 4 h) to give the desired precursors (**26a**–**d**), with 60–87% yield. Complete ^1^H- and ^13^C-NMR assignments of the 6-*O*-(tosyloxyalkyl)-6-*O*-desmethyl-diprenorphine analogs (**26b**–**d**) are presented in Table 1. 6-*O*-(Hydroxyalkyl)-6-*O*-desmethyl-diprenorphine derivatives (**29a**–**d**) were also prepared from the corresponding 6-*O*-(hydroxyalkyl)-6-*O*-desmethyl-3-*O*-trityl-diprenorphines (**25a–d**) by acidic hydrolysis of the 3-*O*-trityl protecting group with acetic acid–water 4:1 (*v*/*v*) (100 °C, 10 min). The 6-*O*-(hydroxyalkyl)- compounds were obtained with yield in the range of 72–77%.

The synthesis of fluoroalkyl-tosylates (**33a**–**c**) and bromoalkoxy-*tert*-butyl-diphenylsilanes (**34a**–**c**) was performed according to the reaction sequences depicted in Figure 6. We verified the identity of the target compounds by NMR spectroscopy (**24b**–**d, 25b**–**d**, **26b**–**d**, **27b**–**d**, **28b**–**d**, and **29b**–**d**) and by HRMS (**26b**–**d**, **28b**–**d**, and **29b**–**d**). The ^1^H and ^13^C-NMR assignments were facilitated by previously reported NMR studies of Bentley compounds [66,67,68,69]. Complete ^1^H- and ^13^C-NMR assignments of the 6-*O*-fluoroalkyl-6-*O*-desmethyl-diprenorphine analogs (**28b**–**d**) are presented in Table 2. Complete ^1^H- and ^13^C-NMR chemical shifts and assignments for 6-*O*-fluoroalyl-6-*O*-desmethyl-3-*O*-trityl-diprenorphine analogs (**27b**–**d**) and for 6-*O*-hydroxyalkyl-6-*O*-desmethyl-diprenorphine derivatives (**29b**–**d**) are presented in Appendix A.

High-resolution mass spectroscopy measurements were carried out for compounds **26b**–**d**, **28b**–**d**, and **29b**–**d** similarly to the previous method by Biri et al. in the field of buprenorphine analogs [70].

### 2.2. In Silico Studies

To investigate the binding dynamics and conformational stability of ligands from series **28** (**28a**–**d**) and series **29** (**29a**–**d**), we conducted multiple molecular dynamics (MD) simulations for each compound in complex with the receptor in its active, inactive, and partial agonist-bound conformations. The stability of the ligand–receptor complexes was assessed by monitoring two key metrics throughout the simulation trajectories: the root mean square deviation (RMSD) of the ligands’ heavy atoms and the center of mass (COM) distance between the ligand’s and the receptor’s binding sites. The ligand RMSD was calculated with respect to its initial docked pose after a rotational and translational fit to the receptor’s backbone, providing a quantitative measure of the ligand’s conformational stability within the binding pocket. A low, equilibrated RMSD value is indicative of a stable binding mode, whereas high or continuously increasing values suggest significant conformational changes or instability [71]. The COM distance was measured between the geometric centers of the ligand and a defined set of binding site residues. This metric evaluates the ligand’s positional stability; a stable COM distance signifies that the ligand remains localized, while large fluctuations can indicate movement within the pocket or dissociation events [72].

#### 2.2.1. Ligand Binding to the Active Receptor State

In the simulations involving the active receptor conformation, all ligands from both series demonstrated stable binding behavior. As shown in Figure 7, the ligand RMSD values for most compounds equilibrated within the first few frames and subsequently fluctuated around a stable mean, generally below 3.5 Å. This suggests that the ligands maintained a consistent binding pose throughout the simulation. The COM distances were similarly stable, clustering within a range of approximately 9.0 to 12.0 Å, which confirms the persistent localization of the ligands within the active site.

#### 2.2.2. Ligand Binding to the Inactive Receptor State

The simulations with the inactive receptor state revealed a significant difference in binding stability, particularly for ligand **28a** (FE-DPN; Figure 8). While most ligands exhibited stable RMSD and COM distance profiles comparable to those in the active state, ligand **28a** (FE-DPN) displayed marked instability. Its RMSD value increased sharply, reaching over 6 Å, and remained highly volatile. This conformational instability was in harmony with its COM distance, which showed large-scale fluctuations and a shift to greater distances (>13.5 Å), indicating that **28a** (FE-DPN) was unable to maintain stable pose and position within the inactive binding pocket. In contrast, the ligands from series **29** remained stable in the inactive state, with RMSD values consistently below 4.0 Å.

#### 2.2.3. Ligand Binding to the Partial Agonist Receptor State

Simulations with the partial agonist receptor state (PDB ID: 7U2K) revealed a complex and distinct dynamic profile, providing further insight into the ligands’ state selectivity (Figure 9). The most critical finding from this new set of simulations involves ligand **28a** (FE-DPN). In stark contrast to its stability in the fully active state, **28a** (FE-DPN) now displays significant conformational instability in the partial agonist conformation. Its RMSD value is highly volatile, fluctuating between 2.5 Å and 4.0 Å throughout the trajectory. Interestingly, this conformational instability is paired with a center of mass (COM) distance that is consistently lower than its series analogs, suggesting that **28a** (FE-DPN) occupies a deeper region of the binding pocket but in a strained and unstable manner. This behavior mirrors that of ligand **29**, which also exhibits a high RMSD (~4.0 Å) coupled with a similarly low COM distance. The remaining ligands in both series demonstrated relatively stable binding profiles, with COM distances clustering in an intermediate range between those observed for the active and inactive states.

#### 2.2.4. Molecular Docking Reveals a Clear Energetic Preference for the Active State

To complement the dynamic analysis from MD simulations, we first assessed the static binding predictions from molecular docking. The calculated binding affinities (in kcal/mol) for each ligand across the three receptor states—agonist (active), antagonist (inactive), and partial agonist—are presented in Figure 10. The results reveal a clear and consistent energetic hierarchy. Altogether, all compounds show a distinct preference for the active receptor state, which yielded the most favorable binding affinities for every ligand tested. In the active state, scores were consistently around −9.5 kcal/mol, with series **28** ligands showing slightly more favorable energies than those in series **29**. Within series **29**, a trend was observed where ligands with longer chains achieved somewhat better scores. A similar pattern was observed for the inactive state, which showed intermediate affinities of approximately −8.5 kcal/mol. Similarly, series **28** was slightly better than series **29**, and longer chains again appeared to confer a modest energetic advantage. The partial agonist state consistently produced the least favorable scores, ranging from −7.5 to −8.0 kcal/mol, yet the same structural trends persisted: series **28** ligands docked more favorably than series **29**, and longer-chained compounds were preferred. This overarching trend indicates that even in a static model, all ligands are energetically predisposed to bind most favorably to the fully active receptor conformation. This provides an initial, static validation for the state-selective behavior observed in the more detailed molecular dynamics simulations.

#### 2.2.5. Conclusion and Implications

The combination of molecular docking and molecular dynamics simulations provides a comprehensive and compelling rationale for the state-selective behavior of the tested ligands. The static docking scores establish an initial energetic basis for selectivity, revealing that all compounds are intrinsically favored to bind to the fully active receptor conformation. The MD simulations then provide the crucial dynamic context, illustrating how this initial energetic preference translates into conformational stability over time.

The collective computational results strongly suggest that ligand **28a** (FE-DPN) possesses a highly specific binding preference for the fully active conformation of the receptor. Its stability in the active state contrasts sharply with its profound instability in both the inactive and the partial agonist states [73]. The compounds of series **29** display more complex, state-dependent behavior. While appearing conformationally tolerant in the active and inactive states, our results show that the binding of parent compound **29** is uniquely destabilized by the partial agonist conformation. This suggests that subtle changes in OR structure can induce distinct stability profiles even within a closely related chemical series, which may translate into differences in functional activity.

The observed COM distances provide a clear structural rationale for the different functional states, revealing a gradient in the binding site topology across the receptor conformations. In general, the ligands bound most closely in the active state (~9.0–12.0 Å), at an intermediate distance in the partial agonist state (~10.5–12.5 Å), and most distally in the inactive state (~13.0 Å). Intriguingly, within this trend, the unstably binding ligands (**28a** and **29**) adopted a closer COM distance in the partial agonist state, suggesting a common mechanistic feature. This may indicate that the partial agonist binding pocket forces these ligands into a strained, high-energy conformation that is nevertheless “deep” in the pocket, a phenomenon that could be central to the mechanism of partial agonism. These findings provide a nuanced structural and dynamic rationale for the observed activities of the compounds and strengthen the case for **28a** (FE-DPN) as a potent candidate for further investigation as a μOR selective ligand.

## 3. Materials and Methods

### 3.1. General Methods

Reagents and solvents were purchased from major commercial suppliers and were used without further purification. Melting points were determined on a Büchi-535 melting point apparatus, without correction. Column chromatography (CC) was performed on Kieselgel 60 Merck 1.09385 (0.040–0.063 mm). Analytical TLC was accomplished on Macherey-Nagel Alugram^®^ Sil G/UV254 40 × 80 mm aluminum sheets (0.25 mm silica gel with fluorescent indicator) with the following eluent systems (each (*v*/*v*)): **A**: dichloromethane–methanol 9:1; **B**: hexane–ethyl acetate 7:3; **C**: hexane–ethyl acetate 1:1; **D**: ethyl acetate–methanol 8:2; **E**: dichloromethane–methanol 95:5. The spots were visualized with a 254 nm UV lamp or with 5% phosphomolybdic acid in ethanol.

Nuclear magnetic resonance (NMR) spectra were recorded on a Bruker AV 500 (Avance 500 MHz) spectrometer at 298 K, using a BBO probehead (HP workstation XW 5000; software: Bruker TOPSPIN 1.3). For ^1^H- and ^19^F-NMR experiments, 10 mg of the appropriate orvinol was dissolved in 500 μL of deuterated chloroform (CDCl_3_). We measured the ^13^C-NMR spectra after dissolving a 20 mg sample of the corresponding oripavine in 500 μL of CDCl_3_. Chemical shifts were reported as δ values in parts per million (ppm), and coupling constants (*J*) were reported in Hertz. ^1^H- and ^13^C-NMR chemical shifts were referenced to the residual peak of CDCl_3_ at δ 7.26 for proton and 77.16 ppm for carbon.

During our high-resolution mass spectroscopy (HRMS; **26b**–**c**, **28b**–**d**, and **29b**–**d**) investigation, the LC–MS system consisted of a Q-Exactive^®^ Plus mass spectrometer (Thermo Scientific, Bremen, Germany) coupled to a Vanquish Neo HPLC system (Thermo Scientific, Germering, Germany). Chromatographic separation was performed on the Ac-claim PepMap^®^ RSLC C18 column (2 μm × 15 cm × 1 mm). The mobile phases were 0.1% formic acid in water (A) and acetonitrile (B). The gradient elution was conducted by linearly increasing B from 1% to 95% over 14 min, followed by a 4 min wash and re-equilibration. The total run time was 18 min with a flow rate of 50 µL/min. Column and autosampler temperatures were 25 and 7 °C, respectively. All samples were analyzed using the positive electrospray ionization (ESI) mode. Each sample was injected in duplicate with same source settings. The source settings were optimized as follows: sheath gas flow rate, 30 arbitrary units; auxiliary gas flow rate, 15 arbitrary units; spray voltage, 3.5 kV; S-lens, 50 arbitrary units; capillary temperature, 325 °C; and auxiliary heater temperature, 75 °C. Full MS scan data were obtained from *m*/*z* 200 to 1500 at a resolution of 70,000 with the AGC target of 3 × 106, and data-dependent MS/MS spectra were acquired at a resolution of 17,500 by using stepped normalized collision energy (NCE), either 20, 25, and 30% or 45, 50, and 55, both with Top-5 MS/MS experiments. Microscans were set to 1 for both MS and MS/MS. The AGC target was set to 1 × 10^5^ and the intensity threshold to 8 × 10^3^, with the maximum injection time of 50 ms. The precursor ions were filtered by the quadrupole, which operates in an isolation window of *m*/*z* 4. All the obtained mass spectra were evaluated by Compound Discoverer 3.3 SP2 software by Thermo Fisher Scientific (Waltham, MA (Massachusets), USA).

### 3.2. Synthetic Procedures

We described the preparation of **24a**, **25a**, **26a**, **27a**, **28a**, and **29a** derivatives (n = 2, -CH_2_CH_2_-) in our previous studies [37,40,69]. For a summary of reaction conditions, yield, and physical constants, see Appendix A.

#### 3.2.1. General Procedure for the Synthesis of 6-*O*-(*tert*-Butyldiphenylsilyloxyalkyl)-6-*O*-desmethyl-3-*O*-trityl-diprenorphine (**24b**–**d**) Derivatives

To a stirred suspension of sodium hydride (60% dispersion in mineral oil, 490 mg, 12.2 mmol, 10 equiv.) in anhydrous *N*,*N*-dimethylformamide (6 mL), a solution of TDDPN (**23**; CAS RN: 157891-92-4, 800 mg, 1.22 mmol) in anhydrous *N*,*N*-dimethylformamide (7 mL) was added at 0 °C under an argon atmosphere. The reaction mixture was stirred for 30 min at the same temperature. A solution of the corresponding *tert*-butyldiphenylsilyl (TBDPS)-protected bromoalkanol (**34a**–**c**; 2.3 mmol) in dry *N*,*N*-dimethylformamide (2.5 mL) was then added dropwise. After stirring at RT for 20 h, the mixture was poured into water (50 mL). The suspension was extracted with dichloromethane (4 × 40 mL), and the combined organic extracts were washed with brine (40 mL) and dried over Na_2_SO_4_. The solvent was removed under reduced pressure, and the resulting crude product mixture was purified by column chromatography on silica gel.

(*5R,6R,7R,9R,13S,14S)-(5α,7α)-17-Cyclopropylmethyl-4,5-epoxy-6-O-(3-tert-butyldiphenylsilyloxypropyl)-18,19-dihydro-α,α-dimethyl-3-triphenylmethoxy-6,14-ethenomorphinan-7-methanol* (6-*O*-(3-TBDPSOP)-6-*O*-desmethyl-3-*O*-trityl-diprenorphine, **24b**, TBDPS-OP-TDDPN).



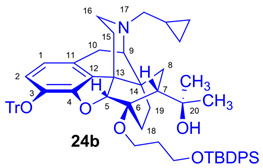



Compound **24b** was prepared from TDDPN (**23**; 800 mg, 1.22 mmol) using the general procedure in Section 3.2.1. CC: Kieselgel: 100 g; eluent: hexane–ethyl acetate 7:3 (*v*/*v*) to ethyl acetate; fractions: 50 mL. Yield: 470 mg (40%), mp. 98–102 °C, *R_f_*_A_ = 0.80, *R_f_*_B_ = 0.25, *R_f_*_C_ = 0.71. ^1^H-NMR (CDCl_3_): δ = 0.06 (m, 2H, cProp*CH*_2*syn*_), 0.42 (m, 1H, 19-H_syn_), 0.46 (m, 2H, cProp*CH*_2*anti*_), 0.76 (m, 1H, cProp*CH*), 0.89 (td, ^2^*J*_19anti,19syn_ = 13.3 Hz, ^3^*J*_19anti,18syn_ = 6.1 Hz, 1H, 19-H_anti_), 0.97 (dd, ^2^*J*_8α,8β_ = 13.3 Hz, ^3^*J*_8α,7β_ = 9.7 Hz, 1H, 8α-H), 1.03 (s, 9H, (*CH*_3_)_3_CSi(Ph)_2_), 1.15 (s, 3H, 20*CH*_3_), 1.31 (s, 3H, 20*CH*_3_), 1.38 (dd, ^2^*J*_15eq,15ax_ = 12.9 Hz, ^3^*J*_15eq,16ax_ = 2.7 Hz, 1H, 15-H_eq_), 1.43 (td, ^2^*J*_18syn,18anti_ = 13.0 Hz, ^3^*J*_18syn,19anti_ = 5.6 Hz, 1H, 18-H_syn_), 1.55 (m, 1H, 18-H_anti_), 1.73–1.78 (m, 2H, CH_2_*CH*_2_CH_2_OTBDPS), 1.79 (app t, ^3^*J*_7β,8α_ = 9.6 Hz, 1H, 7β-H), 1.87 (td, ^2^*J*_15ax,15eq_ = 13.7 Hz, ^3^*J*_15ax,16eq_ = 5.4 Hz, 1H, 15-H_ax_), 2.04 (dd, ^2^*J*_10α,10β_ = 18.3 Hz, ^3^*J*_10α,9α_ = 6.4 Hz, 1H, 10α-H), 2.13 (td, ^2^*J*_16ax,16eq_ = 13.1 Hz, ^3^*J*_16ax,15eq_ = 3.7 Hz, 1H, 16-H_ax_), 2.18 (dd, *J*_AB_ = 12.9 Hz, *J*_AX_ = 6.7 Hz, 1H, N*CH*_2_ (a)), 2.32 (dd, *J*_BA_ = 12.9 Hz, *J*_BX_ = 5.9 Hz, N*CH*_2_ (b)), 2.54 (dd, ^2^*J*_16eq,16ax_ = 12.0 Hz, ^3^*J*_16eq,15ax_ = 5.0 Hz, 1H, 16-H_eq_), 2.75 (ddd, ^2^*J*_8β,8α_ = 13.5 Hz, ^3^*J*_8β,7β_ = 12.0 Hz, ^4^*J*_8β,19syn_ = 3.7 Hz, 1H, 8β-H), 2.85 (d, ^2^*J*_10β,10α_ = 18.3 Hz, 1H, 10β-H), 2.89 (d, ^3^*J*_9α,10α_ = 6.4 Hz, 1H, 9α-H), 3.59–3.80 (m, 4H, 6-*O*-*CH*_2_CH_2_CH_2_OTBDPS and 6-*O*-CH_2_CH_2_*CH*_2_OTBDPS), 4.04 (d, ^4^*J*_5β,18anti_ = 1.3 Hz, 1H, 5β-H), 5.14 (br s, 1H, 20-*OH*), 6.19 (d, *J*_1,2_ = 8.4 Hz, 1H, 1-H), 6.48 (d, *J*_2,1_ = 8.4 Hz, 1H, 2-H), 7.16–7.18 (m, 9H, Tr(*m*,*p*)), 7.33–7.66 (m, 16H, Tr(*o*) and (CH_3_)_3_CSi(*Ph*)_2_). ^13^C-NMR (CDCl_3_): δ = 3.3 (cProp(a)), 4.1 (cProp(b)), 9.4 (cProp*CH*), 17.5 (C-19), 19.2 ((CH_3_)_3_*C*Si(Ph)_2_), 22.5 (C-10), 24.8 (20*CH*_3_), 26.9 ((*CH*_3_)_3_CSi(Ph)_2_), 29.7 and 29.8 (C-18 and 20*CH*_3_), 32.1 (C-8), 36.9 (*O*CH_2_*CH*_2_CH_2_*O*), 35.4 (C-15), 35.7 (C-14), 43.6 (C-16), 46.7 (C-13), 48.1 (C-7), 58.0 (C-9), 59.8 (N*CH*_2_), 60.7 and 61.2 (6-*O*-CH_2_CH_2_*CH*_2_OTBDPS and 6-*O*-*CH*_2_CH_2_CH_2_OTBDPS), 74.1 (C-20), 80.3 (C-6), 91.5 (Tr*CO*), 96.4 (C-5), 117.8 (C-2), 123.1 (C-1), 127.2 (*p*CTr), 127.3 (*m*CTr), 127.6 (TBDPS-*Ph-3,5*), 129.4 (*o*CTr), 129.5 and 129.6 (TBDPS-*Ph-C4*), 130.7 (C-11), 132.0 (C-12), 133.9 and 134.0 (TBDPS-*Ph-C1*), 135.5 and 135.6 (TBDPS-*Ph-C2,6*), 137.2 (C-3), 144.2 (TrC-1), 151.4 (C-4). C_63_H_71_NO_5_Si (950.33).

*(5R,6R,7R,9R,13S,14S)-(5α,7α)-17-Cyclopropylmethyl-4,5-epoxy-6-O-(4-tert-butyldiphenylsilyloxybutyl))-18,19-dihydro-α,α-dimethyl-3-triphenylmethoxy-6,14-ethenomorphinan-7-methanol* (6-*O*-(4-*tert*-butyldiphenylsilyloxybutyl)-6-*O*-desmethyl-3-*O*-trityl-diprenorphine, **24c**, TBDPS-OB-TDDPN).



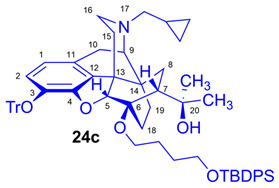



Compound **24c** was synthesized from TDDPN (**23**; 400 mg, 0.61 mmol) using the general procedure in Section 3.2.1. CC: Kieselgel: 60 g; eluent: hexane–ethyl acetate 7:3 (*v*/*v*); fractions: 25 mL. Yield: 419 mg (71%), mp. 77–88 °C, *R_f_*_A_ = 0.81, *R_f_*_B_ = 0.33, *R_f_*_C_ = 0.64. ^1^H-NMR (CDCl_3_): δ = 0.06 (m, 2H, cProp*CH*_2*syn*_), 0.43 (m, 1H, 19-H_syn_), 0.46 (m, 2H, cProp*CH*_2*anti*_), 0.76 (m, 1H, cProp*CH*), 0.89 (td, ^2^*J*_19anti,19syn_ = 13.8 Hz, ^3^*J*_19anti,18syn_ = 5.7 Hz, 1H, 19-H_anti_), 0.97 (dd, ^2^*J*_8α,8β_ = 13.5 Hz, ^3^*J*_8α7β_ = 9.6 Hz, 1H, 8α-H), 1.05 (s, 9H, (*CH*_3_)_3_CSi(Ph)_2_), 1.15 (s, 3H, 20*CH*_3_), 1.34 (s, 3H, 20*CH*_3_), 1.37 (m, 1H, 18-H_syn_), 1.42 (dd, ^2^*J*_15eq,15ax_ = 12.5 Hz, ^3^*J*_15eq,16ax_ = 2.5 Hz, 1H, 15-H_eq_), 1.73–1.78 (m, 5H, 6-*O*-CH_2_*CH*_2_*CH*_2_CH_2_OTBDPS and 18-H_anti_), 1.82 (app t, ^3^*J*_7β,8α_ = 9.9 Hz, 1H, 7β-H), 1.89 (td, ^2^*J*_15ax,15eq_ = 13.0 Hz, ^3^*J*_15ax,16eq_ = 5.5 Hz, 1H, 15-H_ax_), 2.05 (dd, ^2^*J*_10α,10β_ = 18.4 Hz, ^3^*J*_10α,9α_ = 6.0 Hz, 1H, 10α-H), 2.15 (td, ^2^*J*_16ax,16eq_ = 13.0 Hz, ^3^*J*_16ax,15eq_ = 3.6 Hz, 1H, 16-H_ax_), 2.18 (dd, *J*_AB_ = 12.9 Hz, *J*_AX_ = 6.7 Hz, 1H, N*CH*_2_ (a)), 2.33 (dd, *J*_BA_ = 12.9 Hz, *J*_BX_ = 6.0 Hz, N*CH*_2_ (b)), 2.55 (dd, ^2^*J*_16eq,16ax_ = 12.0 Hz, ^3^*J*_16eq,15ax_ = 5.1 Hz, 1H, 16-H_eq_), 2.76 (ddd, ^2^*J*_8β,8α_ = 13.3 Hz, ^3^*J*_8β,7β_ = 12.1 Hz, ^4^*J*_8β,19syn_ = 3.9 Hz, 1H, 8β-H), 2.85 (d, ^2^*J*_10β,10α_ = 18.4 Hz, 1H, 10β-H), 2.90 (d, ^3^*J*_9α,10α_ = 6.0 Hz, 1H, 9α-H), 3.44–3.82 (m, 4H, 6-*O*-*CH*_2_CH_2_CH_2_*CH*_2_OTBDPS), 4.09 (d, ^4^*J*_5β,18anti_ = 1.7 Hz, 1H, 5β-H), 5.26 (br s, 1H, 20-*OH*), 6.19 (d, *J*_1,2_ = 8.0 Hz, 1H, 1-H), 6.48 (d, *J*_2,1_ = 8.0 Hz, 1H, 2-H), 7.16–7.22 (m, 9H, Tr(*m*,*p*)), 7.35–7.67 (m, 16H, Tr(*o*) and (CH_3_)_3_CSi(*Ph*)_2_). ^13^C-NMR (CDCl_3_): δ = 3.2 (cProp (a)), 4.1 (cProp(b)), 9.3 (cPropCH), 17.6 (C-19), 19.2 ((CH_3_)_3_*C*Si(Ph)_2_), 22.5 (C-10), 24.8 (20-CH_3_), 26.9 ((*CH*_3_)CSi(Ph)_2_), 27.2 and 29.3 (6-*O*-CH_2_*CH*_2_*CH*_2_CH_2_OTBDPS), 29.7 and 29.8 (C-18 and 20*CH*_3_), 32.1 (C-8), 35.4 (C-15), 35.7 (C-14), 43.6 (C-16), 46.7 (C-13), 48.0 (C-7), 58.0 (C-9), 59.8 (N*CH*_2_), 63.6 and 64.9 (6-*O*-(CH_2_)_3_*CH*_2_OTBDPS and 6-*O*-*CH*_2_(CH_2_)_3_OTBDPS), 74.1 (C-20), 80.2 (C-6), 91.4 (Tr*CO*), 96.6 (C-5), 117.8 (C-2), 122.9 (C-1), 127.2 (*p*CTr), 127.3 (*m*CTr), 127.6 (TBDPS-*Ph-3,5*), 129.4 (*o*CTr), 129.5 (TBDPS-Ph-C4), 130.7 (C-11), 132.0 (C-12), 134.0 and 134.1 (TBDPS-Ph-C1), 135.5 (TBDPS-Ph-C2,6), 137.2 (C-3), 144.2 (Tr-C1), 151.4 (C-4). C_64_H_73_NO_5_Si (964.35).

*(5R,6R,7R,9R,13S,14S)-(5α,7α)-17-Cyclopropylmethyl-4,5-epoxy-6-O-(5-tert-butyldiphenylsilyloxypentyl))-18,19-dihydro-α,α-dimethyl-3-triphenylmethoxy-6,14-ethenomorphinan-7-methanol* (6-*O*-(5-*tert*-butyldiphenylsilyloxypentyl)-6-*O*-desmethyl-3-*O*-trityl-diprenorphine, **24d**, TBDPS-OPe-TDDPN).



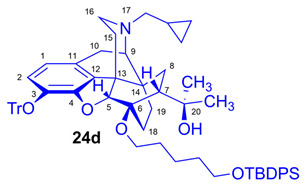



Compound **24d** was synthesized from TDDPN (**23**; 625 mg, 0.95 mmol) using the general procedure in Section 3.2.1. CC: Kieselgel: 70 g; eluent: hexane–ethyl acetate 7:3 (*v*/*v*); fractions: 50 mL. Yield: 573 mg (61%), mp. 70–81 °C, *R_f_*_A_ = 0.90, *R_f_*_B_ = 0.24, *R_f_*_C_ = 0.70. ^1^H-NMR (CDCl_3_): δ = 0.06 (m, 2H, cProp*CH*_2*syn*_), 0.42 (m, 1H, 19-H_syn_), 0.47 (m, 2H, cProp*CH*_2*anti*_), 0.76 (m, 1H, cProp*CH*), 0.89 (td, ^2^*J*_19anti,19syn_ = 13.3 Hz, ^3^*J*_19anti,18syn_ = 5.5 Hz, 1H, 19-H_anti_), 0.97 (dd, ^2^*J*_8α,8β_ = 13.8 Hz, ^3^*J*_8α7β_ = 8.6 Hz, 1H, 8α-H), 1.05 (s, 9H, (*CH*_3_)_3_CSi(Ph)_2_), 1.16 (s, 3H, 20*CH*_3_), 1.24–1.32 (m, 2H, (CH_2_)_2_*CH*_2_(CH_2_)_2_), 1.34 (s, 3H, 20*CH*_3_), 1.37–1.58 (m, 7H, 18-H_syn_, 15-H_eq_, 18-H_anti_, 6-*O*-CH_2_*CH*_2_CH_2_*CH*_2_CH_2_OTBDPS), 1.81 (app t, ^3^*J*_7β,8α_ = 9.7 Hz, 1H, 7β-H), 1.89 (td, ^2^*J*_15ax,15eq_ = 13.5 Hz, ^3^*J*_15ax,16eq_ = 5.4 Hz, 1H, 15-H_ax_), 2.05 (dd, ^2^*J*_10α,10β_ = 18.7 Hz, ^3^*J*_10α,9α_ = 6.3 Hz, 1H, 10α-H), 2.15 (td, ^2^*J*_16ax,16eq_ = 12.8 Hz, ^3^*J*_16ax,15eq_ = 3.6 Hz, 1H, 16-H_ax_), 2.18 (dd, *J*_AB_ = 12.8 Hz, *J*_AX_ = 6.7 Hz, 1H, N*CH*_2_ (a)), 2.33 (dd, *J*_BA_ = 12.8 Hz, *J*_BX_ = 5.9 Hz, N*CH*_2_ (b)), 2.55 (dd, ^2^*J*_16eq,16ax_ = 11.9 Hz, ^3^*J*_16eq,15ax_ = 4.9 Hz, 1H, 16-H_eq_), 2.76 (ddd, ^2^*J*_8β,8α_ = 13.5 Hz, ^3^*J*_8β,7β_ = 12.3 Hz, ^4^*J*_8β,19syn_ = 3.8 Hz, 1H, 8β-H), 2.85 (d, ^2^*J*_10β,10α_ = 18.7 Hz, 1H, 10β-H), 2.90 (d, ^3^*J*_9α,10α_ = 6.3 Hz, 1H, 9α-H), 3.39–3.79 (m, 4H, 6-*O*-*CH*_2_(CH_2_)_3_*CH*_2_OTBDPS), 4.10 (d, ^4^*J*_5β,18anti_ = 1.7 Hz, 1H, 5β-H), 5.29 (br s, 1H, 20-*OH*), 6.19 (d, *J*_1,2_ = 8.4 Hz, 1H, 1-H), 6.47 (d, *J*_2,1_ = 8.4 Hz, 1H, 2-H), 7.15–7.21 (m, 9H, Tr(*m*,*p*)), 7.36–7.68 (m, 16H, Tr(*o*) and (CH_3_)_3_CSi(*Ph*)_2_). ^13^C-NMR (CDCl_3_): δ = 3.2 (cProp (a)), 4.1 (cProp(b)), 9.3 (cPropCH), 17.6 (C-19), 19.2 ((CH_3_)_3_*C*Si(Ph)_2_), 22.3 ((CH_2_)_2_*CH*_2_(CH_2_)_2_), 22.5 (C-10), 24.8 (20-CH_3_), 26.9 ((*CH*_3_)CSi(Ph)_2_), 29.7 and 29.8 (C-18 and 20*CH*_3_), 30.5 and 32.4 (6-*O*-CH_2_*CH*_2_CH_2_*CH*_2_CH_2_OTBDPS), 32.1 (C-8), 35.4 (C-15), 35.7 (C-14), 43.6 (C-16), 46.7 (C-13), 48.0 (C-7), 58.0 (C-9), 59.8 (N*CH*_2_), 63.9 and 64.8 (6-*O*-(CH_2_)_4_*CH*_2_OTBDPS and 6-*O*-*CH*_2_(CH_2_)_4_OTBDPS), 74.2 (C-20), 80.1 (C-6), 91.4 (Tr*CO*), 96.6 (C-5), 117.8 (C-2), 122.9 (C-1), 127.2 (*p*CTr), 127.3 (*m*CTr), 127.6 (TBDPS-*Ph-3,5*), 129.4 (*o*CTr), 129.5 (TBDPS-Ph-C4), 130.7 (C-11), 132.1 (C-12), 134.1 (TBDPS-Ph-C1), 135.5 (TBDPS-Ph-C2,6), 137.2 (C-3), 144.2 (Tr-C1), 151.3 (C-4). C_65_H_75_NO_5_Si (978.38).

#### 3.2.2. General Procedure for the Synthesis of 6-*O*-Hydroxyalkyl-6-*O*-desmethyl-3-*O*-trityl-diprenorphine (**25b**–**d**) Derivatives

The corresponding TBDPS-protected TDDPN derivatives (**24b**–**d;** 0.42 mmol) were dissolved in tetrahydrofuran (15 mL) under an argon atmosphere. A 1M solution of tetrabutylammonium fluoride in tetrahydrofuran (0.54 mL, 0.54 mmol, 1.28 equiv.) was added, and the mixture was stirred for 4 h. The solvent was removed under reduced pressure. Water (50 mL) was added to the residue, and the mixture was extracted with dichloromethane (3 × 40 mL). The combined organic extracts were dried over anhydrous Na_2_SO_4_, and the solution was concentrated under reduced pressure. The crude product was purified by column chromatography on silica gel.

*(5R,6R,7R,9R,13S,14S)-(5α,7α)-17-Cyclopropylmethyl-4,5-epoxy-6-O-(3-hydroxypropyl)-18,19-dihydro-α,α-dimethyl-3-triphenylmethoxy-6,14-ethenomorphinan-7-methanol* (6-*O*-(3-hydroxypropyl)-6-*O*-desmethyl-3-*O*-trityl-diprenorphine, **25b**, HP-TDDPN).



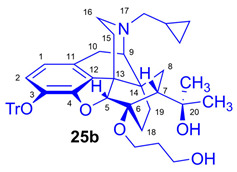



Compound **25b** was synthesized from TBDPS-OP-TDDPN (**24b**; 400 mg, 0.42 mmol) using the general procedure in Section 3.2.2. CC: Kieselgel: 40 g; eluent: dichloromethane–methanol 95:5 (*v*/*v*); fractions: 25 mL. Yield: 254 mg (84%), mp. 110–121 °C, *R_f_*_C_ = 0.09, *R_f_*_D_ = 0.77, *R_f_*_E_ = 0.23. ^1^H-NMR (CDCl_3_): δ = 0.06 (m, 2H, cProp*CH*_2*syn*_), 0.42 (tt, 1H, 19-H_syn_), 0.46 (m, 2H, cProp*CH*_2*anti*_), 0.76 (m, 1H, cProp*CH*), 0.90 (td, ^2^*J*_19anti,19syn_ = 13.4 Hz, ^3^*J*_19anti,18syn_ = 5.7 Hz, 1H, 19-H_anti_), 0.98 (dd, ^2^*J*_8α,8β_ = 13.4 Hz, ^3^*J*_8α,7β_ = 9.3 Hz, 1H, 8α-H), 1.16 (s, 3H, 20*CH*_3_), 1.35 (s, 3H, 20*CH*_3_), 1.38–1.47 (m, 2H, 18-H_syn_, 15-H_eq_), 1.61 (tt, 1H, 18-H_anti_), 1.72–1.75 (m, 2H, CH_2_*CH*_2_CH_2_OH), 1.83 (app t, ^3^*J*_7β,8α_ = 10.1 Hz, 1H, 7β-H), 1.90 (td, ^2^*J*_15ax,15eq_ = 13.5 Hz, ^3^*J*_15ax,16eq_ = 5.5 Hz, 1H, 15-H_ax_), 2.05 (dd, ^2^*J*_10α,10β_ = 18.4 Hz, ^3^*J*_10α,9α_ = 6.4 Hz, 1H, 10α-H), 2.15 (td, ^2^*J*_16ax,16eq_ = 13.4 Hz, ^3^*J*_16ax,15eq_ = 3.9 Hz, 1H, 16-H_ax_), 2.18 (dd, *J*_AB_ = 12.8 Hz, *J*_AX_ = 6.7 Hz, 1H, N*CH*_2_ (a)), 2.32 (dd, *J*_BA_ = 12.8 Hz, *J*_BX_ = 5.7 Hz, N*CH*_2_ (b)), 2.55 (dd, ^2^*J*_16eq,16ax_ = 12.1 Hz, ^3^*J*_16eq,15ax_ = 4.6 Hz, 1H, 16-H_eq_), 2.77 (ddd, ^2^*J*_8β,8α_ = 13.2 Hz, ^3^*J*_8β,7β_ = 12.0 Hz, ^4^*J*_8β,19syn_ = 3.7 Hz, 1H, 8β-H), 2.85 (d, ^2^*J*_10β,10α_ = 18.4 Hz, 1H, 10β-H), 2.90 (d, ^3^*J*_9α,10α_ = 6.4 Hz, 1H, 9α-H), 3.57–3.92 (m, 4H, 6-*O*-CH_2_CH_2_*CH*_2_OH, 6-*O*-*CH*_2_CH_2_CH_2_OH), 4.14 (d, ^4^*J*_5β,18anti_ = 1.7 Hz, 1H, 5β-H), 5.12 (br s, 1H, 20-*OH*), 6.18 (d, *J*_1,2_ = 8.4 Hz, 1H, 1-H), 6.47 (d, *J*_2,1_ = 8.4 Hz, 1H, 2-H), 7.20–7.26 (m, 9H, Tr(*m*,*p*)), 7.42–7.45 (m, 4H, Tr(*o*)). ^13^C-NMR (CDCl_3_): δ = 3.2 (cProp (a)), 4.1 (cProp (b)), 9.3 (cProp*CH*), 17.7 (C-19), 22.5 (C-10), 24.9 (20*CH*_3_), 29.8 (20*CH*_3_, C-18), 32.2 (C-8), 33.4 (6-*O-*CH_2_*CH*_2_CH_2_*O*), 35.4 (C-15), 35.7 (C-14), 43.6 (C-16), 46.7 (C-13), 48.0 (C-7), 58.0 (C-9), 59.8 (N*CH*_2_), 60.1 and 62.0 (6-*O*-CH_2_CH_2_*CH*_2_OH and 6-*O*-*CH*_2_CH_2_CH_2_OH), 74.2 (C-20), 80.4 (C-6), 91.3 (Tr*CO*), 96.7 (C-5), 117.9 (C-2), 122.8 (C-1), 127.2 (*p*CTr), 127.3 (*m*CTr), 129.3 (*o*CTr), 130.6 (C-11), 132.0 (C-12), 137.3 (C-3), 144.2 (TrC-1), 151.1 (C-4). C_47_H_53_NO_5_ (711.93).

*(5R,6R,7R,9R,13S,14S)-(5α,7α)-17-Cyclopropylmethyl-4,5-epoxy-6-O-(4-hydroxybutyl))-18,19-dihydro-α,α-dimethyl-3-triphenylmethoxy-6,14-ethenomorphinan-7-methanol* (6-*O*-(4-hydroxybutyl)-6-*O*-desmethyl-3-*O*-trityl-diprenorphine, **25c**, HB-TDDPN).



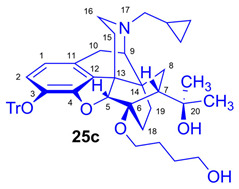



Compound **25c** was synthesized from TBDPS-OB-TDDPN (**24c**; 486 mg, 0.50 mmol) using general procedure in Section 3.2.2. CC: Kieselgel: 50 g; eluent: dichloromethane–methanol 95:5 (*v*/*v*); fractions: 25 mL. Yield: 340 mg (93%), mp. 105–116 °C, *R_f_*_C_ = 0.09, *R_f_*_D_ = 0.85, *R_f_*_E_ = 0.25. ^1^H-NMR (CDCl_3_): δ = 0.05 (m, 2H, cProp*CH*_2*syn*_), 0.43 (m, 1H, 19-H_syn_), 0.46 (m, 2H, cProp*CH*_2*anti*_), 0.76 (m, 1H, cProp*CH*), 0.90 (td, ^2^*J*_19anti,19syn_ = 13.8 Hz, ^3^*J*_19anti,18syn_ = 5.4 Hz, 1H, 19-H_anti_), 0.98 (dd, ^2^*J*_8α,8β_ = 13.8 Hz, ^3^*J*_8α,7β_ = 9.9 Hz, 1H, 8α-H), 1.16 (s, 3H, 20*CH*_3_), 1.35 (s, 3H, 20*CH*_3_), 1.37–1.46 (m, 2H, 15-H_eq_, 18-H_syn_), 1.51–1.60 (m, 5H, 6-*O*-CH_2_*CH*_2_*CH*_2_CH_2_OH and 18-H_anti_), 1.83 (app t, ^3^*J*_7β,8α_ = 10.0 Hz, 1H, 7β-H), 1.90 (td, ^2^*J*_15ax,15eq_ = 13.7 Hz, ^3^*J*_15ax,16eq_ = 5.5 Hz, 1H, 15-H_ax_), 2.05 (dd, ^2^*J*_10α,10β_ = 18.7 Hz, ^3^*J*_10α,9α_ = 6.4 Hz, 1H, 10α-H), 2.12–2.20 (m, 2H, 16-H_ax_, N*CH*_2_ (a)), 2.32 (dd, *J*_BA_ = 12.5 Hz, *J*_BX_ = 5.7 Hz, N*CH*_2_ (b)), 2.55 (dd, ^2^*J*_16eq,16ax_ = 12.0 Hz, ^3^*J*_16eq,15ax_ = 4.7 Hz, 1H, 16-H_eq_), 2.77 (ddd, ^2^*J*_8β,8α_ = 13.5 Hz, ^3^*J*_8β,7β_ = 12.8 Hz, ^4^*J*_8β,19syn_ = 2.7 Hz, 1H, 8β-H), 2.85 (d, ^2^*J*_10β,10α_ = 18.7 Hz, 1H, 10β-H), 2.90 (d, ^3^*J*_9α,10α_ = 6.4 Hz, 1H, 9α-H), 3.46–3.84 (m, 4H, 6-*O*-*CH*_2_CH_2_CH_2_*CH*_2_OH), 4.12 (d, ^4^*J*_5β,18anti_ = 1.8 Hz, 1H, 5β-H), 5.28 (br s, 1H, 20-*OH*), 6.18 (d, *J*_1,2_ = 8.4 Hz, 1H, 1-H), 6.47 (d, *J*_2,1_ = 8.4 Hz, 1H, 2-H), 7.22–7.26 (m, 9H, Tr(*m*,*p*)), 7.43–7.44 (m, 6H, Tr(*o*)). ^13^C-NMR (CDCl_3_): δ = 3.2 (cProp (a)), 4.1 (cProp(b)), 9.3 (cPropCH), 17.7 (C-19), 22.5 (C-10), 24.8 (20-CH_3_), 26.9 and 29.3 (6-*O*-CH_2_*CH*_2_*CH*_2_CH_2_OH), 29.7 (C-18 and 20*CH*_3_), 32.1 (C-8), 35.4 (C-15), 35.7 (C-14), 43.6 (C-16), 46.7 (C-13), 48.0 (C-7), 58.0 (C-9), 59.8 (N*CH*_2_), 62.4 and 64.5 (6-*O*-(CH_2_)_3_*CH*_2_OH and 6-*O*-*CH*_2_(CH_2_)_3_OH), 74.3 (C-20), 80.3 (C-6), 91.4 (Tr*CO*), 96.6 (C-5), 117.8 (C-2), 122.8 (C-1), 127.2 (*p*CTr), 127.3 (*m*CTr), 129.3 (*o*CTr), 130.6 (C-11), 132.0 (C-12), 137.2 (C-3), 144.2 (Tr-C1), 151.2 (C-4). C_48_H_55_NO_5_ (725.95).

*(5R,6R,7R,9R,13S,14S)-(5α,7α)-17-Cyclopropylmethyl-4,5-epoxy-6-O-(5-hydroxypentyl))-18,19-dihydro-α,α-dimethyl-3-triphenylmethoxy-6,14-ethenomorphinan-7-methanol* (6-*O*-(5-hydroxypentyl)-6-*O*-desmethyl-3-*O*-trityl-diprenorphine, **25d**, HPe-TDDPN).



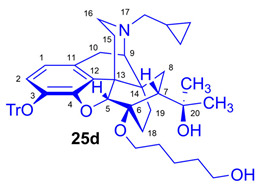



Compound **25d** was synthesized from TBDPSOPe-TDDPN (**24d**; 562 mg, 0.57 mmol) using the general procedure in Section 3.2.2. CC: Kieselgel: 70 g; eluent: dichloromethane–methanol 95:5 (*v*/*v*); fractions: 25 mL. Yield: 385 mg (90%), mp. 101–109 °C, *R_f_*_A_ = 0.15, *R_f_*_D_ = 0.49, *R_f_*_E_ = 0.35. ^1^H-NMR (CDCl_3_): δ = 0.06 (m, 2H, cProp*CH*_2*syn*_), 0.43 (m, 1H, 19-H_syn_), 0.46 (m, 2H, cProp*CH*_2*anti*_), 0.76 (m, 1H, cProp*CH*), 0.89 (td, ^2^*J*_19anti,19syn_ = 13.2 Hz, ^3^*J*_19anti,18syn_ = 5.9 Hz, 1H, 19-H_anti_), 0.98 (dd, ^2^*J*_8α,8β_ = 13.4 Hz, ^3^*J*_8α7β_ = 9.6 Hz, 1H, 8α-H), 1.16 (s, 3H, 20*CH*_3_), 1.32–1.38 (m, 2H, (CH_2_)_2_*CH*_2_(CH_2_)_2_), 1.35 (s, 3H, 20*CH*_3_), 1.39–1.61 (m, 7H, 18-H_syn_, 15-H_eq_, 18-H_anti_, 6-*O*-CH_2_*CH*_2_CH_2_*CH*_2_CH_2_OTBDPS), 1.82 (app t, ^3^*J*_7β,8α_ = 10.0 Hz, 1H, 7β-H), 1.89 (td, ^2^*J*_15ax,15eq_ = 13.7 Hz, ^3^*J*_15ax,16eq_ = 5.4 Hz, 1H, 15-H_ax_), 2.05 (dd, ^2^*J*_10α,10β_ = 18.4 Hz, ^3^*J*_10α,9α_ = 6.3 Hz, 1H, 10α-H), 2.15 (td, ^2^*J*_16ax,16eq_ = 12.7 Hz, ^3^*J*_16ax,15eq_ = 3.7 Hz, 1H, 16-H_ax_), 2.18 (dd, *J*_AB_ = 12.7 Hz, *J*_AX_ = 6.7 Hz, 1H, N*CH*_2_ (a)), 2.32 (dd, *J*_BA_ = 12.7 Hz, *J*_BX_ = 6.0 Hz, N*CH*_2_ (b)), 2.55 (dd, ^2^*J*_16eq,16ax_ = 11.9 Hz, ^3^*J*_16eq,15ax_ = 5.0 Hz, 1H, 16-H_eq_), 2.76 (ddd, ^2^*J*_8β,8α_ = 13.4 Hz, ^3^*J*_8β,7β_ = 12.2 Hz, ^4^*J*_8β,19syn_ = 3.8 Hz, 1H, 8β-H), 2.85 (d, ^2^*J*_10β,10α_ = 18.4 Hz, 1H, 10β-H), 2.90 (d, ^3^*J*_9α,10α_ = 6.3 Hz, 1H, 9α-H), 3.42–3.83 (m, 4H, 6-*O*-*CH*_2_(CH_2_)_3_*CH*_2_OTBDPS), 4.11 (d, ^4^*J*_5β,18anti_ = 1.7 Hz, 1H, 5β-H), 5.36 (br s, 1H, 20-*OH*), 6.18 (d, *J*_1,2_ = 8.0 Hz, 1H, 1-H), 6.47 (d, *J*_2,1_ = 8.0 Hz, 1H, 2-H), 7.20–7.26 (m, 9H, Tr(*m*,*p*)), 7.42–7.44 (m, 6H, Tr(*o*)). ^13^C-NMR (CDCl_3_): δ = 3.2 (cProp (a)), 4.1 (cProp(b)), 9.3 (cPropCH), 17.6 (C-19), 22.2 ((CH_2_)_2_*CH*_2_(CH_2_)_2_), 22.5 (C-10), 24.8 (20-CH_3_), 29.7 and 29.8 (C-18 and 20*CH*_3_), 30.3 and 32.4 (6-*O*-CH_2_*CH*_2_CH_2_*CH*_2_CH_2_OH), 32.1 (C-8), 35.4 (C-15), 35.7 (C-14), 43.6 (C-16), 46.7 (C-13), 48.0 (C-7), 58.0 (C-9), 59.8 (N*CH*_2_), 62.6 and 64.6 (6-*O*-(CH_2_)_4_*CH*_2_OH and 6-*O*-*CH*_2_(CH_2_)_4_OH), 74.3 (C-20), 80.2 (C-6), 91.3 (Tr*CO*), 96.7 (C-5), 117.8 (C-2), 122.9 (C-1), 127.2 (*p*CTr), 127.3 (*m*CTr), 129.4 (*o*CTr), 130.7 (C-11), 132.0 (C-12), 137.2 (C-3), 144.2 (Tr-C1), 151.3 (C-4). C_49_H_57_NO_5_ (739.98).

#### 3.2.3. General Procedure for the Synthesis of 6-*O*-(Tosyloxyalkyl)-6-*O*-desmethyl-diprenorphine (**26b**–**d**) Derivatives

The corresponding 6-*O*-(hydroxyalkyl)-6-*O*-desmethyl-3-*O*-trityl-diprenorphine (**25b**–**d**; 0.67 mmol) derivative was dissolved in dry dichloromethane (10 mL) under argon. The solution was cooled to 0 °C and pyridine (220 μL, 215 mg, 2.72 mmol, 4 equiv.) was added. After stirring for 15 min, toluenesulfonic anhydride (750 mg, 2.29 mmol, 3.4 equiv.) was added in small portions. After stirring at RT for 4 h, the reaction mixture was poured into water (50 mL). The resulting suspension was extracted with dichloromethane (3 × 60 mL). The combined organic layer was dried with anhydrous Na_2_SO_4_ and the solvent removed under reduced pressure. The resulting crude product mixture was separated by column chromatography on silica gel.

*(5R,6R,7R,9R,13S,14S)-(5α,7α)-17-Cyclopropylmethyl-4,5-epoxy-6-O-(3-(4-toluene sulfonyloxy)propyl)-18,19-dihydro-α,α-dimethyl-3-triphenylmethoxy-6,14-ethenomorphinan-7-methanol* (6-*O*-(3-tosyloxypropyl)-6-*O*-desmethyl-3-*O*-trityl-diprenorphine, **26b**, TP-TDDPN).



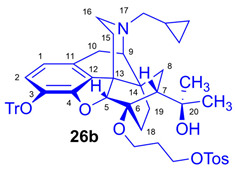



Compound **26b** was prepared from HP-TDDPN (**25b**; 480 mg, 0.67 mmol) using the general procedure in Section 3.2.3. CC: Kieselgel: 70 g; eluent: dichloromethane–methanol 95:5 (*v*/*v*); fractions: 25 mL. Yield: 349 mg (60%), mp. 87–102 °C, *R_f_*_C_ = 0.28, *R_f_*_D_ = 0.85, *R_f_*_E_ = 0.61. ^1^H-NMR (CDCl_3_): δ = 0.05 (m, 2H, cProp*CH*_2*syn*_), 0.43 (m, 1H, 19-H_syn_), 0.46 (m, 2H, cProp*CH*_2*anti*_), 0.75 (m, 1H, cProp*CH*), 0.88 (td, ^2^*J*_19anti,19syn_ = 12.4 Hz, ^3^*J*_19anti,18syn_ = 5.7 Hz, 1H, 19-H_anti_), 0.95 (dd, ^2^*J*_8α,8β_ = 13.3 Hz, ^3^*J*_8α7β_ = 9.7 Hz, 1H, 8α-H), 1.13 (s, 3H, 20*CH*_3_), 1.26 (s, 3H, 20*CH*_3_), 1.30–1.41 (m, 2H, 18-H_syn_, 15-H_eq_), 1.49 (m, 1H, 18-H_anti_), 1.75 (app t, ^3^*J*_7β,8α_ = 10.1 Hz, 1H, 7β-H), 1.79–1.84 (m, 2H, CH_2_*CH*_2_CH_2_OH), 1.86 (td, ^2^*J*_15ax,15eq_ = 13.2 Hz, ^3^*J*_15ax,16eq_ = 5.4 Hz, 1H, 15-H_ax_), 2.05 (dd, ^2^*J*_10α,10β_ = 18.7 Hz, ^3^*J*_10α,9α_ = 6.1 Hz, 1H, 10α-H), 2.14 (td, ^2^*J*_16ax,16eq_ = 13.0 Hz, ^3^*J*_16ax,15eq_ = 3.6 Hz, 1H, 16-H_ax_), 2.18 (dd, *J*_AB_ = 12.9 Hz, *J*_AX_ = 6.7 Hz, 1H, N*CH*_2_ (a)), 2.32 (dd, *J*_BA_ = 12.9 Hz, *J*_BX_ = 5.6 Hz, N*CH*_2_ (b)), 2.41 (s, 3H, Tos-CH_3_), 2.54 (dd, ^2^*J*_16eq,16ax_ = 11.9 Hz, ^3^*J*_16eq,15ax_ = 5.0 Hz, 1H, 16-H_eq_), 2.74 (ddd, ^2^*J*_8β,8α_ = 13.4 Hz, ^3^*J*_8β,7β_ = 12.4 Hz, ^4^*J*_8β,19syn_ = 3.9 Hz, 1H, 8β-H), 2.85 (d, ^2^*J*_10β,10α_ = 18.7 Hz, 1H, 10β-H), 2.89 (d, ^3^*J*_9α,10α_ = 6.1 Hz, 1H, 9α-H), 3.49–4.12 (m, 4H, 6-*O*-CH_2_CH_2_*CH*_2_OH, 6-*O*-*CH*_2_CH_2_CH_2_OH), 3.99 (d, ^4^*J*_5β,18anti_ = 2.3 Hz, 1H, 5β-H), 4.83 (br s, 1H, 20-*OH*), 6.20 (d, *J*_1,2_ = 8.4 Hz, 1H, 1-H), 6.49 (d, *J*_2,1_ = 8.4 Hz, 1H, 2-H), 7.21–7.24 (m, 9H, Tr(*m*,*p*)), 7.31 (d, *J* = 8.1 Hz, 2H, Tos-3,5), 7.39–7.42 (m, 4H, Tr(*o*)), 7.78 (d, *J* = 8.1 Hz, 2H, Tos-2,6). ^13^C-NMR (CDCl_3_): δ = 3.2 (cProp (a)), 4.1 (cProp (b)), 9.3 (cProp*CH*), 17.4 (C-18), 21.6 (Tos-CH_3_), 22.5 (C-10), 24.8 (20*CH*_3_), 29.7 (20*CH*_3_), 30.3 (C-19), 32.1 (C-8), 33.4 (6-*O-*CH_2_*CH*_2_CH_2_*O*), 35.4 (C-15), 35.7 (C-14), 43.5 (C-16), 46.8 (C-13), 48.1 (C-7), 57.9 (C-9), 59.8 (N*CH*_2_), 60.4 and 67.4 (6-*O*-CH_2_CH_2_*CH*_2_OH and 6-*O*-*CH*_2_CH_2_CH_2_OH), 74.2 (C-20), 80.8 (C-6), 91.5 (Tr*CO*), 96.2 (C-5), 118.0 (C-2), 123.2 (C-1), 127.3 (*p*CTr), 127.4 (*m*CTr), 127.9 (Tos-2,6), 129.4 (*o*CTr), 129.8 (Tos-3,5), 130.7 (C-11), 131.9 (C-12), 133.1 (Tos-C1), 137.2 (C-3), 144.2 (TrC-1), 144.6 (Tos-C4), 151.2 (C-4). Calculated for C_54_H_59_NO_7_S (865.4012), found: 866.4104 ([M+H]^+^).

*(5R,6R,7R,9R,13S,14S)-(5α,7α)-17-Cyclopropylmethyl-4,5-epoxy-6-O-(4-(4-toluenesulfonyloxy)butyl))-18,19-dihydro-α,α-dimethyl-3-triphenylmethoxy-6,14-ethenomorphinan-7-methanol* (6-*O*-(4-tosyloxybutyl)-6-*O*-desmethyl-3-*O*-trityl-diprenorphine, **26c**, TB-TDDPN).



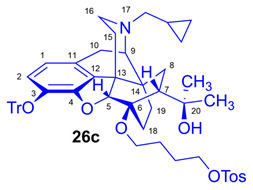



Compound **26c** was synthesized from HB-TDDPN (**25c**; 320 mg, 0.44 mmol) using the general procedure in Section 3.2.3. CC: Kieselgel: 60 g; eluent: dichloromethane–methanol 95:5 (*v*/*v*); fractions: 25 mL. Yield: 337 mg (87%), mp. 80–98 °C, *R_f_*_C_ = 0.38, *R_f_*_D_ = 0.85, *R_f_* _E_ = 0.63. ^1^H-NMR (CDCl_3_): δ = 0.05 (m, 2H, cProp*CH*_2*syn*_), 0.41 (m, 1H, 19-H_syn_), 0.46 (m, 2H, cProp*CH*_2*anti*_), 0.75 (m, 1H, cProp*CH*), 0.88 (td, ^2^*J*_19anti,19syn_ = 13.6 Hz, ^3^*J*_19anti,18syn_ = 5.7 Hz, 1H, 19-H_anti_), 0.96 (dd, ^2^*J*_8α,8β_ = 13.3 Hz, ^3^*J*_8α7β_ = 9.7 Hz, 1H, 8α-H), 1.14 (s, 3H, 20*CH*_3_), 1.31 (s, 3H, 20*CH*_3_), 1.32–1.37 (m, 1H, 18-H_syn_), 1.42 (dd, ^2^*J*_15eq,15ax_ = 13.1 Hz, ^3^*J*_15eq,16ax_ = 2.7 Hz, 1H, 15-H_eq_), 1.40–1.63 (m, 5H, 6-*O*-CH_2_*CH*_2_*CH*_2_CH_2_OTos and 18-H_anti_), 1.78 (app t, ^3^*J*_7β,8α_ = 9.4 Hz, 1H, 7β-H), 1.87 (td, ^2^*J*_15ax,15eq_ = 13.9 Hz, ^3^*J*_15ax,16eq_ = 5.6 Hz, 1H, 15-H_ax_), 2.04 (dd, ^2^*J*_10α,10β_ = 18.8 Hz, ^3^*J*_10α,9α_ = 6.0 Hz, 1H, 10α-H), 2.14 (td, ^2^*J*_16ax,16eq_ = 11.5 Hz, ^3^*J*_16ax,15eq_ = 3.8 Hz, 1H, 16-H_ax_), 2.17 (dd, *J*_AB_ = 12.8 Hz, *J*_AX_ = 6.7 Hz, N*CH*_2_ (a)), 2.32 (dd, *J*_BA_ = 12.8 Hz, *J*_BX_ = 5.7 Hz, N*CH*_2_ (b)), 2.42 (s, 3H, Tos-CH_3_), 2.54 (dd, ^2^*J*_16eq,16ax_ = 12.0 Hz, ^3^*J*_16eq,15ax_ = 5.4 Hz, 1H, 16-H_eq_), 2.75 (ddd, ^2^*J*_8β,8α_ = 13.8 Hz, ^3^*J*_8β,7β_ = 12.4 Hz, ^4^*J*_8β,19syn_ = 3.9 Hz, 1H, 8β-H), 2.84 (d, ^2^*J*_10β,10α_ = 18.8 Hz, 1H, 10β-H), 2.89 (d, ^3^*J*_9α,10α_ = 6.0 Hz, 1H, 9α-H), 3.36–3.99 (m, 4H, 6-*O*-*CH*_2_CH_2_CH_2_*CH*_2_OTos), 4.05 (d, ^4^*J*_5β,18anti_ = 1.7 Hz, 1H, 5β-H), 5.07 (br s, 1H, 20-*OH*), 6.18 (d, *J*_1,2_ = 8.0 Hz, 1H, 1-H), 6.46 (d, *J*_2,1_ = 8.0 Hz, 1H, 2-H), 7.19–7.24 (m, 9H, Tr(*m*,*p*)), 7.33 (d, *J* = 8.1 Hz, 2H, Tos-3,5), 7.38–7.42 (m, 6H, Tr(*o*)), 7.79 (d, *J* = 8.1 Hz, 2H, Tos-2,6). ^13^C-NMR (CDCl_3_): δ = 3.2 (cProp (a)), 4.1 (cProp(b)), 9.3 (cPropCH), 17.6 (C-18), 21.6 (Tos-CH_3_), 22.5 (C-10), 24.9 (20-CH_3_), 25.7 and 26.7 (6-*O*-CH_2_*CH*_2_*CH*_2_CH_2_OTos), 29.7 (C-19 and 20*CH*_3_), 32.1 (C-8), 35.4 (C-15), 35.7 (C-14), 43.6 (C-16), 46.7 (C-13), 48.0 (C-7), 57.9 (C-9), 59.8 (N*CH*_2_), 64.0 and 70.2 (6-*O*-CH_2_CH_2_CH_2_*CH*_2_OTos and 6-*O*-*CH*_2_CH_2_CH_2_CH_2_OTos), 74.2 (C-20), 80.5 (C-6), 91.4 (Tr*CO*), 96.5 (C-5), 117.9 (C-2), 122.9 (C-1), 127.3 (*p*CTr), 127.4 (*m*CTr), 127.9 (Tos-2,6), 129.3 (*o*CTr), 129.8 (Tos-3,5), 130.6 (C-11), 131.9 (C-12), 133.1 (Tos-C1), 137.2 (C-3), 144.2 (TrC-1), 144.6 (Tos-C4), 151.2 (C-4). Calculated for C_55_H_61_NO_7_S (879.4169), found 880.4255 ([M+H]^+^).

(*5R,6R,7R,9R,13S,14S)-(5α,7α)-17-Cyclopropylmethyl-4,5-epoxy-6-O-(5-(4-toluenesulfonyloxy)pentyl))-18,19-dihydro-α,α-dimethyl-3-triphenylmethoxy-6,14-ethenomorphinan-7-methanol* (6-*O*-(5-tosyloxypentyl)-6-*O*-desmethyl-3-*O*-trityl-diprenorphine, **26d**, TPe-TDDPN).



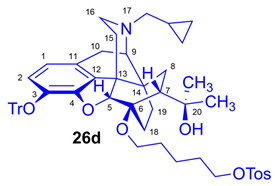



Compound **26d** was synthesized from HPe-TDDPN (**25d**; 380 mg, 0.51 mmol) using the general procedure in Section 3.2.3. CC: Kieselgel: 60 g; eluent: dichloromethane–methanol 98:2 (*v*/*v*); fractions: 25 mL. Yield: 311 mg (67%), mp. 77–88 °C, *R_f_*_C_ = 0.50, *R_f_*_D_ = 0.87, *R_f_*_E_ = 0.70. ^1^H-NMR (CDCl_3_): δ = 0.05 (m, 2H, cProp*CH*_2*syn*_), 0.42 (tt, *J* = 12.7 Hz, *J* = 5.7 Hz, 1H, 19-H_syn_), 0.46 (m, 2H, cProp*CH*_2*anti*_), 0.75 (m, 1H, cProp*CH*), 0.88 (td, ^2^*J*_19anti,19syn_ = 13.3 Hz, ^3^*J*_19anti,18syn_ = 5.9 Hz, 1H, 19-H_anti_), 0.97 (dd, ^2^*J*_8α,8β_ = 13.1 Hz, ^3^*J*_8α7β_ = 9.8 Hz, 1H, 8α-H), 1.14 (s, 3H, 20*CH*_3_), 1.32 (s, 3H, 20*CH*_3_), 1.26–1.62 (m, 9H, 18-H_syn_, 15-H_eq_, 6-*O*-CH_2_*CH*_2_*CH*_2_*CH*_2_CH_2_OTos and 18-H_anti_), 1.80 (app t, ^3^*J*_7β,8α_ = 9.7 Hz, 1H, 7β-H), 1.89 (td, ^2^*J*_15ax,15eq_ = 13.7 Hz, ^3^*J*_15ax,16eq_ = 5.7 Hz, 1H, 15-H_ax_), 2.04 (dd, ^2^*J*_10α,10β_ = 18.4 Hz, ^3^*J*_10α,9α_ = 6.4 Hz, 1H, 10α-H), 2.14 (td, ^2^*J*_16ax,16eq_ = 13.2 Hz, ^3^*J*_16ax,15eq_ = 3.8 Hz, 1H, 16-H_ax_), 2.18 (dd, *J*_AB_ = 12.6 Hz, *J*_AX_ = 6.7 Hz, N*CH*_2_ (a)), 2.32 (dd, *J*_BA_ = 12.6 Hz, *J*_BX_ = 5.7 Hz, N*CH*_2_ (b)), 2.43 (s, 3H, Tos-CH_3_), 2.55 (dd, ^2^*J*_16eq,16ax_ = 12.1 Hz, ^3^*J*_16eq,15ax_ = 4.7 Hz, 1H, 16-H_eq_), 2.76 (ddd, ^2^*J*_8β,8α_ = 13.1 Hz, ^3^*J*_8β,7β_ = 11.9 Hz, ^4^*J*_8β,19syn_ = 3.9 Hz, 1H, 8β-H), 2.85 (d, ^2^*J*_10β,10α_ = 18.4 Hz, 1H, 10β-H), 2.89 (d, ^3^*J*_9α,10α_ = 6.4 Hz, 1H, 9α-H), 3.36–3.98 (m, 4H, 6-*O*-*CH*_2_(CH_2_)_3_*CH*_2_OTos), 4.08 (d, ^4^*J*_5β,18anti_ = 1.7 Hz, 1H, 5β-H), 5.18 (br s, 1H, 20-*OH*), 6.18 (d, *J*_1,2_ = 8.4 Hz, 1H, 1-H), 6.46 (d, *J*_2,1_ = 8.4 Hz, 1H, 2-H), 7.18–7.22 (m, 9H, Tr(*m*,*p*)), 7.33 (d, *J* = 8.4 Hz, 2H, Tos-3,5), 7.39–7.44 (m, 6H, Tr(*o*)), 7.79 (d, *J* = 8.4 Hz, 2H, Tos-2,6). ^13^C-NMR (CDCl_3_): δ = 3.2 (cProp (a)), 4.1 (cProp(b)), 9.3 (cProp*CH*), 17.6 (C-19), 21.6 (Tos-CH_3_), 22.0 (6-*O*-O(CH_2_)_2_*CH*_2_(CH_2_)_2_OTos), 22.5 (C-10), 24.8 (20-*CH*_3_), 29.7 and 29.8 (C-18 and 20*CH*_3_), 28.6 and 30.0 (6-*O*-CH_2_*CH*_2_CH_2_*CH*_2_CH_2_OTos), 32.1 (C-8), 35.4 (C-15), 35.7 (C-14), 43.6 (C-16), 46.7 (C-13), 48.0 (C-7), 57.9 (C-9), 59.8 (N*CH*_2_), 64.3 and 70.3 (6-*O*-(CH_2_)_4_*CH*_2_OTos and 6-*O*-*CH*_2_(CH_2_)_4_OTos), 74.2 (C-20), 80.3 (C-6), 91.4 (Tr*CO*), 96.6 (C-5), 117.8 (C-2), 122.8 (C-1), 127.2 (*p*CTr), 127.3 (*m*CTr), 127.9 (Tos-2,6), 129.3 (*o*CTr), 129.8 (Tos-3,5), 130.6 (C-11), 132.0 (C-12), 133.2 (Tos-C1), 137.2 (C-3), 144.2 (TrC-1), 144.6 (Tos-C4), 151.2 (C-4). Calculated for C_56_H_63_NO_7_S (893.4325), found: 894.4405 ([M+H]^+^).

#### 3.2.4. General Procedure for the Preparation of 6-*O*-Fluoroalkyl-6-*O*-desmethyl-3-*O*-trityl-diprenorphine (**27b**–**d**) Derivatives

A solution of TDDPN (**23**; CAS RN: 157891–92-4, 650 mg, 1 mmol) in dry *N*,*N*-dimethylformamide (3 mL) was added dropwise to a suspension of sodium hydride (60% dispersion in mineral oil, 50 mg, 1.2 mmol, 1.2 equiv.) in anhydrous *N*,*N*-dimethylformamide (3 mL) under an argon atmosphere. The mixture was stirred at RT for 15 min. Then a solution of the appropriate fluoroalkyl toluenesulfonate (**33a**–**c**; 1.6 mmol) in anhydrous *N*,*N*-dimethylformamide (3 mL) was added via syringe, and the mixture was stirred at RT for 20 h. The product mixture was poured into water (30 mL), and the suspension was extracted with dichloromethane (4 × 70 mL). The combined organic extracts were washed with brine (40 mL) and dried over anhydrous Na_2_SO_4_. The resulting crude product was purified by column chromatography on silica gel.

*(5R,6R,7R,9R,13S,14S)-(5α,7α)-17-Cyclopropylmethyl-4,5-epoxy-6-O-(3-fluoropropyl)-18,19-dihydro-α,α-dimethyl-3-triphenylmethoxy-6,14-ethenomorphinan-7-methanol* (6-*O*-(3-fluoropropyl)-6-*O*-desmethyl-3-*O*-trityl-diprenorphine, **27b**, FP-TDDPN).



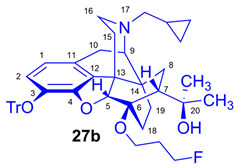



Compound **27b** was synthesized from TDDPN (**23**; 650 mg, 1 mmol) using the general procedure in Section 3.2.4. CC: Kieselgel: 60 g; eluent: hexane–ethyl acetate 7:3 to 6:4 (*v*/*v*); fractions: 25 mL. Yield: 310 mg (43%), mp. 80–105 °C, *R_f_*_A_ = 0.42, *R_f_*_B_ = 0.18, *R_f_*_D_ = 0.83. ^1^H-NMR (CDCl_3_): δ = 0.06 (m, 2H, cProp*CH*_2*syn*_), 0.42 (m, 1H, 19-H_syn_), 0.46 (m, 2H, cProp*CH*_2*anti*_), 0.76 (m, 1H, cProp*CH*), 0.90 (td, ^2^*J*_19anti,19syn_ = 12.6 Hz, ^3^J_19anti,18syn_ = 5.8 Hz, 1H, 19-H_anti_), 0.98 (dd, ^2^*J*_8α,8β_ = 13.3 Hz, ^3^*J*_8α,7β_ = 9.7 Hz, 1H, 8α-H), 1.16 (s, 3H, 20-*CH*_3_), 1.35 (s, 3H, 20-*CH*_3_), 1.38 (td, ^2^*J*_18syn,18anti_ = 13.9 Hz, ^3^*J*_18syn,19anti_ = 5.3 Hz, 1H, 18-H_syn_); 1.43 (dd, ^2^*J*_15eq,15ax_ = 13.0 Hz, ^3^*J*_15eq,16ax_ = 3.7 Hz, 1H, 15-H_eq_), 1.55–1.60 (m, 1H, 15-H_ax_), 1.84 (m, 1H, 7β-H), 1.85–1.92 (m, 3H, 18-H_anti_, CH_2_*CH*_2_CH_2_F(b,b’)), 2.05 (dd, ^2^*J*_10α,10β_ = 18.3 Hz, ^3^*J*_10α,9α_ = 6.3 Hz, 1H, 10α-H), 2.15 (td, ^2^*J*_16ax,16eq_ = 11.7 Hz, ^3^*J*_16ax,15eq_ = 3.7 Hz, 1H, 16-H_ax_), 2.19 (dd, *J*_BA_ = 12.6 Hz, *J*_BX_ = 6.8 Hz, 1H, N*CH*_2_ (a)), 2.32 (dd, *J*_BA_ = 12.7 Hz, *J*_BX_ = 5.7 Hz, 1H, N*CH*_2_ (b)), 2.55 (dd, ^2^*J*_16eq,16ax_ = 11.7 Hz, ^3^*J*_16eq,15ax_ = 4.9 Hz, 1H, 16-H_eq_), 2.77 (ddd, ^2^*J*_8β,8α_ = 13.4 Hz, ^3^*J*_8β,7β_ = 11.7 Hz, ^4^*J*_8β,19syn_ = 3.7 Hz, 1H, 8β-H), 2.86 (d, ^2^J_10β,10α_ = 18.3 Hz, 1H, 10β-H), 2.90 (d, ^3^*J*_9α,10α_ = 6.3 Hz, 1H, 9α-H), 3.60 (dt, *J* = 9.5 Hz, 6.5 Hz, 1H, *CH*_2_CH_2_CH_2_F (a)), 3.89 (dt, *J* = 9.6 Hz, 5.9 Hz, 1H, *CH*_2_CH_2_CH_2_F (a’)), 4.10 (d, ^4^*J*_5β,18anti_ = 1.9 Hz, 1H, 5β-H), 4.42 (m, 1H, CH_2_CH_2_*CH*_2_F(c)), 4.51 (m, 1H, CH_2_CH_2_*CH*_2_F(c’), 5.08 (br s, 1H, 20-OH), 6.20 (d, ^3^*J*_1,2_ = 8.1 Hz, 1H, 1-H), 6.49 (d, ^3^*J*_2,1_ = 8.1 Hz, 1H, 2-H), 7.20–7.26 (m, 9H, Tr(*m*,*p*)), 7.40–7.45 (m, 6H, *o*Tr). ^13^C-NMR (CDCl_3_): δ = 3.2 (cProp(a)), 4.1 (cProp(b)), 9.3 (cPropCH), 17.5 (C-18), 22.5 (C-10), 24.8 (20-CH_3_), 29.7 (20-CH_3_), 29.8 (C-19), 31.7 (d, ^2^*J*_C,F_ = 20.2 Hz, CH_2_*CH*_2_CH_2_F), 32.1 (C-8), 35.4 (C-15), 35.7 (C-14), 43.6 (C-16), 46.8 (C-13), 48.1 (C-7), 57.9 (C-9), 59.8 (N*CH*_2_), 61.0 (d, ^3^*J*_C,F_ = 5.5 Hz, *CH*_2_CH_2_CH_2_F), 74.2 (C-20), 80.6 (C-6), 81.2 (d, ^1^*J*_C,H_ = 165.9 Hz, CH_2_CH_2_*CH*_2_F), 91.4 (Tr*CO*), 96.5 (C-5), 117.9 (C-2), 123.0 (C-1), 127.2 (*p*CTr), 127.3 (*m*CTr), 129.4 (*o*CTr), 130.7 (C-11), 131.9 (C-12), 137.2 (C-3), 144.2 (Tr-C1), 151.3 (C-4). ^19^F-NMR δ = −220.7 dq (*J* = 26.1 Hz, 48.1 Hz). C_47_H_52_FNO_4_ (713.92).

*(5R,6R,7R,9R,13S,14S)-(5α,7α)-17-Cyclopropylmethyl-4,5-epoxy-6-O-(4-fluorobutyl)-18,19-dihydro-α,α-dimethyl-3-triphenylmethoxy-6,14-ethenomorphinan-7-methanol* (6-*O*-(4-fluorobutyl)-6-*O*-desmethyl-3-*O*-trityl-diprenorphine, **27c**, FB-TDDPN).



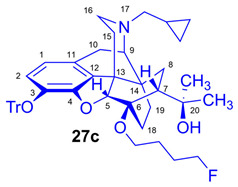



Compound **27c** was synthesized from TDDPN (**23**; 650 mg, 1 mmol) using general procedure in Section 3.2.4. CC: Kieselgel: 60 g; eluent: hexane–ethyl acetate 6:4 (*v*/*v*); fractions: 25 mL. Yield: 440 mg (60%), mp. 93–102 °C, *R_f_*_A_ = 0.75, *R_f_*_B_ = 0.22, *R_f_*_D_ = 0.88. ^1^H-NMR (CDCl_3_): δ = 0.06 (m, 2H, cProp*CH*_2*syn*_), 0.43 (m, 1H, 19-H_syn_), 0.46 (m, 2H, cProp*CH*_2*anti*_), 0.76 (m, 1H, cProp*CH*), 0.90 (td, ^2^*J*_19anti,19syn_ = 12.6 Hz, ^3^J_19anti,18syn_ = 5.7 Hz, 1H, 19-H_anti_), 0.98 (dd, ^2^*J*_8α,8β_ = 13.6 Hz, ^3^*J*_8α,7β_ = 9.7 Hz, 1H, 8α-H), 1.16 (s, 3H, 20-*CH*_3_), 1.35 (s, 3H, 20-*CH*_3_), 1.40 (td, ^2^*J*_18syn,18anti_ = 13.2 Hz, ^3^*J*_18syn,19anti_ = 5.4 Hz, 1H, 18-H_syn_); 1.44 (dd, ^2^*J*_15eq,15ax_ = 13.2 Hz, ^3^*J*_15eq,16ax_ = 2.6 Hz, 1H, 15-H_eq_), 1.57–1.72 (m, 5H, 18-H_anti_, CH_2_CH_2_*CH*_2_CH_2_F(b,b’), CH_2_*CH*_2_CH_2_CH_2_F(c,c’),), 1.83 (m, 1H, 7β-H), 1.90 (td, ^2^*J*_15ax,15eq_ = 12.8 Hz, ^3^*J*_15ax,16eq_ = 5.5 Hz, 1H, 15-Hax), 2.05 (dd, ^2^*J*_10α,10β_ = 18.5 Hz, ^3^J_10α,9α_ = 6.3 Hz, 1H, 10α-H), 2.15 (td, ^2^*J*_16ax,16eq_ = 10.4 Hz, ^3^*J*_16ax,15eq_ = 3.7 Hz, 1H, 16-H_ax_), 2.18 (dd, *J*_BA_ = 12.9 Hz, *J*_BX_ = 6.7 Hz, 1H, N*CH*_2_ (a)), 2.33 (dd, *J*_BA_ = 12.9 Hz, *J*_BX_ = 5.8 Hz, 1H, N*CH*_2_ (b)), 2.55 (dd, ^2^*J*_16eq,16ax_ = 12.0 Hz, ^3^*J*_16eq,15ax_ = 4.7 Hz, 1H, 16-H_eq_), 2.77 (ddd, ^2^*J*_8β,8α_ = 13.6 Hz, ^3^*J*_8β,7β_ = 9.9 Hz, ^4^*J*_8β,19syn_ = 3.8 Hz, 1H, 8β-H), 2.86 (d, ^2^J_10β,10α_ = 18.5 Hz, 1H, 10β-H), 2.90 (d, ^3^*J*_9α,10α_ = 6.3 Hz, 1H, 9α-H), 3.48 (dt, *J* = 9.5 Hz, 6.5 Hz, 1H, *CH*_2_CH_2_CH_2_CH_2_F (a)), 3.83 (dt, *J* = 9.6 Hz, 5.9 Hz, 1H, *CH*_2_CH_2_CH_2_CH_2_F (a’)), 4.11 (d, ^4^*J*_5β,18anti_ = 2.0 Hz, 1H, 5β-H), 4.35 (m, 1H, CH_2_CH_2_CH_2_*CH*_2_F(d)), 4.48 (m, 1H, CH_2_CH_2_CH_2_*CH*_2_F(d’), 5.21 (br s, 1H, 20-OH), 6.19 (d, ^3^*J*_1,2_ = 8.0 Hz, 1H, 1-H), 6.47 (d, ^3^*J*_2,1_ = 8.0 Hz, 1H, 2-H), 7.21–7.24 (m, 9H, Tr(*m*,*p*)), 7.41–7.44 (m, 6H, *o*Tr). ^13^C-NMR (CDCl_3_): δ = 3.2 (cProp (a)), 4.1 (cProp(b)), 9.3 (cPropCH), 17.7 (C-18), 22.5 (C-10), 24.9 (20-CH_3_), 26.4 (d, ^3^*J*_C,F_ = 4.6 Hz, CH_2_*CH*_2_CH_2_CH_2_F), 27.2 (d, ^2^*J*_C,F_ = 20.2 Hz, CH_2_CH_2_*CH*_2_CH_2_F), 29.7 and 29.8 (20-CH_3_ and C-19), 32.1 (C-8), 35.4 (C-15), 35.7 (C-14), 43.6 (C-16), 46.7 (C-13), 48.0 (C-7), 58.0 (C-9), 59.8 (N*CH*_2_), 64.2 (d, ^4^*J*_C,F_ = 1.9 Hz, *CH*_2_CH_2_CH_2_CH_2_F), 74.2 (C-20), 80.4 (C-6), 83.7 (d, ^1^*J*_C,H_ = 165.1 Hz, CH_2_CH_2_CH_2_*CH*_2_F), 91.4 (Tr*CO*), 96.6 (C-5), 117.9 (C-2), 122.9 (C-1), 127.2 (*p*CTr), 127.3 (*m*CTr), 129.3 (*o*CTr), 130.6 (C-11), 132.0 (C-12), 137.2 (C-3), 144.2 (Tr-C1), 151.2 (C-4). ^19^F-NMR δ = −218.5 ddd (*J* = 25.2 Hz, 47.4 Hz, 72.8 Hz). C_48_H_54_FNO_4_ (727.95).

*(5R,6R,7R,9R,13S,14S)-(5α,7α)-17-Cyclopropylmethyl-4,5-epoxy-6-O-(5-fluoropentyl)-18,19-dihydro-α,α-dimethyl-3-triphenylmethoxy-6,14-ethenomorphinan-7-methanol* (6-*O*-(5-fluoropentyl)-6-*O*-desmethyl-3-*O*-trityl-diprenorphine, **27d**, FPe-TDDPN).



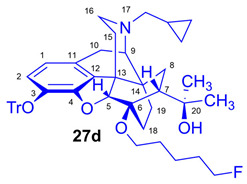



Compound **27d** was synthesized from TDDPN (**23**; 650 mg, 1 mmol) using the general procedure in Section 3.2.4. CC: Kieselgel: 50 g; eluent: hexane–ethyl acetate 6:4 (*v*/*v*); fractions: 25 mL. Yield: 487 mg (66%), mp. 82–92 °C, *R_f_*_A_ = 0.87, *R_f_*_B_ = 0.20, *R_f_*_D_ = 0.90. ^1^H-NMR (CDCl_3_): δ = 0.06 (m, 2H, cProp*CH*_2*syn*_), 0.41 (m, 1H, 19-H_syn_), 0.46 (m, 2H, cProp*CH*_2*anti*_), 0.76 (m, 1H, cProp*CH*), 0.90 (td, ^2^*J*_19anti,19syn_ = 12.5 Hz, ^3^J_19anti,18syn_ = 5.8 Hz, 1H, 19-H_anti_), 0.98 (dd, ^2^*J*_8α,8β_ = 13.1 Hz, ^3^*J*_8α,7β_ = 9.6 Hz, 1H, 8α-H), 1.16 (s, 3H, 20-*CH*_3_), 1.35 (s, 3H, 20-*CH*_3_), 1.37–1.70 (m, 9H, 18-H_syn_, 15-H_eq_, 18-H_anti_, (*CH*_2_)_3_), 1.83 (app t, ^3^*J*_7β,8α_ = 10.0 Hz, 1H, 7β-H), 1.90 (td, ^2^*J*_15ax,15eq_ = 12.7 Hz, ^3^*J*_15ax,16eq_ = 5.5 Hz, 1H, 15-H_ax_), 2.05 (dd, ^2^*J*_10α,10β_ = 18.4 Hz, ^3^J_10α,9α_ = 6.4 Hz, 1H, 10α-H), 2.15 (td, ^2^*J*_16ax,16eq_ = 13.6 Hz, ^3^*J*_16ax,15eq_ = 3.5 Hz, 1H, 16-H_ax_), 2.18 (dd, *J*_BA_ = 12.7 Hz, *J*_BX_ = 6.7 Hz, 1H, N*CH*_2_ (a)), 2.32 (dd, *J*_BA_ = 12.7 Hz, *J*_BX_ = 5.7 Hz, 1H, N*CH*_2_ (b)), 2.55 (dd, ^2^*J*_16eq,16ax_ = 11.9 Hz, ^3^*J*_16eq,15ax_ = 5.0 Hz, 1H, 16-H_eq_), 2.77 (ddd, ^2^*J*_8β,8α_ = 13.1 Hz, ^3^*J*_8β,7β_ = 12.1 Hz, ^4^*J*_8β,19syn_ = 3.7 Hz, 1H, 8β-H), 2.85 (d, ^2^*J*_10β,10α_ = 18.4 Hz, 1H, 10α-H), 2.90 (d, ^3^*J*_9α,10α_ = 6.4 Hz, 1H, 9α-H), 3.45 (dt, *J* = 9.1 Hz, 6.3 Hz, 1H, 6-*O*-*CH*_2_(CH_2_)_4_F (a)), 3.81 (dt, *J* = 9.1 Hz, 6.3 Hz, 1H, 6-*O*-*CH*_2_(CH_2_)_4_F (a’)), 4.11 (d, ^4^*J*_5β,18anti_ = 1.7 Hz, 1H, 5β-H), 4.35 (t, *J* = 6.1 Hz, 1H, CH_2_F (a)), 4.45 (t, *J* = 6.1 Hz, 2H, CH_2_F (a’)), 5.27 (br s, 1H, 20-OH), 6.18 (d, ^3^*J*_1,2_ = 8.0 Hz, 1H, 1-H), 6.47 (d, ^3^*J*_2,1_ = 8.0 Hz, 1H, 2-H), 7.20–7.25 (m, 9H, Tr(*m*,*p*)), 7.41–7.45 (m, 6H, *o*Tr). ^13^C-NMR (CDCl_3_): δ = 3.2 (cProp (a)), 4.1 (cProp(b)), 9.3 (cProp*CH*), 17.7 (C-18), 21.8 (d, ^3^*J*_C,F_ = 5.5 Hz, (CH_2_)_2_*CH*_2_(CH_2_)_2_F), 22.5 (C-10), 24.8 (20-CH_3_), 29.7 (20-CH_3_), 29.8 (C-19), 30.1 (d, ^2^*J*_C,F_ = 20.2 Hz, (CH_2_)_3_*CH*_2_CH_2_F), 30.3 (6-O-CH_2_*CH*_2_(CH_2_)_3_F), 32.1 (C-8), 35.4 (C-15), 35.7 (C-14), 43.6 (C-16), 46.7 (C-13), 48.0 (C-7), 58.0 (C-9), 59.8 (N*CH*_2_), 64.5 (6-*O*-*CH*_2_(CH_2_)_4_F), 74.2 (C-20), 80.3 (C-6), 83.9 (d, ^1^*J*_C,H_ = 164.8 Hz, (CH_2_)_4_*CH*_2_F), 91.4 (Tr*CO*), 96.7 (C-5), 117.8 (C-2), 122.9 (C-1), 127.2 (*p*CTr), 127.3 (*m*CTr), 129.4 (*o*CTr), 130.7 (C-11), 132.0 (C-12), 137.2 (C-3), 144.2 (Tr-C1), 151.3 (C-4). ^19^F-NMR δ = −218.0 m. C_49_H_56_FNO_4_ (741.97).

#### 3.2.5. General Procedure for the Preparation of 6-*O*-Fluoroalkyl-6-*O*-desmethyl-diprenorphine (**28b**–**d**) Derivatives

6-*O*-Fluoroalkyl-6-*O*-desmethyl-3-*O*-trityl-diprenorphine derivative (**27b**–**d**; 0.35 mmol) was dissolved in a mixture of acetic acid (27 mL) and water (7 mL) under an argon atmosphere. The solution was stirred at 100 °C for 5 min. The reaction mixture was cooled to RT and then poured into ice water (40 mL). The solution was basified with a 25 mass % aqueous ammonia solution (pH 9). The mixture was extracted with dichloromethane (4 × 50 mL). The combined organic phase was dried over Na_2_SO_4_, and the solvent was removed under reduced pressure. The resulting crude product was purified by column chromatography on silica gel.

*(5R,6R,7R,9R,13S,14S)-(5α,7α)-17-Cyclopropylmethyl-4,5-epoxy-6-O-(3-fluoropropyl)-18,19-dihydro-α,α-dimethyl-6,14-ethenomorphinan-7-methanol* (6-*O*-(3-fluoropropyl)-6-*O*-desmethyl-diprenorphine, **28b**, FP-DPN).



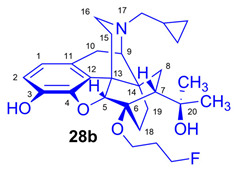



Compound **28b** was synthesized from FP-TDDPN (**27b**; 250 mg, 0.35 mmol) using the general procedure in Section 3.2.5. CC: Kieselgel: 50 g; eluent: hexane–ethyl acetate 7:3 (*v*/*v*); fractions: 25 mL. Yield: 129 mg (78%), mp. 98–115 °C, *R_f_*_A_ = 0.58, *R_f_*_B_ = 0.21, *R_f_*_C_ = 0.28. ^1^H-NMR (CDCl_3_): δ = 0.10 (m, 2H, cProp*CH*_2*syn*_), 0.49 (m, 2H, cProp*CH*_2*anti*_), 0.74 (m, 1H, 19-H_syn_), 0.80 (m, 1H, cProp*CH*), 1.02–1.10 (m, 2H, 19-H_anti_, 8α-H), 1.20 (s, 3H, 20-*CH*_3_), 1.39 (s, 3H, 20-*CH*_3_), 1.67 (dd, ^2^*J*_15eq,15ax_ = 13.1 Hz, ^3^*J*_15eq,16ax_ = 2.7 Hz, 1H, 15-H_eq_), 1.76–1.79 (m, 2H, 18-H_syn_, 18-H_anti_), 1.89–2.05 (m, 4H, 7β-H, 10α-H, CH_2_*CH*_2_CH_2_F(b,b’)), 2.18–2.30 (m, 3H, N*CH*_2_ (a), 15-H_ax_, 16-H_ax_), 2.37 (dd, *J*_BA_ = 12.6 Hz, *J*_BX_ = 5.7 Hz, 1H, N*CH*_2_ (b)), 2.63 (dd, ^2^*J*_16eq,16ax_ = 12.0 Hz, ^3^*J*_16eq,15ax_ = 5.4 Hz, 1H, 16-H_eq_), 2.87 (ddd, ^2^*J*_8β,8α_ = 13.5 Hz, ^3^*J*_8β,7β_ = 12.0 Hz, ^4^*J*_8β,19syn_ = 4.0 Hz, 1H, 8β-H), 2.98 (d, ^2^J_10β,10α_ = 18.7 Hz, 1H, 10β-H), 3.01 (d, ^3^*J*_9α,10α_ = 6.7 Hz, 1H, 9α-H), 3.78 (dt, *J* = 9.3 Hz, 6.3 Hz, 1H, *CH*_2_CH_2_CH_2_F (a)), 4.07 (dt, *J* = 9.3 Hz, 5.9 Hz, 1H, *CH*_2_CH_2_CH_2_F (a’)), 4.42 (d, ^4^*J*_5β,18anti_ = 1.2 Hz, 1H, 5β-H), 4.51 (m, 1H, CH_2_CH_2_*CH*_2_F(c)), 4.60 (m, 1H, CH_2_CH_2_*CH*_2_F(c’), 4.94 (br s, 1H, 3-OH), 5.13 (br s, 1H, 20-OH), 6.50 (d, ^3^*J*_1,2_ = 8.0 Hz, 1H, 1-H), 6.69 (d, ^3^*J*_2,1_ = 8.0 Hz, 1H, 2-H). ^13^C-NMR (CDCl_3_): δ = 3.3 (cProp (a)), 4.2 (cProp(b)), 9.4 (cProp*CH*), 17.8 (C-18), 22.6 (C-10), 24.8 (20-CH_3_), 29.7 (20-CH_3_), 29.9 (C-19), 31.6 (d, ^2^*J*_C,F_ = 20.2 Hz, CH_2_*CH*_2_CH_2_F), 32.3 (C-8), 35.5 (C-15), 35.9 (C-14), 43.7 (C-16), 47.3 (C-13), 48.2 (C-7), 58.1 (C-9), 59.8 (N*CH*_2_), 60.8 (d, ^3^*J*_C,F_ = 4.6 Hz, *CH*_2_CH_2_CH_2_F), 74.4 (C-20), 80.8 (C-6), 81.1 (d, ^1^*J*_C,H_ = 165.3 Hz, CH_2_CH_2_*CH*_2_F), 97.6 (C-5), 116.3 (C-2), 119.5 (C-1), 128.3 (C-11), 132.3 (C-12), 137.2 (C-3), 145.4 (C-4). ^19^F-NMR δ = −221.1 tt (*J* = 26.4 Hz, 47.2 Hz). Calculated for C_28_H_38_FNO_4_ (471.2785), found: 472.2866 ([M+H]^+^).

(*5R,6R,7R,9R,13S,14S)-(5α,7α)-17-Cyclopropylmethyl-4,5-epoxy-6-O-(4-fluorobutyl)-18,19-dihydro-α,α-dimethyl-3-hydroxy-6,14-ethenomorphinan-7-methanol* (6-*O*-(4-fluorobutyl)-6-*O*-desmethyl-diprenorphine, **28c**, FB-DPN).



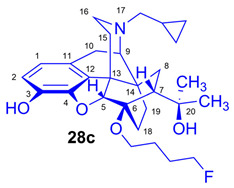



Compound **28c** was synthesized from FB-TDDPN (**27c**; 400 mg, 0.55 mmol) using the general procedure in Section 3.2.5. Column chromatography: Kieselgel: 50 g; eluent: hexane–ethyl acetate 6:4 (*v*/*v*); fractions: 25 mL. Yield: 190 mg (71%), mp. 81–96 °C, *R_f_*_A_ = 0.60, *R_f_*_B_ = 0.13, *R_f_*_C_ = 0.38. ^1^H-NMR (CDCl_3_): δ = 0.09 (m, 2H, cProp*CH*_2*syn*_), 0.49 (m, 2H, cProp*CH*_2*anti*_), 0.73 (m, 1H, 19-H_syn_), 0.80 (m, 1H, cProp*CH*), 1.01–1.09 (m, 2H, 19-H_anti_, 8α-H), 1.20 (s, 3H, 20-*CH*_3_), 1.39 (s, 3H, 20-*CH*_3_), 1.66 (dd, ^2^*J*_15eq,15ax_ = 13.3 Hz, ^3^*J*_15eq,16ax_ = 2.5 Hz, 1H, 15-H_eq_), 1.77–1.82 (m, 6H, 18-H_syn_, 18-H_anti_, CH_2_CH_2_*CH*_2_CH_2_F(b,b’), CH_2_*CH*_2_CH_2_CH_2_F(c,c’),), 1.94 (app t, 1H, 7β-H), 2.03 (dd, ^2^*J*_10α,10β_ = 18.4 Hz, ^3^J_10α,9α_ = 6.5 Hz, 1H, 10α-H), 2.17–2.30 (m, 3H, 15-H_ax_, 16-H_ax_, N*CH*_2_ (a)), 2.37 (dd, *J*_BA_ = 12.7 Hz, *J*_BX_ = 6.0 Hz, 1H, N*CH*_2_ (b)), 2.63 (dd, ^2^*J*_16eq,16ax_ = 12.0 Hz, ^3^*J*_16eq,15ax_ = 5.0 Hz, 1H, 16-H_eq_), 2.86 (ddd, ^2^*J*_8β,8α_ = 13.7 Hz, ^3^*J*_8β,7β_ = 12.2 Hz, ^4^*J*_8β,19syn_ = 3.7 Hz, 1H, 8β-H), 2.97 (d, ^2^J_10β,10α_ = 18.4 Hz, 1H, 10β-H), 3.01 (d, ^3^*J*_9α,10α_ = 6.5 Hz, 1H, 9α-H), 3.67 (dt, *J* = 9.3 Hz, 6.1 Hz, 1H, *CH*_2_CH_2_CH_2_CH_2_F (a)), 3.99 (dt, *J* = 9.3 Hz, 6.1 Hz, 1H, *CH*_2_CH_2_CH_2_CH_2_F (a’)), 4.40 (d, ^4^*J*_5β,18anti_ = 1.3 Hz, 1H, 5β-H), 4.42 (m, 1H, CH_2_CH_2_CH_2_*CH*_2_F(d)), 4.51 (m, 1H, CH_2_CH_2_CH_2_*CH*_2_F(d’), 5.13 (br s, 1H, 3-OH), 5.30 (br s, 1H, 20-OH), 6.50 (d, ^3^*J*_1,2_ = 8.0 Hz, 1H, 1-H), 6.68 (d, ^3^*J*_2,1_ = 8.0 Hz, 1H, 2-H). ^13^C-NMR (CDCl_3_): δ = 3.3 (cProp (a)), 4.1 (cProp(b)), 9.4 (cProp*CH*), 17.9 (C-18), 22.6 (C-10), 24.8 (20-CH_3_), 26.6 (d, ^3^*J*_C,F_ = 4.6 Hz, CH_2_*CH*_2_CH_2_CH_2_F), 27.3 (d, ^2^*J*_C,F_ = 20.1 Hz, CH_2_CH_2_*CH*_2_CH_2_F), 29.7 and 29.9 (20-CH_3_ and C-19), 32.2 (C-8), 35.3 (C-15), 35.9 (C-14), 43.7 (C-16), 47.3 (C-13), 48.1 (C-7), 58.1 (C-9), 59.8 (N*CH*_2_), 64.2 (d, ^4^*J*_C,F_ = 1.0 Hz, *CH*_2_CH_2_CH_2_CH_2_F), 74.5 (C-20), 80.6 (C-6), 83.8 (d, ^1^*J*_C,F_ = 165.0 Hz, CH_2_CH_2_CH_2_*CH*_2_F), 97.6 (C-5), 116.3 (C-2), 119.4 (C-1), 128.2 (C-11), 132.3 (C-12), 137.2 (C-3), 145.4 (C-4). ^19^F-NMR δ = −218.4 ddd (*J* = 25.2 Hz, 47.9 Hz, 93.6 Hz). Calculated for C_29_H_40_FNO_4_ (485.2941), found: 486.3023 ([M+H]^+^).

*(5R,6R,7R,9R,13S,14S)-(5α,7α)-17-Cyclopropylmethyl-4,5-epoxy-6-O-(5-fluoropentyl)-18,19-dihydro-α,α-dimethyl-3-hydroxy-6,14-ethenomorphinan-7-methanol* (6-*O*-(5-fluoropentyl)-6-*O*-desmethyl-diprenorphine, **28d**, FPe-DPN).



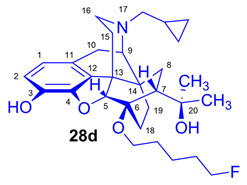



Compound **28d** was prepared from FPe-TDDPN (**27d**; 430 mg, 0.58 mmol) using the general procedure in Section 3.2.5. CC: Kieselgel: 50 g; eluent: hexane–ethyl acetate 6:4 (*v*/*v*); fractions: 25 mL. Yield: 220 mg (76%), mp. 81–90 °C, *R_f_*_A_ = 0.67, *R_f_*_B_ = 0.09, *R_f_*_C_ = 0.43. ^1^H-NMR (CDCl_3_): δ = 0.09 (m, 2H, cProp*CH*_2*syn*_), 0.49 (m, 2H, cProp*CH*_2*anti*_), 0.73 (m, 1H, 19-H_syn_), 0.80 (m, 1H, cProp*CH*), 1.01–1.09 (m, 2H, 19-H_anti_, 8α-H), 1.20 (s, 3H, 20-*CH*_3_), 1.39 (s, 3H, 20-*CH*_3_), 1.46–1.79 (m, 9H, 18-H_syn_, 15-H_eq_, 18-H_anti_, (*CH*_2_)_3_), 1.93 (app t, ^3^*J*_7β,8α_ = 10.0 Hz, 1H, 7β-H), 2.03 (td, ^2^*J*_15ax,15eq_ = 13.8 Hz, ^3^*J*_15ax,16eq_ = 5.7 Hz, 1H, 15-H_ax_), 2.20 (dd, ^2^*J*_10α,10β_ = 18.7 Hz, ^3^J_10α,9α_ = 6.7 Hz, 1H, 10α-H), 2.22–2.30 (m, 2H, 16-H_ax_, N*CH*_2_ (a)), 2.37 (dd, *J*_BA_ = 12.6 Hz, *J*_BX_ = 6.0 Hz, 1H, N*CH*_2_ (b)), 2.63 (dd, ^2^*J*_16eq,16ax_ = 12.1 Hz, ^3^*J*_16eq,15ax_ = 5.0 Hz, 1H, 16-H_eq_), 2.86 (ddd, ^2^*J*_8β,8α_ = 13.8 Hz, ^3^*J*_8β,7β_ = 12.1 Hz, ^4^*J*_8β,19syn_ = 3.7 Hz, 1H, 8β-H), 2.97 (d, ^2^J_10β,10α_ = 18.7 Hz, 1H, 10β-H), 3.00 (d, ^3^*J*_9α,10α_ = 6.7 Hz, 1H, 9α-H), 3.63 (dt, *J* = 8.9 Hz, 6.4 Hz, 1H, 6-*O*-*CH*_2_(CH_2_)_4_F (a)), 3.94 (dt, J = 8.9 Hz, 6.4 Hz, 1H, 6-*O*-*CH*_2_(CH_2_)_4_F (a’)), 4.38 (t, J = 6.0 Hz, 1H, CH_2_F (a)), 4.40 (d, ^4^*J*_5β,18anti_ = 1.1 Hz, 1H, 5β-H), 4.48 (t, *J* = 6.0 Hz, 2H, CH_2_F (a’)), 5.10 (br s, 1H, 3-OH), 5.34 (br s, 1H, 20-OH), 6.49 (d, ^3^*J*_1,2_ = 8.0 Hz, 1H, 1-H), 6.68 (d, ^3^*J*_2,1_ = 8.0 Hz, 1H, 2-H). ^13^C-NMR (CDCl_3_): δ = 3.3 (cProp (a)), 4.1 (cProp(b)), 9.4 (cProp*CH*), 17.7 (C-18), 22.0 (d, ^3^*J*_C,F_ = 5.5 Hz, (CH_2_)_2_*CH*_2_(CH_2_)_2_F), 22.6 (C-10), 24.8 (20-CH_3_), 29.7 (20-CH_3_), 29.9 (C-19), 30.1 (d, ^2^*J*_C,F_ = 20.3 Hz, (CH_2_)_3_*CH*_2_CH_2_F), 30.2 (6-O-CH_2_*CH*_2_(CH_2_)_3_F), 32.2 (C-8), 35.5 (C-15), 35.9 (C-14), 43.7 (C-16), 47.2 (C-13), 48.0 (C-7), 58.1 (C-9), 59.8 (N*CH*_2_), 64.5 (6-*O*-*CH*_2_(CH_2_)_4_F), 74.5 (C-20), 80.5 (C-6), 83.9 (d, ^1^*J*_C,H_ = 164.2 Hz, (CH_2_)_4_*CH*_2_F), 97.7 (C-5), 116.3 (C-2), 119.4 (C-1), 128.3 (C-11), 132.3 (C-12), 137.2 (C-3), 145.5 (C-4). ^19^F-NMR δ = −218.0 m. Calculated for C_30_H_42_FNO_4_ (499.3098), found: 500.3180 ([M+H]^+^).

#### 3.2.6. General Procedure for the Preparation of 6-*O*-(Hydroxyalkyl)-6-*O*-desmethyl-diprenorphine (**29b**–**d**) Derivatives

6-*O*-Hydroxyalkyl-6-*O*-desmethyl-3-*O*-trityl-diprenorphine (**25b**–**d**) derivatives were dissolved in a 4:1 (*v*/*v*) acetic acid–water mixture (1 mg of substrate (**25b**–**d**) in 140 μL of solvent). The reaction mixture was stirred at 100 °C for 10 min. The absence of the starting material was confirmed by analytical TLC. The solution was cooled to RT, poured into ice water, and basified with a 25 mass % aqueous ammonia solution (pH = 9). The product was extracted with dichloromethane. The combined organic layer was washed with brine and dried with anhydrous Na_2_SO_4_, and the solvent was removed under reduced pressure. The residue was purified by column chromatography on silica gel.

*(5R,6R,7R,9R,13S,14S)-(5α,7α)-17-Cyclopropylmethyl-4,5-epoxy-6-O-(3-hydroxypropyl)-18,19-dihydro-α,α-dimethyl-3-hydroxy-6,14-ethenomorphinan-7-methanol* (6-*O*-(3-hydroxypropyl)-6-*O*-desmethyl-diprenorphine, **29b**, HP-DPN).



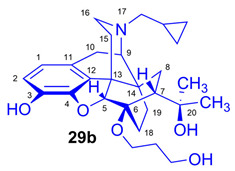



Compound **29b** was prepared from FP-TDDPN (**25b**; 235 mg, 0.33 mmol) using the general procedure in Section 3.2.6. CC: Kieselgel: 30 g; eluent: dichloromethane–methanol 95:5 (*v*/*v*); fractions: 25 mL. Yield: 120 mg (77%), mp. 208–210 °C (dec.), *R_f_*_A_ = 0.33, *R_f_*_D_ = 0.68, *R_f_*_E_ = 0.14. ^1^H-NMR (CDCl_3_): δ = 0.09 (m, 2H, cProp*CH*_2*syn*_), 0.49 (m, 2H, cProp*CH*_2*anti*_), 0.73 (m, 1H, 19-H_syn_), 0.79 (m, 1H, cProp*CH*), 1.01 (m, 1H, 19-H_anti_), 1.06 (dd, ^2^*J*_8α,8β_ = 13.4 Hz, ^3^*J*_8α7β_ = 9.3 Hz, 1H, 8α-H), 1.20 (s, 3H, 20*CH*_3_), 1.39 (s, 3H, 20*CH*_3_), 1.63 (dd, ^2^*J*_15eq,15ax_ = 13.1 Hz, ^3^*J*_15eq,16ax_ = 2.3 Hz, 1H, 15-H_eq_), 1.75–1.85 (m, 4H, 18-H_syn_, 18-H_anti_, CH_2_*CH*_2_CH_2_OH), 1.93 (app t, ^3^*J*_7β,8α_ = 10.1 Hz, 1H, 7β-H), 2.01 (td, ^2^*J*_15ax,15eq_ = 13.6 Hz, ^3^*J*_15ax,16eq_ = 5.7 Hz, 1H, 15-H_ax_), 2.17–2.29 (m, 3H, 10α-H, 16-H_ax_, N*CH*_2_ (a)), 2.36 (dd, *J*_BA_ = 12.9 Hz, *J*_BX_ = 5.7 Hz, N*CH*_2_ (b)), 2.61 (dd, ^2^*J*_16eq,16ax_ = 12.1 Hz, ^3^*J*_16eq,15ax_ = 5.0 Hz, 1H, 16-H_eq_), 2.85 (ddd, ^2^*J*_8β,8α_ = 13.5 Hz, ^3^*J*_8β,7β_ = 12.3 Hz, ^4^*J*_8β,19syn_ = 3.7 Hz, 1H, 8β-H), 2.96 (d, ^2^*J*_10β,10α_ = 18.4 Hz, 1H, 10β-H), 3.00 (d, ^3^*J*_9α,10β_ = 6.4 Hz, 1H, 9α-H), 3.13 (br s, 1H, 6-OCH_2_CH_2_CH_2_*OH*), 3.69–4.17 (m, 4H, 6-*O*-CH_2_CH_2_*CH*_2_OH, 6-*O*-*CH*_2_CH_2_CH_2_OH), 4.39 (d, ^4^*J*_5β,18anti_ = 1.4 Hz, 1H, 5β-H), 5.34 (br s, 1H, 20-*OH*), 6.49 (d, *J*_1,2_ = 8.0 Hz, 1H, 1-H), 6.69 (d, *J*_2,1_ = 8.0 Hz, 1H, 2-H), 6.76 (br s, 1H, 3-OH). ^13^C-NMR (CDCl_3_): δ = 3.3 (cProp (a)), 4.1 (cProp (b)), 9.4 (cProp*CH*), 18.0 (C-18), 22.6 (C-10), 24.9 (20*CH*_3_), 29.7 (20*CH*_3_), 29.9 (C-19), 32.3 (C-8), 33.1 (6-*O-*CH_2_*CH*_2_CH_2_*O*), 35.4 (C-15), 35.9 (C-14), 43.7 (C-16), 47.0 (C-13), 48.0 (C-7), 58.2 (C-9), 59.2 (N*CH*_2_), 59.8 and 61.4 (6-*O*-CH_2_CH_2_*CH*_2_OH and 6-*O*-*CH*_2_CH_2_CH_2_OH), 74.7 (C-20), 80.6 (C-6), 97.4 (C-5), 116.8 (C-2), 119.5 (C-1), 127.7 (C-11), 132.2 (C-12), 137.6 (C-3), 145.5 (C-4). Calculated for C_28_H_39_NO_5_ (469.2828), found: 470.2907 ([M+H]^+^).

*(5R,6R,7R,9R,13S,14S)-(5α,7α)-17-Cyclopropylmethyl-4,5-epoxy-6-O-(4-hydroxybutyl)-18,19-dihydro-α,α-dimethyl-3-hydroxy-6,14-ethenomorphinan-7-methanol* (6-*O*-(4-hydroxybutyl)-6-*O*-desmethyl-diprenorphine, **29c**, HB-DPN).



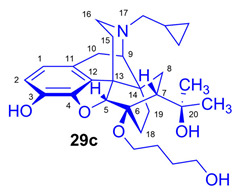



Compound **29c** was prepared from FB-TDDPN (**25c**; 285 mg, 0.39 mmol) using the general procedure in Section 3.2.6. CC: Kieselgel: 40 g; eluent: dichloromethane–methanol 95:5 (*v*/*v*); fractions: 25 mL. Yield: 145 mg (77%), mp. 209–210 °C (dec.), *R_f_*_A_ = 0.45, *R_f_*_D_ = 0.75, *R_f_*_E_ = 0.15. ^1^H-NMR (CDCl_3_): δ = 0.09 (m, 2H, cProp*CH*_2*syn*_), 0.49 (m, 2H, cProp*CH*_2*anti*_), 0.72–0.81 (m, 2H, 19-H_syn_, cProp*CH*), 1.02 (m, 1H, 19-H_anti_), 1.07 (dd, ^2^*J*_8α,8β_ = 12.4 Hz, ^3^*J*_8α,7β_ = 9.3 Hz, 1H, 8α-H), 1.20 (s, 3H, 20*CH*_3_), 1.39 (s, 3H, 20*CH*_3_), 1.58–1.66 (m, 2H, 15-H_eq_, 18-H_syn_), 1.69–1.79 (m, 5H, 6-*O*-CH_2_*CH*_2_*CH*_2_CH_2_OH and 18-H_anti_), 1.93 (app t, ^3^*J*_7β,8α_ = 10.0 Hz, 1H, 7β-H), 2.02 (td, ^2^*J*_15ax,15eq_ = 13.6 Hz, ^3^*J*_15ax,16eq_ = 5.7 Hz, 1H, 15-H_ax_), 2.17–2.30 (m, 3H, 10α-H, 16-H_ax_, N*CH*_2_ (a)), 2.37 (dd, *J*_BA_ = 12.6 Hz, *J*_BX_ = 5.7 Hz, N*CH*_2_ (b)), 2.40 (br s, 1H, 6-*O*-(CH_2_)_4_OH), 2.62 (dd, ^2^*J*_16eq,16ax_ = 11.9 Hz, ^3^*J*_16eq,15ax_ = 5.0 Hz, 1H, 16-H_eq_), 2.85 (ddd, ^2^*J*_8β,8α_ = 13.5 Hz, ^3^*J*_8β,7β_ = 11.9 Hz, ^4^*J*_8β,19syn_ = 3.7 Hz, 1H, 8β-H), 2.97 (d, ^2^*J*_10β,10α_ = 18.4 Hz, 1H, 10β-H), 3.00 (d, ^3^*J*_9α,10α_ = 6.7 Hz, 1H, 9α-H), 3.60–4.06 (m, 4H, 6-*O*-*CH*_2_CH_2_CH_2_*CH*_2_OH), 4.39 (d, ^4^*J*_5β,18anti_ = 1.0 Hz, 1H, 5β-H), 5.31 (br s, 1H, 20-*OH*), 6.40 (br s, 1H, 3-OH), 6.50 (d, *J*_1,2_ = 8.1 Hz, 1H, 1-H), 6.69 (d, *J*_2,1_ = 8.1 Hz, 1H, 2-H). ^13^C-NMR (CDCl_3_): δ = 3.3 (cProp (a)), 4.2 (cProp(b)), 9.4 (cPropCH), 18.1 (C-18), 22.6 (C-10), 24.9 (20-CH_3_), 26.9 and 28.7 (6-*O*-CH_2_*CH*_2_*CH*_2_CH_2_OH), 29.8 and 29.9 (C-19 and 20*CH*_3_), 32.3 (C-8), 35.5 (C-15), 35.9 (C-14), 43.7 (C-16), 47.1 (C-13), 48.0 (C-7), 58.2 (C-9), 59.8 (N*CH*_2_), 62.1 and 63.9 (6-*O*-CH_2_CH_2_CH_2_*CH*_2_OH and 6-*O*-*CH*_2_CH_2_CH_2_CH_2_OH), 74.5 (C-20), 80.6 (C-6), 97.3 (C-5), 116.6 (C-2), 119.6 (C-1), 127.9 (C-11), 132.3 (C-12), 137.5 (C-3), 145.2 (C-4). Calculated for C_29_H_41_NO_5_ (483.2985), found: 484.3065 ([M+H]^+^).

(*5R,6R,7R,9R,13S,14S)-(5α,7α)-17-Cyclopropylmethyl-4,5-epoxy-6-O-(5-hydroxypentyl))-18,19-dihydro-α,α-dimethyl-3-hydroxy-6,14-ethenomorphinan-7-methanol* (6-*O*-(5-hydroxypentyl)-6-*O*-desmethyl-diprenorphine, **29d**, HPe-DPN).



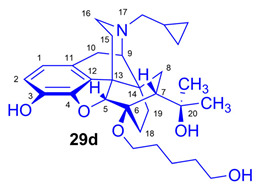



Compound **29d** was prepared from FB-TDDPN (**25d**; 216 mg, 0.29 mmol) using the general procedure in Section 3.2.6. Column chromatography: Kieselgel: 30 g; eluent: dichloromethane–methanol 95:5 (*v*/*v*); fractions: 25 mL. Yield: 105 mg (72%), mp. 86–116 °C, *R_f_*_A_ = 0.51, *R_f_*_D_ = 0.71, *R_f_*_E_ = 0.20. ^1^H-NMR (CDCl_3_): δ = 0.09 (m, 2H, cProp*CH*_2*syn*_), 0.49 (m, 2H, cProp*CH*_2*anti*_), 0.75 (tt, 1H, 19-H_syn_), 0.78 (m, 1H, cProp*CH*), 1.02 (m, 1H, 19-H_anti_), 1.07 (dd, ^2^*J*_8α,8β_ = 13.7 Hz, ^3^*J*_8α7β_ = 9.7 Hz, 1H, 8α-H), 1.20 (s, 3H, 20*CH*_3_), 1.39 (s, 3H, 20*CH*_3_), 1.42–1.59 (m, 3H, (CH_2_)_2_*CH*_2_(CH_2_)_2_, 18-H_syn_), 1.62–1.87 (m, 6H, 15-H_eq_, 18-H_anti_, 6-*O*-CH_2_*CH*_2_CH_2_*CH*_2_CH_2_OH), 1.94 (app t, ^3^*J*_7β,8α_ = 10.0 Hz, 1H, 7β-H), 2.03 (td, ^2^*J*_15ax,15eq_ = 13.6 Hz, ^3^*J*_15ax,16eq_ = 5.3 Hz, 1H, 15-H_ax_), 2.16 (br s, 1H, 6-*O*-(CH_2_)_5_*OH*), 2.20 (dd, ^2^*J*_10α,10β_ = 18.4 Hz, ^3^*J*_10α,9α_ = 6.3 Hz, 1H, 10α-H), 2.23–2.30 (m, 2H, 16-H_ax_ and N*CH*_2_ (a)), 2.37 (dd, *J*_BA_ = 12.7 Hz, *J*_BX_ = 5.8 Hz, N*CH*_2_ (b)), 2.62 (dd, ^2^*J*_16eq,16ax_ = 12.0 Hz, ^3^*J*_16eq,15ax_ = 5.0 Hz, 1H, 16-H_eq_), 2.85 (ddd, ^2^*J*_8β,8α_ = 13.5 Hz, ^3^*J*_8β,7β_ = 12.2 Hz, ^4^*J*_8β,19syn_ = 3.7 Hz, 1H, 8β-H), 2.97 (d, ^2^*J*_10β,10α_ = 18.4 Hz, 1H, 10β-H), 3.00 (d, ^3^*J*_9α,10α_ = 6.3 Hz, 1H, 9α-H), 3.60–3.97 (m, 4H, 6-*O*-*CH*_2_CH_2_CH_2_CH_2_*CH*_2_OH), 4.38 (d, ^4^*J*_5β,18anti_ = 1.7 Hz, 1H, 5β-H), 5.38 (br s, 1H, 20-*OH*), 6.16 (br s, 1H, 3-OH), 6.50 (d, *J*_1,2_ = 8.0 Hz, 1H, 1-H), 6.68 (d, *J*_2,1_ = 8.0 Hz, 1H, 2-H). ^13^C-NMR (CDCl_3_): δ = 3.3 (cProp (a)), 4.1 (cProp(b)), 9.4 (cPropCH), 17.9 (C-18), 21.8 ((CH_2_)_2_*CH*_2_(CH_2_)_2_), 22.6 (C-10), 24.9 (20-CH_3_), 29.7 and 29.9 (C-19 and 20*CH*_3_), 30.0 and 32.3 (6-*O*-CH_2_*CH*_2_CH_2_*CH*_2_CH_2_OH), 32.3 (C-8), 35.5 (C-15), 35.9 (C-14), 43.7 (C-16), 47.0 (C-13), 48.0 (C-7), 58.2 (C-9), 59.8 (N*CH*_2_), 62.4 and 64.2 (6-*O*-(CH_2_)_4_*CH*_2_OH and 6-*O*-*CH*_2_(CH_2_)_4_OH), 74.6 (C-20), 80.5 (C-6), 97.5 (C-5), 116.7 (C-2), 119.5 (C-1), 127.9 (C-11), 132.3 (C-12), 137.5 (C-3), 145.4 (C-4). Calculated for C_30_H_43_NO_5_ (497.3141), found: 498.3221 ([M+H]^+^).

#### 3.2.7. Synthesis of Fluoroalkyl-Tosylates (**33a**–**c**; Figure 6)

*3-Fluoro-1-(toluene-p-sulfonyloxy)propane* (**33a**; FPOTos, CAS RN: 312-68-5).

Compound **33a** was prepared from 3-fluoro-propan-1-ol (**30a**; 5.0 g, 64 mmol) with tosyl chloride (95 mmol) in dichloromethane in the presence of pyridine and DMAP by applying the procedure by Goswami et al. [63]. CC: Kieselgel: 300 g; eluent: hexane–ethyl acetate 10:1 (*v*/*v*); fractions: 250 mL. Yield: 11.46 g (77%) colorless oil. *R_f_* (hexane–ethyl acetate 10:1) = 0.20. ^1^H-NMR (CDCl_3_): δ = 2.04 (dpent, *J*_1_ = 25.7 Hz, *J*_2_ = 6.0 Hz, 2H, FCH_2_*CH*_2_CH_2_O), 2.45 (s, 3H, Tos-*CH*_3_), 4.16 (t, *J* = 6.0 Hz, 2H, FCH_2_CH_2_*CH*_2_), 4.48 (dt, *J*_1_ = 46.9 Hz, *J*_2_ = 5.6 Hz, 2H, F*CH*_2_CH_2_CH_2_O), 7.35 (d, *J* = 8.1 Hz, 2H, Tos-3,5), 7.79 (d, *J* = 8.1 Hz, 2H, Tos-2,6). ^13^C-NMR (CDCl_3_): δ = 21.5 (Tos-*CH*_3_), 29.9 (d, ^2^*J*_C,F_ = 20.4 Hz, OCH_2_*CH*_2_CH_2_F), 66.1 (d, ^3^*J*_C,F_ = 4.8 Hz, O*CH*_2_CH_2_CH_2_F), 79.5 (d, ^1^*J*_C,F_ = 165.8 Hz, OCH_2_CH_2_*CH*_2_Br), 127.8 (Tos-2,6), 129.8 (Tos-3,5), 132.7 (Tos-1), 144.9 (Tos-4). ^19^F-NMR (CDCl_3_): δ = −223.4 m. C_10_H_13_FO_3_S (232.27).

*4-Fluoro-1-(toluene-p-sulfonyloxy)butane* (**33b**; FBOTos, CAS RN: 433-10-3).

Compound **33b** was synthesized from 1,4-butandiol (**30b**; 3.46 g, 38.4 mmol) via 1,4-butandiol-ditosylate (**31**) by using the method by van Wieringen et al. [64]. CC: Kieselgel: 200 g; eluent: hexane–ethyl acetate 7:3 (*v*/*v*); fractions: 100 mL. Yield: 620 mg (**31** → **33b**: 41%). ^1^H-NMR (CDCl_3_): δ = 1.67–1.82 (m, 4H, FCH_2_*CH*_2_*CH*_2_CH_2_O), 2.44 (s, 3H, Tos-*CH*_3_), 4.07 (t, *J* = 6.0 Hz, 2H, F(CH_2_)_3_*CH*_2_O), 4.40 (dt, *J*_1_ = 47.3 Hz, *J*_2_ = 5.7 Hz, 2H, F*CH*_2_(CH_2_)_3_O), 7.34 (d, *J* = 8.4 Hz, 2H, Tos-3,5), 7.78 (d, *J* = 8.4 Hz, 2H, Tos-2,6). ^13^C-NMR (CDCl_3_): δ = 21.6 (Tos-*CH*_3_), 25.0 (d, ^3^*J*_C,F_ = 4.6 Hz, OCH_2_*CH*_2_CH_2_CH_2_F), 26.4 (d, ^2^*J*_C,F_ = 20.1 Hz, O(CH_2_)_2_*CH*_2_CH_2_F), 69.8 (O*CH*_2_CH_2_CH_2_CH_2_F), 83.0 (d, ^1^*J*_C,F_ = 165.1 Hz, O(CH_2_)_3_*CH*_2_F), 127.8 (Tos-2,6), 129.8 (Tos-3,5), 133.0 (Tos-1), 144.8 (Tos-4). ^19^F-NMR (CDCl_3_): δ = −219.5 m. C_11_H_15_FO_3_S (246.30).

*5-Fluoro-1-(toluene-p-sulfonyloxy)pentane* (**33c**, FPeOTos; CAS RN: 565-44-6).

Compound **33c** was prepared from 5-bromo-pentan-1-ol (**30c**; 4.0 g, 38.4 mmol) via 5-tosyloxy-pentanol (**32**) according to the method by Galante et al. [65]. CC: Kieselgel: 350 g; eluent: hexane–ethyl acetate 8:2 (*v*/*v*); fractions: 250 mL. Yield: 2.09 g (**32** → **33c**: 64%) colorless oil. ^1^H-NMR (CDCl_3_): δ = 1.40–1.47 (m, 2H, F(CH_2_)_2_*CH*_2_(CH_2_)_2_O), 1.58–1.66 (m, 2H, FCH_2_*CH*_2_(CH_2_)_3_O), 1.67–1.72 (m, 2H, F(CH_2_)_3_*CH*_2_CH_2_O), 2.44 (s, 3H, Tos-*CH*_3_), 4.03 (t, *J* = 6.4 Hz, 2H, F(CH_2_)_4_*CH*_2_O), 4.39 (dt, *J*_1_ = 47.2 Hz, *J*_2_ = 6.0 Hz, 2H, F*CH*_2_(CH_2_)_4_O), 7.34 (d, *J* = 8.4 Hz, 2H, Tos-3,5), 7.78 (d, *J* = 8.4 Hz, 2H, Tos-2,6). ^13^C-NMR (CDCl_3_): δ = 21.3 (d, ^3^*J*_C,F_ = 4.6 Hz, O(CH_2_)_2_*CH*_2_(CH_2_)_2_F), 21.6 (Tos-*CH*_3_), 29.6 (d, ^2^*J*_C,F_ = 19.2 Hz, O(CH_2_)_3_*CH*_2_CH_2_F), 70.2 (O*CH*_2_(CH_2_)_4_F), 83.6 (d, ^1^*J*_C,F_ = 165.3 Hz, O(CH_2_)_4_*CH*_2_F), 127.8 (Tos-2,6), 129.8 (Tos-3,5), 133.0 (Tos-1), 144.7 (Tos-4). ^19^F-NMR (CDCl_3_): δ = −218.7 m. C_12_H_17_FO_3_S (260.33).

#### 3.2.8. General Procedure for Preparation of *tert*-Butyldiphenylsilyl (TBDPS)-Protected Bromoalkanols (**34a**–**c**; Figure 6)

The corresponding bromoalkanol (**30d–f**; 25 mmol) was dissolved in dry dichloromethane (5 mL) under an argon atmosphere. *N*-ethyldiisopropylamine (3.59 g, 4.6 mL, 27.7 mmol) was added, and the mixture was stirred at RT for 15 min. *tert*-Butyldiphenylsilyl chloride (TBDPSCl; 7.71 g, 7.3 mL, 28 mmol) and DMAP (20 mg) were added, and the reaction mixture was stirred for 48 h at RT. The solvent was removed under reduced pressure, and the crude product was purified by CC on silica gel. CC: Kieselgel: 200 g; eluent: hexane–ethyl acetate 50:1 (*v*/*v*); fractions: 250 mL.

*(3-Bromopropoxy)-tert-butyldiphenylsilane* (**34a**; CAS RN: 177338-13-5).

Compound **34a** was prepared from 3-bromo-propanol (**30d**; 3.52 g, 2.3 mL, 25 mmol) using the general procedure in Section 3.2.8. Yield: 7.4 g (77%) colorless oil. ^1^H-NMR (CDCl_3_): δ = 1.05 (s, 9H, (*CH*_3_)_3_CSiPh_2_), 2.03 (m, 2H, BrCH_2_*CH*_2_CH_2_O), 3.59 (t, *J* = 6.7 Hz, 2H, Br*CH*_2_CH_2_CH_2_), 3.72 (t, *J* = 6.4 Hz, 2H, BrCH_2_CH_2_*CH*_2_O), 7.37–7.45 (m, 6H, 2 × *Ph-4* and 2 × *Ph-3,5*), 7.65–7.68 (m, 4H, 2 × *Ph-2,6*). ^13^C-NMR (CDCl_3_): δ = 19.2 ((CH_3_)_3_*C*), 26.8 ((*CH*_3_)_3_C), 30.5 (OCH_2_*CH*_2_CH_2_Br), 35.4 (OCH_2_CH_2_*CH*_2_Br), 61.3 (O*CH*_2_CH_2_CH_2_Br), 127.7 (*Ph-3,5*), 129.6 and 133.6 (*Ph-4* and *Ph-1*), 135.5 (*Ph-2,6*). C_19_H_25_BrOSi (377.39).

*(4-Bromobutoxy)-tert-butyldiphenylsilane* (**34b**; CAS RN: 125010-58-4).

Compound **34b** was prepared from 4-bromo-butan-1-ol (**30e**; 3.82 g, 25 mmol) using the general procedure in Section 3.2.8. Yield: 5.6 g (57%) colorless oil. ^1^H-NMR (CDCl_3_): δ = 1.05 (s, 9H, (*CH*_3_)_3_CSiPh_2_), 1.67–1.72 (m, 4H, BrCH_2_*CH*_2_*CH*_2_CH_2_O), 3.42 (t, *J* = 6.7 Hz, 2H, Br*CH*_2_(CH_2_)_3_O), 3.69 (t, *J* = 6.0 Hz, 2H, Br(CH_2_)_3_*CH*_2_O), 7.37–7.44 (m, 6H, 2 × *Ph-4* and 2 × *Ph-3,5*), 7.65–7.67 (m, 4H, 2 × *Ph-2,6*). ^13^C-NMR (CDCl_3_): δ = 19.2 ((CH_3_)_3_*C*), 26.8 ((*CH*_3_)_3_C), 29.4 and 31.0 (OCH_2_*CH*_2_*CH*_2_CH_2_Br), 33.8 (O(CH_2_)_3_*CH*_2_Br), 62.9 (O*CH*_2_(CH_2_)_3_Br), 127.6 (*Ph-3,5*), 129.6 and 133.8 (*Ph-4* and *Ph-1*), 135.5 (*Ph-2,6*). C_20_H_27_BrOSi (391.42).

*(4-Bromopentoxy)-tert-butyldiphenylsilane* (**34c**; CAS RN: 125010-60-8).

Compound **34c** was prepared from 5-bromo-pentan-1-ol (**30f**; 4.17 g, 25 mmol) using the general procedure in Section 3.2.8. Yield: 2.9 g (28%) colorless oil. ^1^H-NMR (CDCl_3_): δ = 1.05 (s, 9H, (*CH*_3_)_3_CSiPh_2_), 1.48–1.60 (m, 4H, OCH_2_*CH*_2_*CH*_2_(CH_2_)Br), 1.74–1.85 (m, 2H, O(CH_2_)_3_*CH*_2_CH_2_Br), 3.38 (t, *J* = 6.7 Hz, 2H, Br*CH*_2_(CH_2_)_3_O), 3.68 (t, *J* = 6.4 Hz, 2H, Br(CH_2_)_3_*CH*_2_O), 7.36–7.44 (m, 6H, 2 × *Ph-4* and 2 × *Ph-3,5*), 7.66–7.67 (m, 4H, 2 × *Ph-2,6*). ^13^C-NMR (CDCl_3_): δ = 19.2 ((CH_3_)_3_*C*), 24.5 (O(CH_2_)_2_*CH*_2_(CH_2_)_2_Br), 26.8 ((*CH*_3_)_3_C), 31.6 and 32.5 (O(CH_2_*CH*_2_CH_2_*CH*_2_CH_2_)_2_Br), 33.8 (O(CH_2_)_4_*CH*_2_Br), 63.5 (O*CH*_2_(CH_2_)_4_Br), 127.6 (*Ph-3,5*), 129.5 and 133.9 (*Ph-4* and *Ph-1*), 135.5 (*Ph-2,6*). C_21_H_29_BrOSi (405.45).

### 3.3. Computational Part

#### 3.3.1. Ligand and Receptor Preparation for Docking

The initial 3D structures for the eight ligands under examination were provided as MOL2 files. These structures were then processed using Open Babel (v3.1.1) [74] to generate a diverse set of conformers for each ligand. A conformational search was performed using a genetic algorithm (--conformer) to generate a maximum of fifty conformers (--nconf 50). Each of these conformers was subsequently energy-minimized using the MMFF94s force field [75], and the resulting ensemble of low-energy conformers for each ligand was exported into a single SDF file.

Initial crystal structures of the µOR in its inactive (PDB ID: 4DKL), partial agonist (PDB ID: 7U2K), and active (PDB ID: 5C1M) states were downloaded from the Protein Data Bank. For docking purposes, the receptors were prepared using AutoDockTools v1.5.6 (ADT). This involved removing all non-protein atoms, adding polar hydrogens, and assigning Gasteiger charges. The final prepared receptor structures and the ligand conformers were then converted into PDBQT format to meet the criteria for AutoDock Vina [76].

#### 3.3.2. Molecular Docking Protocol

An initial docking simulation was performed for the pool of conformers for each of the eight ligands into the 4DKL, 5C1M, and 7U2K receptor structures by using AutoDock Vina [76]. Following this initial screening, the conformers yielding the most favorable binding affinity energies were selected. These conformers then underwent a more exhaustive docking simulation consisting of five repeats. From these repeats, the single pose with the best binding energy for each ligand–receptor pair was selected as the starting structure for the subsequent molecular dynamics simulations.

#### 3.3.3. System Preparation for Molecular Dynamics

Receptor Preparation: The original PDB structures of the inactive (4DKL), active (5C1M), and partial agonist (7U2K) receptors were used as the starting point for MD-specific preparation. All non-protein atoms were removed, and the PDBFixer library [77] was employed to repair the three protein structures by adding all missing heavy atoms and hydrogens, assuming a physiological pH of 7.4 to assign appropriate protonation states. The final, complete receptor models were then processed using the pdb4amber utility in AmberTools22 [78] to ensure Amber-compatible atom and residue naming conventions and to apply standard *N*-terminal and *C*-terminal charged patches.

Ligand Parameterization: For each of the selected ligand poses, atom types were assigned from the second-generation General Amber Force Field (GAFF2), and partial atomic charges were calculated using the AM1-BCC method, as implemented in the antechamber module of AmberTools22 [79,80,81]. Any missing force field parameter for the ligands was generated using the parmchk2 utility, creating the corresponding force field modifications (.frcmod) files.

#### 3.3.4. Molecular Dynamics Simulation Protocol

All simulations were performed using the Amber22 simulation package with the sander engine [78].

System Assembly and Membrane Embedding: The tleap module was used to assemble each simulation system. The proteins were described by the ff19SB force field [78], the ligands by the GAFF2 force field [79], and the lipid bilayer by the Lipid 21 force field. Prior to membrane insertion, each receptor–ligand complex was oriented using a custom Python 3.12 script that aligned the principal axis of its Cα atoms with the *Z*-axis. The pre-oriented complex was then embedded into a POPC (1-palmitoyl-2-oleoyl-glycero-3-phosphocholine) bilayer consisting of 200 lipids per leaflet, using the packmol-memgen tool [78]. The resulting protein–membrane system was subsequently solvated with TIP3P water molecules [82] in an orthogonal box with fixed lateral dimensions of 80 Å × 80 Å. The net charge of each system was neutralized by adding a minimal number of chloride (Cl^⊖^) counter-ions.

Minimization, Heating, and Equilibration: A two-stage energy minimization step was first performed to relax each system. Following minimization, systems were gradually heated from 0 K to 300 K over 200 picoseconds (ps) using the Langevin thermostat [83] under constant volume (NVT) conditions, with a gentle restraint on the complex. Subsequently, each system was equilibrated for 500 ps under constant temperature (300 K) and pressure (1 bar) conditions (NPT ensemble), using a restraint on the protein backbone heavy atoms to allow the solvent density to stabilize.

Production MD: Production simulations were run in the NPT ensemble at 300 K and 1 bar. The SHAKE algorithm was used to constrain bonds involving hydrogen, allowing for a 2 femtosecond (fs) timestep [84]. To sample the binding site, the backbone heavy atoms (@C,CA,N) of the receptor located further than 8.0 Å from any ligand atom were gently restrained. Coordinates and restart files were saved every 10,000 steps (20 ps) to generate a continuous trajectory for analysis.

#### 3.3.5. Trajectory Analysis and Visualization

The molecular dynamics trajectories were processed and analyzed using a suite of Python-based scientific libraries. The MDAnalysis (2.9.0) library [85] was employed for core trajectory manipulation tasks, including reading, writing, and aligning the simulation frames. It was also used to perform structural analyses such as root mean square deviation (RMSD) calculations. Numerical operations and data handling were managed using NumPy and Pandas, respectively. All graphical representations and plots of the data were generated using the Matplotlib library (3.10.5) [86].

## 4. Conclusions

We synthesized new 6-*O*-substituted-6-*O*-desmethyl-analogs (**24**–**27b**–**d**) of diprenorphine (**17**) from 3-*O*-trityl-6-*O*-desmethyl-diprenorphine (TDDPN, **23**, the «Luthra precursor»). To study the impact of the 6-*O*-side chain, we prepared numerous derivatives by varying the chain length (*n* = 3, 4, 5): 6-*O*-fluoroalyl-6-*O*-desmethyl-diprenorphine (**28b**–**d**), 6-*O*-(hydroxyalkyl)-6-*O*-desmethyl-diprenorphine (**29b**–**d**), and 6-*O*-tosyloxyalkyl)-6-*O*-desmethyl-diprenorphine (**26a**–**d**) derivatives with longer 6-*O*-side chains (*n* = 3,4,5). Further biochemical/pharmacological investigations of the synthesized novel derivatives could shed light on the influence of the 6-*O*-side chain on OR affinity and subtype selectivity and could deliver new knowledge on structure–activity relationships. The compounds with the 6-*O*-(tosyloxyaklyl)- substituent (**26b**–**d**) can be suitable precursors for the synthesis of 6-*O*-([^18^F]fluoroalkyl)-6-*O*-desmethyl-diprenorphine-type ([^18^F]**28b**–**d**) radiotracers through direct aliphatic nucleophilic substitution (S_N_2) with [^18^F]fluoride under mild basic conditions.

Computational Investigations: Altogether, according to docking energy, all compounds show a preference for the active receptor state. Actives state: Series **28** is slightly better than series **29**; and in series **29**, longer chains are somewhat better. Inactive state: Series **28** is slightly better than series **29,** and longer chains are somewhat better. Partially active state: Series **28** is slightly better than series **29**; and longer chains are somewhat better. Root mean square deviation (RMSD): In series **29**, the lowest values were observed in the active receptor state. The rank order is active > inactive > partially active; in series **28**, the trend is as for series **29**. The most stable one among the active ones were **28a** and **28b**. Center of mass (COM): The displacement of COM is generally the smallest in the active and partially active receptor states. The smallest displacements of COM were observed in the active state for **28**; the inactive state for **28**, **28b**, and **28c**; and the partially active state for **28a**. Our final conclusion is that **28a** is the lead compound, but it is not the only candidate for further pharmacological investigations.

## Figures and Tables

**Figure 1 ijms-26-09427-f001:**
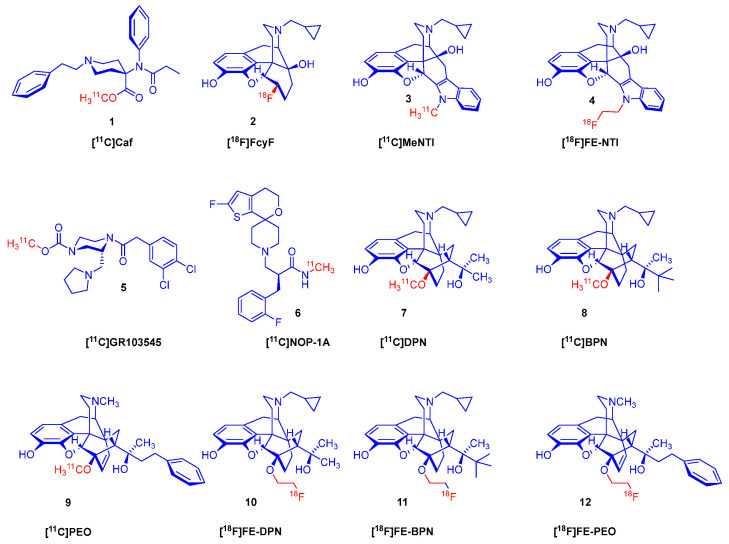
Structures of selected opioid receptor radiotracers.

**Figure 2 ijms-26-09427-f002:**
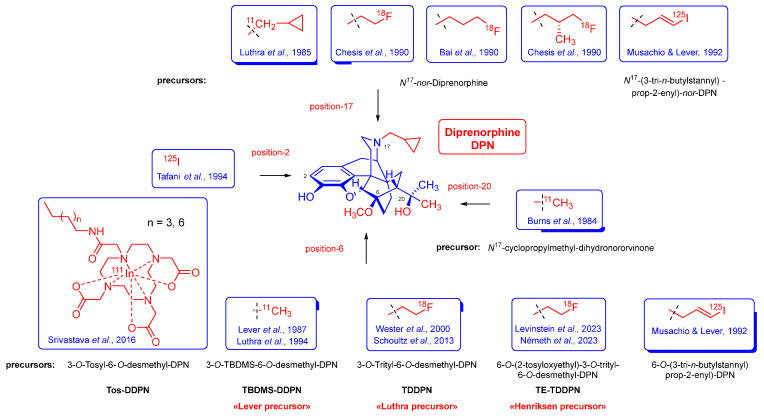
Strategies for the synthesis of radiolabeled diprenorphine derivatives [26,27,28,29,30,31,32,33,34,35,36,37].

**Figure 3 ijms-26-09427-f003:**
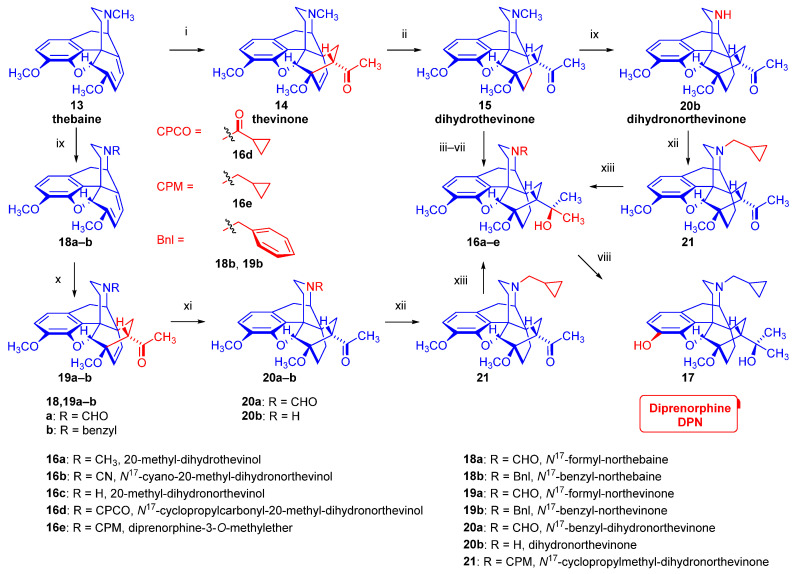
Synthesis of diprenorphine. *Reagents and conditions*: (i) Methyl vinyl ketone, reflux, 1 h, 93%. (ii) H_2_, 10% Pd-C, EtOH, 4 bar, 50 °C, 10h, 75%. (iii) Methylmagnesium iodide, diethylether—benzene, reflux, 2 h. (iv) Cyanogen bromide, CH_2_Cl_2_, room temperature, 16 h. (v) KOH, diethylene glycol, 165–170 °C, 75 min. (vi) Cyclopropylcarbonyl chloride, CH_2_Cl_2_, Et_3_N, 2 days. (vii) LiAlH_4_, THF, reflux, 4 h. (viii) KOH, diethylene glycol, 210–220 °C, nitrogen atmosphere, 2 h. (ix) (1) *N*-demethylation: **A**. 1.6 equiv. DEAD, benzene, reflux, 16 h; **B**. Pyridine hydrochloride, EtOH, room temperature, 24 h. (2) *N*-alkylation: cyclopropylmethyl bromide, NaHCO_3_, DMF, 90 °C, 20 h. (x) Methyl vinyl ketone, reflux, 1 h, 85% (**19a**) and 87% (**19b**). (xi) **A**: H_2_, Pd-C, MeOH, 60 °C, 6 bar, 1 h, 91% (**20a**) and 7 h, 71% (**20b**); **B**: **20a** → **20b**, saturated HCl in EtOH, reflux, 4 h, 81%. (xii) Cyclopropylmethyl bromide, NaHCO_3_, DMF, 90 °C, 20 h, 63%. (xiii) Methylmagnesium iodide, diethylether—toluene, reflux, 2 h, 94%.

**Figure 4 ijms-26-09427-f004:**
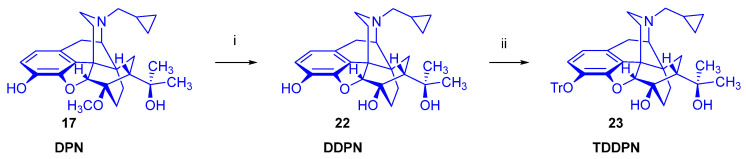
Synthesis of the «Luthra precursor»: 3-*O*-trityl-6-*O*-desmethyl-diprenorphine. *Reagents and conditions*: (i) 15 equiv. LiAlH_4_, 4 equiv. CCl_4_, THF, reflux, 48 h; (ii) trityl chloride, Et_3_N, CH_2_Cl_2_, room temperature, 48 h.

**Figure 5 ijms-26-09427-f005:**
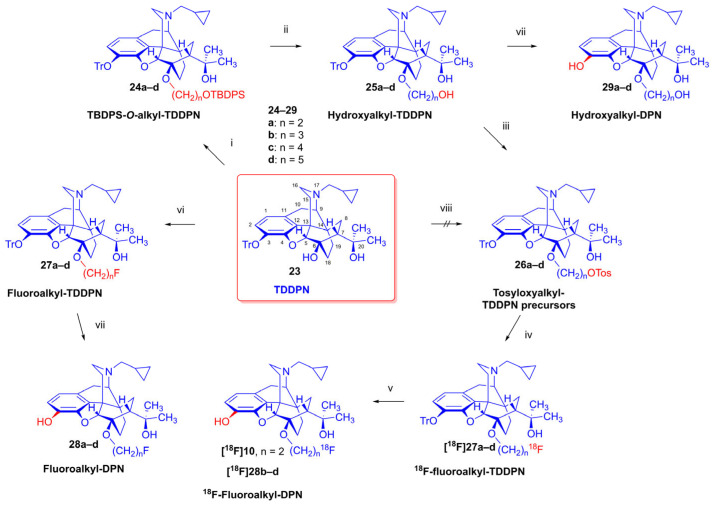
Synthesis of 3-*O*-trityl-6-*O*-(tosyloxyalkyl)-6-*O*-desmethyl-diprenorphine derivatives for use as precursors for radiosynthesis of ^18^F-fluroalkylated-DPN radiotracers. *Reagents and conditions*: (i) TBDPSO(CH_2_)_n_Br, NaH, DMF, RT; (ii) 1M Bu_4_NF in THF, RT; (iii) Tos_2_O, pyridine, CH_2_Cl_2_; (iv) [K^+^⊂K_222_]^18^F^⊖^, CH_3_CN, 90 °C, 10 min; (v) 1M HCl in EtOH, 40 °C, 5–10 min; (vi) TosO(CH_2_)_n_F, NaH, DMF, RT; (vii) AcOH-H_2_O, 100 °C, 5–10 min; (viii) TosO(CH_2_)nOTos, NaH, DMF, RT.

**Figure 6 ijms-26-09427-f006:**
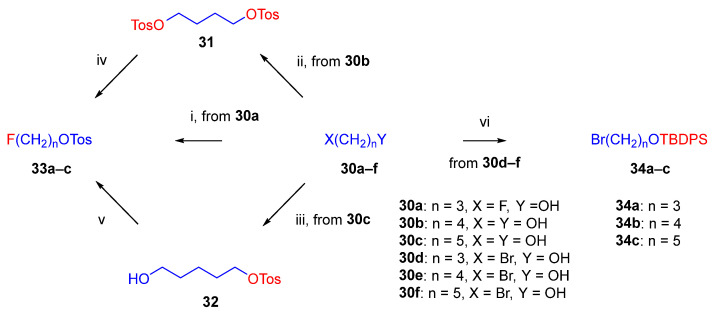
Synthesis of fluoroalkyl-tosylates and bromoalkoxy-*tert*-butyl-diphenylsilanes. *Reagents and conditions*: (i) TosCl, DMAP, pyridine, CH_2_Cl_2_, 16 h, RT; (ii) TosCl, Et_3_N, CH_2_Cl_2_, 16 h, RT; (iii) TosCl, pyridine, CH_2_Cl_2_, 16 h, RT; (iv) 1M Bu_4_NF, acetonitrile, 24 h, reflux; (v) DAST, CH_2_Cl_2_, 4 h, RT; (vi) TBDMSCl, DIPEA, DMAP, CH_2_Cl_2_, 16 h, RT.

**Figure 7 ijms-26-09427-f007:**
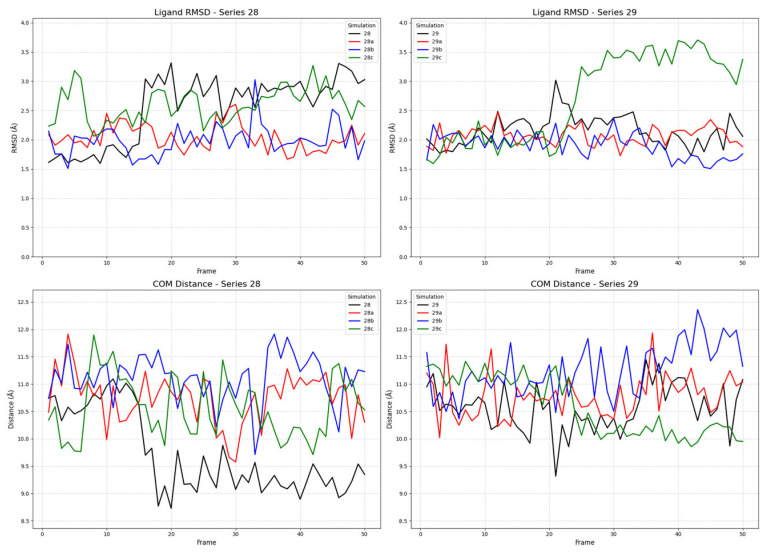
Ligand RMSD (**top**) and COM distance (**bottom**) for series **28** (**left**) and series **29** (**right**) compounds in complex with the active receptor state.

**Figure 8 ijms-26-09427-f008:**
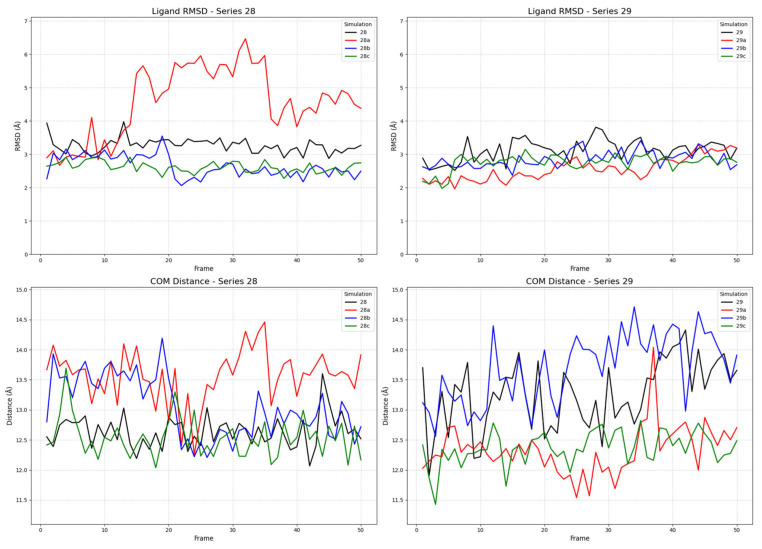
Ligand RMSD (**top**) and COM distance (**bottom**) for series **28** (**left**) and series **29** (**right**) compounds in complex with the inactive receptor state. The instability of ligand **28a** (red trace) is evident.

**Figure 9 ijms-26-09427-f009:**
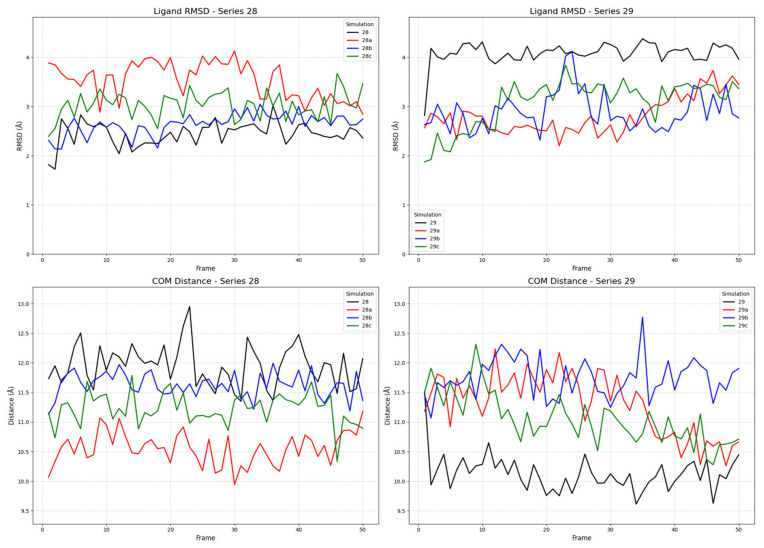
Ligand RMSD (**top**) and COM distance (**bottom**) for series **28** (**left**) and series **29** (**right**) compounds in complex with the partial agonist receptor state. The conformational instability of ligand **28a** (red trace, **top left**) and ligand **29** (black trace, **top right**) is evident.

**Figure 10 ijms-26-09427-f010:**
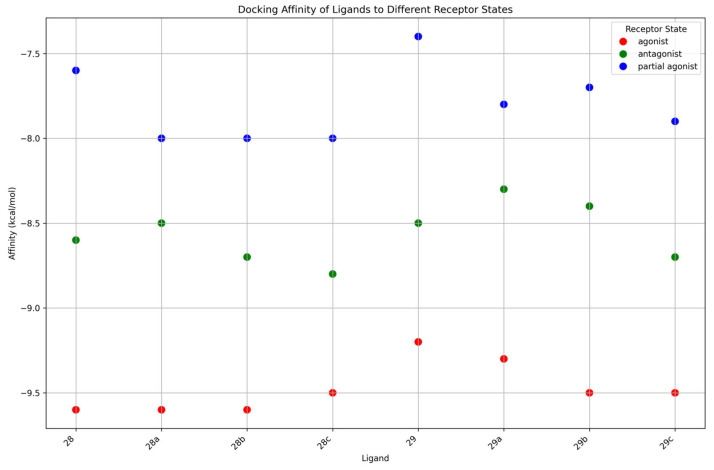
Docking affinity scores (kcal/mol) for all ligands across the three receptor states. The agonist (active) state consistently shows the most favorable (most negative) binding energy for all compounds.

**Table 1 ijms-26-09427-t001:** ^1^H- and ^13^C-NMR chemical shifts and coupling constants of TP-TDDPN (**26b**), TB-TDDPN (**26c**), and TPe-TDDPN (**26d**) ^a^ in CDCl_3_.

Position	TP-TDDPN (26b)		TB-TDDPN (26c)		TPe-TDDPN (26d)	
	δ ^13^C	δ ^1^H (m, *J* in Hz)	δ ^13^C	δ ^1^H (m, *J* in Hz)	δ ^13^C	δ ^1^H (m, *J* in Hz)
1	123.2	6.20 (d, 8.4)	122.9	6.18 (d, 8.0)	122.8	6.18 (d, 8.4)
2	118.0	6.49 (d, 8.4)	117.9	6.46 (d, 8.0)	117.8	6.46 (d, 8.4)
3	137.2	-	137.2	-	137.2	-
4	151.2	-	151.2	-	151.2	-
5β	96.2	3.99 (d, 2.3)	96.5	4.05 (d, 1.7)	96.6	4.08 (d, 1.7)
6	80.8	-	80.5	-	80.3	-
7β	48.1	1.75 (app t, 10.1)	48.0	1.78 (app t, 9.4)	48.0	1.80 (app t, 9.7)
8α	32.1	0.95 (dd, 13.3, 9.7)	32.1	0.96 (dd, 13.3, 9.7)	32.1	0.97 (dd, 13.1, 9.8)
8β		2.74 (ddd, 13.4, 12.4, 3.9)		2.75 (ddd, 13.8, 12.4, 3.9)		2.76 (ddd, 13.1, 11.9, 3.9)
9α	57.9	2.89 (d, 6.1)	57.9	2.89 (d, 6.0)	57.9	2.89 (d, 6.4)
10α	22.5	2.05 (dd, 18.7, 6.1)	22.5	2.04 (dd, 18.8, 6.0)	22.5	2.04 (dd, 18.4, 6.4)
10β		2.85 (d, 18.7)		2.84 (d, 18.8)		2.85 (d, 18.4)
11	130.7	-	130.6	-	130.6	-
12	131.9	-	131.9	-	132.0	-
13	46.8	-	46.7	-	46.7	-
14	35.7	-	35.7	-	35.7	-
15_ax_	35.4	1.86 (td, 13.2, 5.4)	35.4	1.87 (td, 13.9, 5.6)	35.4	1.89 (td, 13.7, 5.7)
15_eq_		1.30–1.41(m ^b^)		1.42 (dd, 13.1, 2.7)		1.26–1.62 (m ^d^)
16_ax_	43.5	2.14 (td, 13.0, 3.6)	43.6	2.14 (td, 11.5, 3.8)	43.6	2.14 (td, 13.2, 3.8)
16_eq_		2.54 (dd, 11.9, 5.0)		2.54 (dd, 12.0, 5.4)		2.55 (dd, 12.1, 4.7)
18_syn_	17.4	1.30–1.41 (m ^b^)	17.6	1.32–1.37 (m)	17.6	1.26–1.62 (m ^d^)
18_anti_		1.49 (m)		1.40–1.63 (m ^c^)		1.26–1.62 (m ^d^)
19_syn_	30.3	0.43 (m)	29.7	0.41 (m)	29.8	0.42 (tt, 12.7, 5.7)
19_anti_		0.88 (td, 12.4, 5.7)		0.88 (td, 13.6, 5.7)		0.88 (td, 13.3, 5.9)
20	74.2	-	74.2	-	74.2	-
**Others**						
20-*CH*_3_	24.8	1.13 (s)	24.9	1.14 (s)	24.8	1.14 (s)
20-*CH*_3_	29.7	1.26 (s)	29.7	1.31 (s)	29.7	1.32 (s)
6-*O*-(CH_2_)_n_O*Tos*						
Tos-*CH*_3_	21.6	2.41 (s)	21.6	2.42 (s)	21.6	2.43 (s)
Tos-3,5	-	7.31 (d, 8.1)	-	7.33 (d, 8.1)	-	7.33 (d, 8.4)
Tos-2,6	-	7.78 (d, 8.1)	-	7.79 (d, 8.1)	-	7.79 (d, 8.4)
*Tos*-C1	133.1	-	133.1	-	133.2	-
*Tos*-C2,6	129.9	-	127.9	-	127.9	-
*Tos*-C3,5	129.8	-	129.8	-	129.8	-
*Tos*-C4	144.6	-	144.6	-	144.6	-
cProp*CH*_2*syn*_	3.2	0.05 (m)	3.2	0.05 (m)	3.2	0.05 (m)
cProp*CH*_2*anti*_	4.1	0.43 (m)	4.1	0.46 (m)	4.1	0.46 (m)
cProp*CH*	9.3	0.75 (m)	9.3	0.75 (m)	9.3	0.75 (m)
N*CH*_2_ (a)	59.8	2.18 (dd, 12.9, 6.7)	59.8	2.17 (dd, 12.8, 6.7)	59.8	2.18 (dd, 12.6, 6.7)
N*CH*_2_ (b)		2.32 (dd, 12.9, 5.6)		2.32 (dd, 12.8, 5.7)		2.32 (dd, 12.6, 5.7)
Trityl (Tr)						
Tr (*m,p*)	-	7.21–7.24 (m)	-	7.19–7.24 (m)	-	7.18–7.22 (m)
Tr (*o*)	-	7.39–7.42 (m)	-	7.38–7.42 (m)	-	7.39–7.44 (m)
Ph_3_CO	91.5	-	91.4	-	91.4	-
*p*CTr	127.3	-	127.3	-	127.2	-
*m*CTr	127.4	-	127.4	-	127.3	-
*o*CTr	129.4	-	129.3	-	129.3	-
TrC1	144.2	-	144.2	-	144.2	-
20-OH	-	4.83 (br s)	-	5.07 (br s)	-	5.18 (br s)
6-*O*-(*CH*_2_)_3_*O*Tos	33.4	1.79–1.84 (m)	-	-	-	-
6-*O*-(*CH*_2_)_3_*O*Tos	60.4, 67.4	3.49–4.12 (m)	-	-	-	-
6-*O*-(*CH*_2_)_4_*O*Tos	-	-	25.7, 26.7	1.40–1.63 (m ^c^)	-	-
6-*O*-(*CH*_2_)_4_*O*Tos	-	-	64.0, 70.2	3.36–3.99 (m)	-	-
6-*O*-(*CH*_2_)_5_*O*Tos	-	-	-	-	64.3, 70.3	1.26–1.62 (m)
6-*O*-(*CH*_2_)_5_*O*Tos	-	-	-	-	28.6, 30.0	3.36–3.98 (m)

^a^: Observation frequency: 500.130 MHz (for ^1^H-NMR), 125.758 MHz (for ^13^C-NMR), in CDCl_3_, δ in ppm. ^b^: Overlapping signals of H-15_eq_ and H-18_syn_. ^c^: Overlapping multiplets of H-18_anti_ and 6-O-(*CH*_2_)_4_ OTos. ^d^: Overlapping multiplets of H-15_eq_ and H-18_syn_.

**Table 2 ijms-26-09427-t002:** ^1^H- and ^13^C-NMR chemical shifts and coupling constants of FP-DPN (**28b**), FB-DPN (**28c**), and FPe-DPN (**28d**) ^a^ in CDCl_3_.

Position	TP-DPN (28b)		TB-DPN (28c)		TPe-DPN (28d)	
	δ ^13^C	δ ^1^H (m, *J* in Hz)	δ ^13^C	δ ^1^H (m, *J* in Hz)	δ ^13^C	δ ^1^H (m, *J* in Hz)
1	119.5	6.50 (d, 8.0)	119.4	6.50 (d, 8.0)	119.4	6.49 (d, 8.0)
2	116.3	6.69 (d, 8.0)	116.3	6.68 (d, 8.0)	116.3	6.68 (d, 8.0)
3	137.2	-	137.2	-	137.2	-
4	145.4	-	145.4	-	145.5	-
5β	97.6	4.42 (d, 1.2)	97.6	4.40 (d, 1.3)	97.7	4.40 (d, 1.1)
6	80.8	-	80.6	-	80.5	-
7β	48.2	1.89–2.05 (m ^e^)	48.1	1.94 (m)	48.0	1.93 (app t, 10.0)
8α	32.3	1.02–1.10 (m ^f^)	32.2	1.01–1.09 (m ^m^)	32.2	1.01–1.09 (m ^s^)
8β		2.87 (ddd, 13.5, 12.0, 4.0)		2.86 (ddd, 13.7, 12.2, 3.7)		2.86 (ddd, 13.8, 12.1, 3.7)
9α	58.1	3.01 (d, 6.7)	58.1	3.01 (d, 6.5)	58.1	3.00 (d, 6.7)
10α	22.6	1.89–2.05 (m ^e^)	22.6	2.03 (dd, 18.4, 6.5)	22.6	2.20 (dd, 18.7, 6.7)
10β		2.98 (d, 18.7)		2.97 (d, 18.4)		2.97 (d, 18.7)
11	128.3	-	128.2	-	128.3	-
12	132.3	-	132.3	-	132.3	-
13	47.3	-	47.3	-	47.2	-
14	35.9	-	35.9	-	35.9	-
15_ax_	35.5	2.18–2.30 (m ^g^)	35.3	2.17–2.30 (m ^n^)	35.5	2.03 (td, 13.8, 5.7)
15_eq_		1.67 (dd, 13.1, 2.7)		1.66 (dd, 13.3, 2.5)		1.46–1.79 (m ^t^)
16_ax_	43.7	2.18–2.30 (m ^g^)	43.7	2.17–2.30 (m ^n^)	43.7	2.22–2.30 (m) ^u^
16_eq_		2.63 (dd, 12.0, 5.4)		2.63 (dd, 12.0, 5.0)		2.63 (dd, 12.1, 5.0)
18_syn_	17.8	1.76–1.79 (m ^h^)	17.9	1.77–1.82 (m ^o^)	17.7	1.46–1.79 (m ^t^)
18_anti_		1.76–1.79 (m ^h^)		1.77–1.82 (m ^o^)		1.46–1.79 (m ^t^)
19_syn_	29.9	0.74 (m)	29.7	0.73 (m)	29.9	0.73 (m)
19_anti_		1.02–1.10 (m ^f^)		1.01–1.09 (m ^m^)		1.01–1.09 (m ^s^)
20	74.4	-	74.5	-	74.5	-
**Others**						
20-*CH*_3_	24.8	1.20 (s)	24.8	1.20 (s)	24.8	1.20 (s)
20-*CH*_3_	29.7	1.39 (s)	29.9	1.39 (s)	29.7	1.39 (s)
cProp*CH*_2*syn*_	3.3	0.10 (m)	3.3	0.09 (m)	3.3	0.09 (m)
cProp*CH*_2*anti*_	4.2	0.49 (m)	4.1	0.49 (m)	4.1	0.49 (m)
cProp*CH*	9.4	0.80 (m)	9.4	0.80 (m)	9.4	0.80 (m)
N*CH*_2_ (a)	59.8	2.18–2.30 (m ^g^)	59.8	2.17–2.30 (m ^n^)	59.8	2.22–2.30 (m) ^u^
N*CH*_2_ (b)		2.37 (dd, 12.6, 5.7)		2.37 (dd, 12.7, 6.0)		2.37 (dd, 12.6, 6.0)
20-OH	-	5.13 (br s)	-	5.30 (br s)	-	5.34 (br s)
3-OH		4.94 (br s)		5.13 (br s)		5.10 (br s)
6-*O*-CH_2_*CH*_2_CH_2_F (b,b’)	31.6 d ^b^	1.89–2.05 (m ^e^)	-	-	-	-
6-*O*-*CH*_2_CH_2_CH_2_F (a)	60.8 d ^c^	3.78 (dt, 9.3, 6.3, 1H)	-	-	-	-
6-*O*-*CH*_2_CH_2_CH_2_F (a’)		4.07 (dt, 9.3, 5.9, 1H)	-	-	-	-
6-*O*-CH_2_CH_2_*CH*_2_F (c,c’)	81.1 d ^d^	4.51–4.60 (m, 2H)	-	-	-	-
6-*O*-*CH*_2_CH_2_CH_2_CH_2_F (a,a’)	-	-	64.2 d ^i^	3.67 (dt, 9.3, 6.1)	-	-
	-	-		3.99 (dt, 9.3, 6.1)	-	-
6-*O*-CH_2_*CH*_2_CH_2_CH_2_F (b,b’)	-	-	26.6 d ^j^	1.77–1.82 (m ^o^)	-	-
6-*O*-CH_2_CH_2_*CH*_2_CH_2_F (c,c’)	-	-	27.3 d ^k^	1.77–1.82 (m ^o^)	-	-
6-*O*-CH_2_CH_2_CH_2_*CH*_2_F (d,d’)	-	-	83.8 ^l^ d	4.42–4.51 (m)	-	-
6-*O*-*CH*_2_CH_2_CH_2_CH_2_CH_2_F (a)	-	-	-	-	64.5	3.63 (dt, 8.9, 6.4)
6-*O*-*CH*_2_CH_2_CH_2_CH_2_CH_2_F (a’)	-	-	-	-		3.94 (dt, 8.9, 6.4)
6-*O*-CH_2_*CH*_2_CH_2_CH_2_CH_2_F (b,b’)	-	-	-	-	22.0 d ^p^	1.46–1.79 (m ^t^)
6-*O*-CH_2_CH_2_*CH*_2_CH_2_CH_2_F (c,c’)	-	-	-	-	30.1 d ^q^	1.46–1.79 (m ^t^)
6-*O*-CH_2_CH_2_CH_2_*CH*_2_CH_2_F (d,d’)	-	-	-	-	30.2	1.46–1.79 (m ^t^)
6-*O*-CH_2_CH_2_CH_2_CH_2_*CH*_2_F (e)	-	-	-	-	83.9 d ^r^	4.38 (t, 6.0)
6-*O*-CH_2_CH_2_CH_2_CH_2_*CH*_2_F (e’)	-	-	-	-		4.48 (t, 6.0)

^a^: Observation frequency: 500.130 MHz (for ^1^H-NMR), 125.758 MHz (for ^13^C-NMR), in CDCl_3_, δ in ppm. ^b^: ^2^*J*_C,F_ = 20.2 Hz. ^c^: ^3^*J*_C,F_ = 4.6 Hz. ^d^: ^1^*J*_C,F_ = 97.6 Hz. ^e^: Overlapping signals of H-7β, H-10α, and 6-O-CH_2_*CH*_2_CH_2_F (b,b’). ^f^: Overlapping signals of H-8α and H-19_anti_. ^g^: Overlapping multiplets of H-15_ax_, H-16_ax_, and N*CH*_2_ (a). ^h^: Overlapping signals of H-18_syn_ and H-18_anti_. ^i^: ^4^*J*_C,F_ = 1.0Hz. ^j^: ^3^J_C,F_ = 4.6 Hz. ^k^: ^2^*J*_C,F_ = 20.1 Hz. ^l^: ^1^*J*_C,F_ = 165.0 Hz. ^m^: Overlapping signals of H-8α and H-19_anti_. ^n^: Overlapping multiplets of H-15_ax_, H-16_ax_, and N*CH*_2_ (a). ^o^: Overlapping signals of H-18_anti_, H-18_syn_, 6-*O*-CH_2_*CH*_2_CH_2_CH_2_F (b,b’), and 6-*O*-CH_2_CH_2_*CH*_2_CH_2_F (c,c’). ^p^: ^3^*J*_C,F_ = 5.5 Hz. ^q^: ^2^*J*_C,F_ = 20.3 Hz. ^r^: ^1^*J*_C,F_ = 164.2 Hz. ^s^: Overlapping signals of H-8α and H-19_anti_. ^t^: Overlapping multiplets of H-18_syn_, H-15_eq_, H-18_anti_, and 6-*O*-CH_2_*CH*_2_*CH*_2_*CH*_2_CH_2_F. ^u^: Overlapping multiplets of H-16_ax_ and N*CH*_2_ (a).

## Data Availability

The raw data supporting the conclusions of this article will be made available by the corresponding authors (J.M. and F.Ö.) on request.

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
