# Peer review of "Synthesis and In Silico Profile Modeling of 6-O-Fluoroalkyl-6-O-desmethyl-diprenorphine Analogs"

_ijms, 2025, doi:10.3390/ijms26199427_

Round 1
Reviewer 1 Report
Comments and Suggestions for Authors
Síntese e modelagem de perfil in silico de análogos de 6- O -fluoroalquil-6- O -desmetil-diprenor- 2- fina
O manuscrito intitulado "Síntese e modelagem de perfil in silico de análogos de 6-O-fluoroalquil-6-O-desmetil-diprenor-2-fina" aborda um tópico científico de alta relevância, pois se insere no contexto atual de síntese de análogos bioativos aliados a estudos in silico , área de crescente interesse para o desenvolvimento de novos compostos com potencial farmacológico.
The study demonstrates a solid theoretical basis, supported by an adequate and up-to-date literature review, which allows the reader to understand the importance of the proposed structural modification and the rationale for the molecular modeling studies. In addition, the review integrates pertinent references, covering both synthetic and in silico profiling methodological aspects.
The methodology described for the synthesis of the analogues is clear and well organized, with potential for reproducibility in other laboratories. The computational modeling studies add value to the work, as they provide predictive information that can guide future experimental investigations, reducing both costs and research time.
The manuscript contributes to the advancement of knowledge at the interface between organic synthesis and computational medicinal chemistry, presenting data that may assist in the development of new molecules with pharmacological applications.
In section 2.1.3. 6-O-desmethyl-6,14-ethenomorphinans and Synthesis of 6-O-substituted-6-O-desmethyl-diprenorphine derivatives, number the structures in the text in the same way as in the previous section.
Melhorar a resolução das estruturas perderam a qualidade na conversão do pdf
Tabela 1- página 9, Tabela 2 página 11, indicar constantes de acoplamentos faltou indicar a distância de ligação (J1, J3 etc)
Supplementar information
Table S1 and S2 as constantes de acoplamentos faltou indicar a distância de ligação, igual foi feito na descrição do artigo Improve the resolution of the structures, as their quality was lost during the PDF conversion.
Table 1 (page 9) and Table 2 (page 11): Indicate the coupling constants. The bond distances (J1, J3, etc.) were not provided and should be included.
Supplementary Information – Tables S1 and S2: The coupling constants should also indicate the bond distances, as was done in the description within the main article.
In this spectrum, two values for deuterated chloroform appear; however, the second one does not correspond to δ 7.26, and based on the alignment of the main spectrum, it cannot be justified as the deuterated solvent.
- Article, page 16, and Spectrum S1 (SM): The signal at δ 3.59–3.67 (m, 2H) shows an integration corresponding to 4 protons in the spectrum, which should be revised. (see attachment in pdf)
- The signals at δ 2.85 (d, ²J₁₀β,₁₀α = 18.3 Hz, 1H, 10β-H) and δ 2.89 (d, ³J₉α,₁₀α = 6.4 Hz, 1H, 9α-H) require a review of both chemical shifts and integration values.
- The mass spectra for all derivatives were not presented and should be included.
- Sugere-se colocar os valores de deslocamento químico diretamente acima dos picos de RMN para facilitar a interpretação de todos os espectros.

Author Response
Answers to referee-1
We would like to thank you for the evaluation of our manuscript. Please find below our responses to your comments.
Comments and Suggestions for Authors
The manuscript entitled: «Synthesis and in silico profile modelling of 6-fluoroalkyl-6-O-desmethyl-diprenorphine analogues» addresses a highly relevant scientific topic, as it falls within the current context of the synthesis of bioactive analogues combined with in silico studies, an area of growing interest for the development of new compounds with pharmacological potential.
The study demonstrates a solid theoretical basis, supported by an adequate and up-to-date literature review, which allows the reader to understand the importance of the proposed structural modification and the rationale for the molecular modeling studies. In addition, the review integrates pertinent references, covering both synthetic and in silico profiling methodological aspects.
The methodology described for the synthesis of the analogues is clear and well organized, with potential for reproducibility in other laboratories. The computational modeling studies add value to the work, as they provide predictive information that can guide future experimental investigations, reducing both costs and research time.
The manuscript contributes to the advancement of knowledge at the interface between organic synthesis and computational medicinal chemistry, presenting data that may assist in the development of new molecules with pharmacological applications.
We are thankful for this positive evaluation, which has helped us to improve our manuscript.
In section 2.1.3. 6-O-desmethyl-6,14-ethenomorphinans and Synthesis of 6-O-substituted-6-O-desmethyl-diprenorphine derivatives, number the structures in the text in the same way as in the previous section.
In some cases, we inserted the corresponding (already existing) compound-numbers (17, 22, 23). To improve clarity, we now include additional Supplementary Material ChartS1 with the structures of orvinol (BPN, DHE, PEO), and 6-O-desmethyl orvinol derivatives (DBPN, DDHE, and DPEO), which also circumvents the renumbering issue.
Chart S1. Chemical structures of selected orvinol and 6-O-desmethyl-orvinol derivatives
Supplementary information
Table 1 (page 9) and Table 2 (page 11): Indicate the coupling constants. The bond distances (J1, J3, etc.) were not provided and should be included. Supplementary Information – Tables S1 and S2: The coupling constants should also indicate the bond distances, as was done in the description within the main article.
We believe that these Tables (Table 1 (page 9), Table 2 (page 11), Table S1 and Table S2 in Supplementary Materials) are already very bulky and information dense, and therefore omitted some of this information. We paid special attention to giving a precise NMR analysis and complete assignation of the spectra. Please see for example compound 24b (page 17 Line 529): 2.75 (ddd, 2J8β,8α = 13.5 Hz, 3J8β,7β = 12.0 Hz, 4J8β,19syn = 3.7 Hz, 1H, 8β-H ). Details of the requested information can be found in the Materials and Methods part for each compound.
In this spectrum, two values for deuterated chloroform appear; however, the second one does not correspond to δ 7.26, and based on the alignment of the main spectrum, it cannot be justified as the deuterated solvent.
In Figure S1 we present the 1H-NMR spectrum of 6-O-(3-tert-butyldiphenylsilyloxypropyl)-6-O-desmethyl-3-O-trityl-diprenorphine (24b). For better clarity of presentation, we also inserted some enlarged spectrum parts. The first residual signal of the NMR solvent (7.26 ppm/CDCl3) belongs to the whole spectrum (0.00 – 8.50 ppm) on the far-left side. The «second deuterated CDCl3 signal» (7.26 ppm/CDCl3) is of course the same signal as described below, but belonging now to the enlarged spectrum part (7.1–7.8 ppm). There is only one reference signal.
Article, page 16, and Spectrum S1 (SM): The signal at δ 3.59–3.67 (m, 2H) shows an integration corresponding to 4 protons in the spectrum, which should be revised. (see attachment in pdf)
We have corrected this section to read: 3.59-3.80 (m, 4H, 6-O-CH2CH2CH2OTBDPS and 6-O-CH2CH2CH2OTBDPS),
The signals at δ 2.85 (d, 2J10β,10α = 18.3 Hz, 1H, 10β-H) and δ 2.89 (d, 3J9α,10α = 6.4 Hz, 1H, 9α-H) require a review of both chemical shifts and integration values.
The protons of the B-ring, in benzylic position (10α-H and 10β-H) and 9α-H, compose a three-spin system. 10β-H appears as a doublet at 2.85 ppm with a geminal coupling (2J10β,10α = 18.3 Hz) to 10α-H. 10α-Proton occurs at 2.04 ppm as doublet of doublets (geminal coupling to 10β-H and vicinal coupling to 9α-H [3J9α,10α = 6.4 Hz]. In our case, the 10β-proton and 9α-H are overlapping. Thus, we believe that we have described these signals correctly: 2.85 (d, 2J10β,10α = 18.3 Hz, 1H, 10β-H), 2.89 (d, 3J9α,10α = 6.4 Hz, 1H, 9α-H).
The mass spectra for all derivatives were not presented and should be included.
We respectfully request that you waive the request regarding the high-resolution mass spectra of the intermediates. We performed HRMS measurements for all relevant compounds (potential PET precursors: 26b–d, potential PET reference substances/OR ligands 28b–d, and OR-ligands 29b–d). Based on our previous experience. we considered it unlikely that molar peaks would be detected for compounds bearing the TBDPS protecting group.
These derivatives are in most cases very unstable (see our previous study 1). The reaction of TDDPN («Luthra precursor») with (2-bromoethoxy) tert-butyldiphenylsilane (page 348 in Marton et al.) did not result in full conversions even after 24–48 h. Depending on the reaction conditions, the starting material TDDPN was recovered (15–37 %), while the desired TBDPS-oxoetyl-3-O-trityl-DPN was isolated in a low yield (4–12 %). Mass spectroscopic analysis of TBDPS-oxoetyl-3-O-trityl-DPN by the TSP method gave [M-239]+ (upon loss of TBDPS) as the only detectable fragment, with no sign of the expected molecular peak ion [M+1]+. In the current study, we consider the 1H and 13C-NMR spectra and their whole assignation to be sufficient evidence for the presence of these intermediates. Verification was also obtained by chemical reaction.
[1] Marton, J.; Cumming, P.; Bauer, B.; and Henriksen G. A New Precursor for the Radiosynthesis of 6-O-(2-[18F]Fluoroethyl)-6-O-desmethyl-diprenorphine ([18F]FE-DPN) by Nucleophilic Radiofluorination. Lett. Org. Chem. 2021, 18, 5, 344–352. DOI: 10.2174/1570178617999200719153812.

Reviewer 2 Report
Comments and Suggestions for Authors
Summary:
The manuscript describes the synthesis of novel 6-O-fluoroalkyl and 6-O-hydroxyalkyl diprenorphine analogues together with in silico docking and MD simulations. The synthetic work is thorough, with clear routes and full spectroscopic characterization. The computational studies complement the chemistry and provide useful insight into receptor state selectivity.
Strengths:
Combines organic synthesis with molecular modeling in a coherent way.
Well-documented NMR/HRMS data.
Figures and schemes are clear.
Weaknesses / Suggestions:
Introduction is somewhat too long and could be streamlined.
Chemistry section would benefit from a concise summary table of yields and conditions.
Discussion could better highlight biological or PET imaging implications.
Minor typographical and formatting issues should be corrected.
Recommendation:
Accept after minor revisions.
Author Response
Answers to referee-2
The manuscript describes the synthesis of novel 6-O-fluoroalkyl and 6-O-hydroxyalkyl diprenorphine analogues together with in silico docking and MD simulations. The synthetic work is thorough, with clear routes and full spectroscopic characterization. The computational studies complement the chemistry and provide useful insight into receptor state selectivity.
Strengths:
Combines organic synthesis with molecular modeling in a coherent way.
Well-documented NMR/HRMS data.
Figures and schemes are clear.
Weaknesses / Suggestions:
- Introduction is somewhat too long and could be streamlined.
There are conflicting opinions regarding our «review-style» in presenting the historical development/background of our research topic. We maintain our view in that the manuscript should provide an overview of previous studies to facilitate contextualisation.
- Chemistry section would benefit from a concise summary table of yields and conditions.
Thank you for your suggestion! We gladly accepted your recommendation and added Table S3 to the Supplementary Materials to summarize our results with yields, physical constants (melting points) and quantities of the synthesized derivatives.
Table S3. Reaction conditions, yields, and physical constants for compounds prepared
|
Method |
Educt |
Product |
Reagent |
T [°C] |
t |
m [mg] |
Yield [°C] |
mp [°C] |
|
3.2.1 |
|
|
|
|
|
|
|
|
|
|
23 |
24b |
34a |
RT |
20 h |
470 |
40 |
98–102 |
|
|
23 |
24c |
34b |
RT |
20 h |
419 |
71 |
77–88 |
|
|
23 |
24d |
34c |
RT |
20 h |
573 |
61 |
70–81 |
|
3.2.2 |
|
|
|
|
|
|
|
|
|
|
24b |
25b |
1M Bu4NF |
RT |
4 h |
254 |
84 |
110–121 |
|
|
24c |
25c |
1M Bu4NF |
RT |
4 h |
340 |
93 |
105–116 |
|
|
24d |
25d |
1M Bu4NF |
RT |
4 h |
385 |
90 |
101–109 |
|
3.2.3 |
|
|
|
|
|
|
|
|
|
|
25b |
26b |
Tos2O |
RT |
4 h |
349 |
60 |
87–102 |
|
|
25c |
26c |
Tos2O |
RT |
4 h |
337 |
87 |
80–98 |
|
|
25d |
26d |
Tos2O |
RT |
4 h |
311 |
67 |
77–88 |
|
3.2.4 |
|
|
|
|
|
|
|
|
|
|
23 |
27b |
33a |
RT |
20 h |
310 |
43 |
80–105 |
|
|
23 |
27c |
33b |
RT |
20 h |
440 |
60 |
93–102 |
|
|
23 |
27d |
33c |
RT |
20 h |
487 |
66 |
82–92 |
|
3.2.5 |
|
|
|
|
|
|
|
|
|
|
27b |
28b |
AcOHa |
RT |
5 min |
129 |
78 |
98–115 |
|
|
27c |
28c |
AcOHa |
RT |
5 min |
190 |
71 |
81–96 |
|
|
27d |
28d |
AcOHa |
RT |
5 min |
220 |
76 |
81–90 |
|
3.2.6 |
|
|
|
|
|
|
|
|
|
|
25b |
29b |
AcOHb |
100 |
10 min |
120 |
77 |
208–210 |
|
|
25c |
29c |
AcOHb |
100 |
10 min |
145 |
77 |
209–210 |
|
|
25d |
29d |
AcOHb |
100 |
10 min |
105 |
72 |
86–116 |
|
|
|
|
|
|
|
|
|
|
a 27:7 (v/v) acetic acid-water mixture, 100 °C; b 4:1 (v/v) acteic acid-water mixture
- Discussion could better highlight biological or PET imaging implications.
In silico investigations. According to docking energy calculations, all compounds show a preference for the active receptor state.
|
Actives state |
series 28 is slightly better than series 29 |
in series 29: longer chain is somewhat better |
|
Inactive state |
series 28 is slightly better than series 29 |
longer chains are somewhat better. |
|
Partially active state |
series 28 is slightly better than series 29 |
longer chains are somewhat better. |
|
Root mean square deviation (RMSD): |
series 29: lowest values were observed in the active receptor state |
The rank order is active > inactive > partially active; series 28: the trend is as for series 29. The most stable in the active state were 28a and 28b.
|
|
Center of mass (COM) |
displacement of COM is generally the least in the active and partially active receptor states |
Smallest displacement of COM observed: active state 28; inactive state 28, 28b, 28c; partially active state: 28a |
Our conclusion is that 28a is the lead compound, but is not the only candidate for further pharmacological investigations. Compounds 28b and 28c may also be promising.
Biological
Despite the prevailing assumption that the N17-cyclopropylmethyl group imparts antagonistic effect of semi-synthetic opiates, a growing number of compounds contradicting this paradigm1,2. It is especially difficult to predict the agonist/antagonistic properties of the OR-ligands in the field of 6,14-ethenomorphinans. For example, the C-ring bridged morphinan (20R)-N17-cyclopropylmethyl-noretorphine with antagonistic N17-substituent is a pure OR agonist and is approximately 1000 times more potent than morphine as an analgesic3. We plan to perform the complete biochemical/pharmacological characterization of the OR ligands synthesized in this study (28a–d, 29a–d).
PET imaging
We are convinced that our recently published results for the optimization of the radiosynthesis of [18F]fluoroethyl-diprenorphine4 shall facilitate the preparation of the novel radiotracers ([18F]28b–d) by the nucleophilic one-pot, two-step radiosynthesis in an SN2 reaction.
- Minor typographical and formatting issues should be corrected.
Our native English-speaking co-author (P.C.) have carefully proofread the revised version of the manuscript.
Recommendation: Accept after minor revisions.
We are grateful for your support and kind words.
[1] Zhang, Y. et al. Bioorg. Med Chem. Lett. 2013, 23, 13, 3719–3722. Doi: 10.1016/j.bmcl.2013.05.027
[2] Greiner, E. et al. J. Med. Chem. 2003, 46, 9, 1758–1763. Doi: 10.1021/jm021118o
[3] Casy, A. F.; Parfitt, R. T. Opioid Analgesics, Chemistry and Receptors, Plenum Press New York, 1986, page 80.
[4] Németh, E. et al. Int. J. Mol. Sci. 2023, 24, 13152; doi: 10.3390/ijms241713152
